# Hunger shifts attention and attribute weighting in dietary choice

Jennifer March*, Sebastian Gluth

Department of Psychology and Hamburg Center of Neuroscience, University of Hamburg, Hamburg, Germany

## eLife Assessment

This is an **important** study showing that people who are hungry (vs. sated) put more weight on taste (vs. health) in their food choices. The experiment is well-designed and includes choice behavior, eye-tracking, and state-of-the-art computational modeling, resulting in **compelling** evidence supporting the conclusions.

**Abstract** Hunger is a biological drive which can promote unhealthy dietary decisions. Yet, the cognitive mechanisms underlying this effect, and in particular the interactive role of attention and choice processes, remain elusive. To address this gap, we conducted an eye-tracking experiment, in which 70 participants completed a multi-attribute food choice task in hungry and sated states. Confirming our preregistered hypotheses, participants' preference for tasty over healthy food items was amplified by hunger. Attention mediated this influence of hunger, as hungry participants focused more on tasty options, leading them to make less healthy decisions. Rigorous model comparisons revealed that an extension of the recently proposed multi-attribute attentional drift diffusion model best explained choice and response times. According to this model, hunger did not only increase the relative taste compared to health weight, but it also increased the fixation-related discounting of health but not taste information. Our results suggest that the cognitive mechanisms underlying unhealthy dietary decisions under hunger are characterized by a nuanced interplay between attention and the significance assigned to the options' underlying attributes.

*For correspondence: jennifer.march@uni-hamburg.de

Competing interest: The authors declare that no competing interests exist.

## Introduction

Throughout a single day, we make numerous food choices. These choices are largely influenced by the food and its environment, as well as by the decision maker's trait and state factors (*Chen and Antonelli, 2020*). For example, it has been shown that health information such as nutritional scores on food options (*Rramani et al., 2020*) or health primes (*Hare et al., 2011*; *Sullivan and Huettel, 2021*) can increase the number of healthy choices. On the other hand, a hungry decision maker is more likely to make unhealthy decisions (*Cheung et al., 2017*; *Hoefling and Strack, 2010*). Evolutionarily, a preference for energy dense foods was adaptive and ensured survival under conditions of scarcity (*Hanßen et al., 2022*; *Mattson, 2019*). While the food environment in Western societies has become increasingly obesogenic, with high caloric food options being affordable and easily available, the neurobiological mechanism continues to reward the consumption of energy dense foods contributing to a global surge in obesity rates (*Mitchell et al., 2011*; *World Health Organization, 2021*; *Lobstein et al., 2023*). The critical involvement of reward circuitries in the brain in determining food choice highlights the importance of cognitive affective drivers, alongside homeostatic ones, in shaping food-related behavior (*Plassmann et al., 2022*; *Rangel, 2013*). Here, we set out to shed light on these

**eLife digest** Many of us will recognise the impact that hunger can have on our food choices, such as making less healthy decisions when grocery shopping. From a scientific point of view, it is an evolutionary advantage to select calorie-rich food when resources are scarce. However, this natural behaviour can lead to unhealthy habits in societies where such foods are easily available.

Dietary choices are based on signals from the body as well as complex cognitive processes. Previous studies have identified that hunger drives increased attention towards foods high in calories. However, it is not clear how this process happens and how it influences decisions.

To address this, March and Gluth studied 70 people who completed a food choice task while either hungry or full. In each trial, participants chose between two foods based on an image (to convey taste) and nutritional scores (to convey health). As expected, hunger shifted choices towards the more appetising and less healthy foods.

Eye-tracking data measured during the experiment revealed that hungry participants spent more time looking at the food images than the nutritional information. March and Gluth then used computational modelling to confirm these findings, showing that hunger increases how strongly people value taste over health and decreases how much they consider nutritional information.

The findings provide new insights into the significant role of attention and valuation in food choices. This could help to advance research on improving dietary habits, which is increasingly important given rising global rates of obesity. The methods used by March and Gluth may also contribute to wider research into how attention to different factors shapes decisions.

cognitive mechanisms underlying food choice which drive energy intake and weight, by investigating the effect of hunger on attention and valuation processes in multi-attribute dietary choice.

Consistent with the evolutionary mechanism that reinforces high-energy dense food options, behavioral (*Cameron et al., 2014*; *Cheung et al., 2017*; *Epstein et al., 2003*), and neuroimaging studies (e.g. *Banica et al., 2023*; *Dagher, 2012*; *Malik et al., 2008*) indicate that under hunger (high-caloric) food options are viewed more rewarding, are more frequently chosen over healthy alternatives, and draw more attention. Meta-analyses have revealed an attentional bias towards food versus neutral stimuli, which was further amplified by hunger state (*Hardman et al., 2021*; *Pool et al., 2016*). Given these findings, it appears critical to thoroughly understand the interplay between attention and decision-making processes in shaping maladaptive food choices under hunger. To better explain the mechanisms by which hunger affects attention and valuation processes in dietary choice, we leverage recent advances in modeling attentional dynamics in the accumulation of evidence in decision-making (*Gluth et al., 2020*; *Krajbich et al., 2010*; *Shimojo et al., 2003*). This work has provided evidence for a strong positive association between the time people spend looking at a (food) option and the probability with which they choose it (*Krajbich, 2019*). Recently, these models have also incorporated the distinct attentional influence of the options' underlying attributes such as taste and health (*Fisher, 2021*; *Yang and Krajbich, 2023*). To the best of our knowledge, there is no study modeling attention and choice dynamics under different hunger states leaving the cognitive and attentional mechanisms underlying hunger-driven food choice unknown.

To fill this gap, we conducted a within-subject experiment, in which 70 participants completed a binary food choice task in hungry and sated states while their eye movements were being recorded (*Figure 1*). The considered attributes of the binary options were taste and health as represented by food images and their nutritional scores, respectively. Confirming our preregistered hypotheses, participants were more likely to choose tasty over healthy food items, and this difference was amplified under hunger. Notably, attention mediated the influence of hunger on dietary decisions, as participants focused more on taste information under hunger, leading them to make less healthy decisions. To better understand the cognitive mechanisms underlying hunger-driven dietary choice, we implemented different variants of the diffusion decision model (DDM, *Ratcliff, 1978*), which included the consideration of both attributes (*Maier et al., 2020*; *Sullivan and Huettel, 2021*) and the incorporation and extension of attentional mechanisms (*Fisher, 2021*; *Krajbich et al., 2010*; *Yang and Krajbich, 2023*). Critically, we extended the recently proposed multi-attribute attentional DDM (*Yang and Krajbich, 2023*) to allow the discounting of unattended information to differ across different

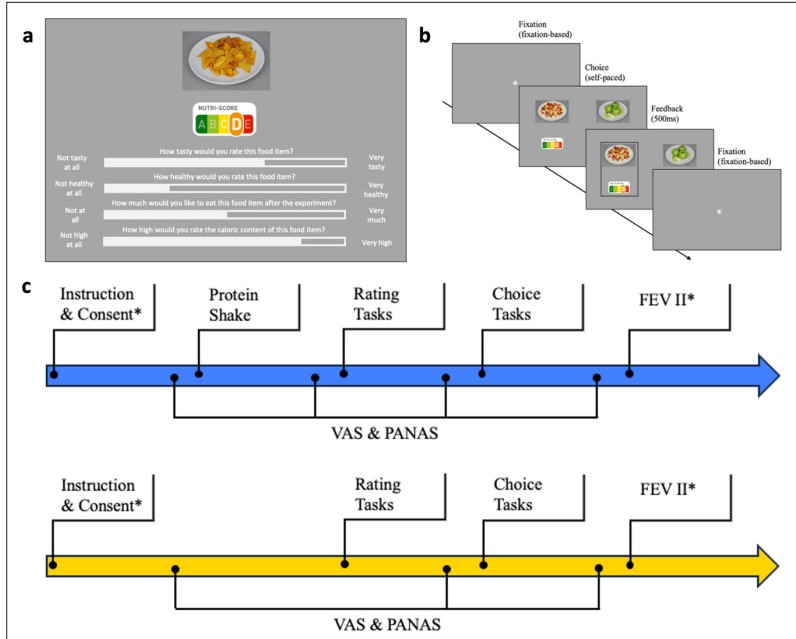

**Figure 1.** Experimental design. (**a**) Food rating task. Participants rated all food images and their corresponding Nutri-Scores (see *Methods*) in terms of taste, health, wanting, and perceived caloric content on a continuous scale (**b**) Trial sequence of food choice task. In each trial, participants made a binary choice between two food options represented by food image and corresponding Nutri-Scores; Feedback and fixation-based fixation dots were implemented (**c**) Experimental procedures; blue refers to sated, yellow to hungry condition (order counterbalanced). VAS refers to visual analog scale used to assess subjective feelings of hunger. Positive and negative affect scale (PANAS) refers to a questionnaire assessing mood (see *Appendix 1*). FEV II refers to a questionnaire assessing eating behavior (see *Appendix 2*); *indicates that these steps were only required in the first session.

attributes (here: taste vs. health). This model not only provided the best account of our behavioral data, but also revealed a twofold mechanism, wherein hunger affects valuation of choice options by shifting the relative weighting of taste information and by exacerbating the attentional discounting of health (but not taste) information.

## Results

We used a within-subject experiment, in which 70 participants were tested in hungry and sated conditions in counterbalanced order. In the sated condition, participants received a Protein Shake, with its size being determined by their metabolic rate (see *Methods*). The experiment consisted of a food rating and a multi-attribute binary food choice task, as well as control measures including hunger state, mood, and eating behavior (*Figure 1*; see *Methods* for details).

### Hunger state manipulation

First, we tested whether the manipulation of hunger state was successful (*Figure 2a*). Upon arrival at the lab, participants' hunger ratings did not differ between the stated condition ($M_{satedt1} = 51.98$, $SD_{satedt1} = 27.54$) and the hungry condition ($M_{hungryt1} = 57.99$, $SD_{hungryt1} = 23.54$, $t(63)=-1.265$, p=0.211, d=0.159). The RM-ANOVA indicated that the change in hunger ratings between the last and first time point differed across conditions ($F(1)=26.31$, p<.001, d=0.708). Specifically, in the hungry condition the change was positive, meaning participants got hungrier throughout the experiment ($M_{hungrydiff} = 22.4$, $SD_{hungrydiff} = 20$), whereas in the sated condition, this difference was negative ($M_{sateddiff} = -36.3$, $SD_{sateddiff} = 31.3$). Thus, our hunger state manipulation had the desired effect on the subjective feeling of hunger. Notably, there were no effects of hunger state on positive and negative affect across timepoints (*Appendix 1—figure 1*).

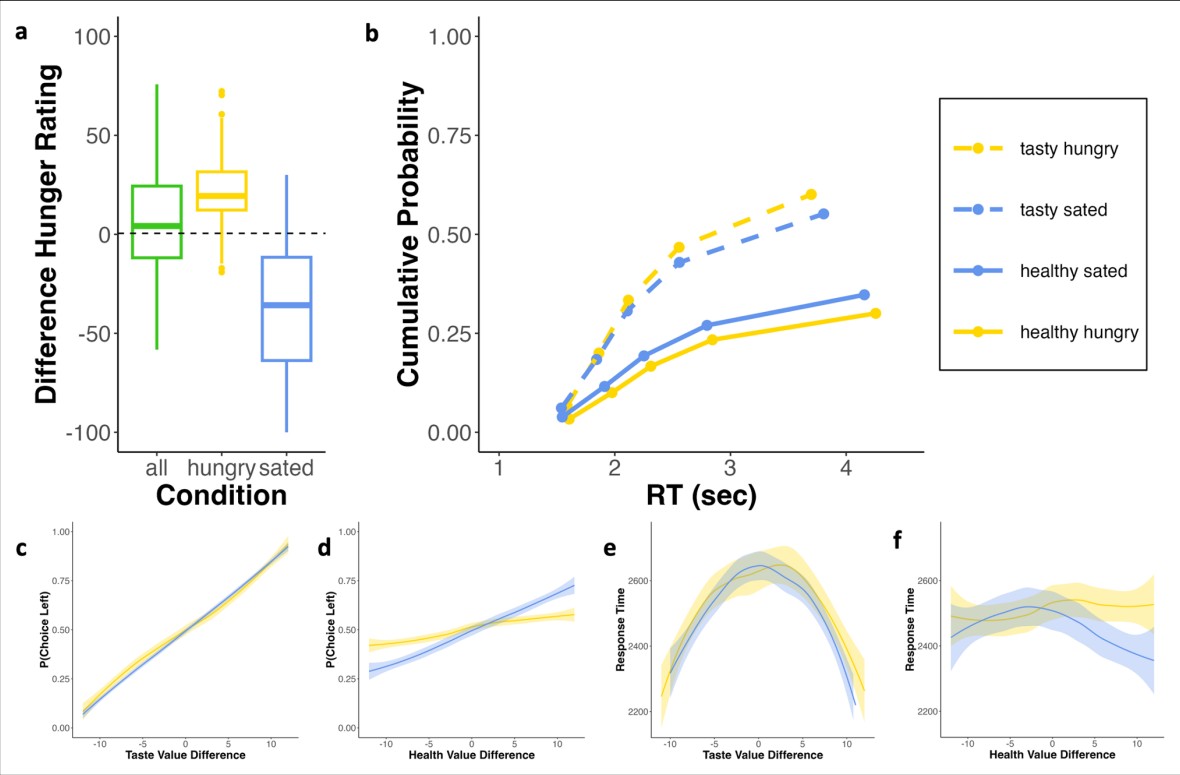

**Figure 2.** Behavioral results. (**a**) Manipulation check: The green boxplot displays the difference (hungry-sated) in hunger state at arrival at the lab, yellow and blue boxplots display the difference (last timepoint-first timepoint) in hunger state in the hungry and sated condition, respectively. (**b**) Response time (RT) quantile plot displaying the cumulative probability of tasty (dashed lines) and healthy choices (solid lines) separately for the two conditions (quantiles are 0.1, 0.3, 0.5, 0.7, 0.9 of choices). (**c, d**) Probability to choose the left option as a function of taste and health value difference (left-right), respectively. Importantly, the dependency of choice on health information was eliminated under hunger. (**e, f**) Corresponding mean RTs as a function of taste and health value difference, respectively. For illustration purposes, value differences were segmented into 25 bins, and a locally weighted scatterplot smoothing technique was applied with a span of 0.75. Plots (**c–f**) are based on all trials. Transparent shades indicate the standard errors of the smoothed choice probability and RT for the respective value bins (see also *Figure 2—figure supplement 3*).

The online version of this article includes the following source data and figure supplement(s) for figure 2:

**Figure supplement 1.** Principal component analysis.

**Figure supplement 2.** Factor loadings on the components for the respective datasets.

**Figure supplement 2—source data 1.** Source table.

**Figure supplement 3.** Quantile plots based on wanting and caloric information.

**Figure supplement 4.** Choice as a function of value difference.

## Drivers of food choice

For each food item, we had six measures: Nutri-Score, four subjective ratings (taste, health, wanting, estimated caloric content), and objective caloric content. To assess whether our preregistered goal to study dietary decisions in terms of contrasting taste vs. health aspects was justified, we performed a principal component analysis (PCA) on these measures. Results revealed that 81% of variance was explained by two components, the first loading positively on caloric information (subjective and objective) and negatively on health information (subjective health rating and Nutri-Score), while the second one loaded positively on taste and wanting (*Figure 2—figure supplements 1 and 2*). Importantly, loadings of taste measures on the health component and loadings of health measures on the taste component were low suggesting independence of these factors. As the PCA clearly suggested our different measures to be linked to participants' decisions by two main components that represent health and taste aspects, respectively, we focus on tasty vs healthy decisions in our main behavioral and modeling analyses.

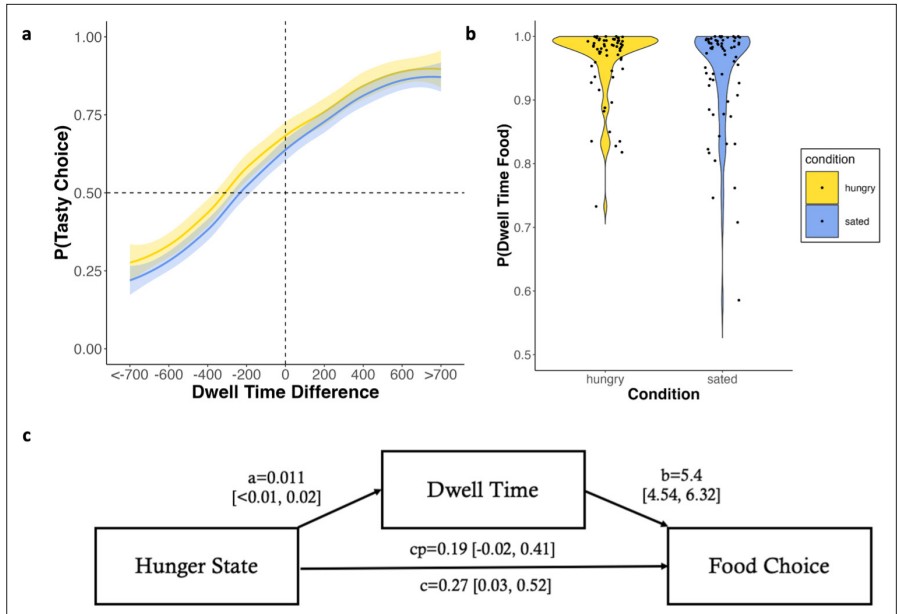

**Figure 3.** Eye-tracking results. (**a**) Dwell time difference between the tasty and healthy option was positively associated with the probability of choosing the tasty option in both conditions. (**b**) The average probability to look at food image (taste attribute) compared to Nutri-Score (health attribute) was even higher in the hungry than sated condition. (**c**) Path diagram with posterior means of the parameters, associated 95%-credible interval in squared brackets.

The online version of this article includes the following source data and figure supplement(s) for figure 3:

**Figure supplement 1.** Proportion of first and last fixations.

**Figure supplement 2.** Mediation coefficients.

**Figure supplement 2—source data 1.** Source table.

**Figure supplement 3.** Standard deviations of subject-level effects (random effects), their covariances, and correlations.

**Figure supplement 3—source data 1.** Source table.

## Effect of hunger state on choice and RT

In both conditions, a larger value difference (VD) with respect to taste was predictive of tasty choice (*Figure 2c*), while a larger VD with respect to health was predictive of healthy choice, particularly in the sated condition (*Figure 2d*). The GLMM of choice (tasty vs. healthy) indicated that overall participants preferred tasty over healthy options ($\beta_{intercept} = 0.73$, SE = 0.098, p<0.001). In line with our preregistered hypotheses, we found that participants were less likely to choose the tasty option when being sated as compared to hungry ($\beta_{sated} = -0.211$, SE = 0.103, p=0.04). Moreover, longer relative dwell time on the tasty option increased the likelihood of tasty choice ($\beta_{dwelltime} = 0.998$, SE = 0.027, p<0.001) (*Figure 3a*) (see *Appendix 4—table 1* for model specifications, random effects, and an alternative model with additional predictors).

Importantly, we assessed the robustness of our findings by testing alternative GLMMs to predict choices, in which we replaced the taste/health ratings as predictors by wanting/health ratings (*Appendix 5—table 1*) and by high/low caloric information (*Appendix 6—table 1*). Of note, the effects of hunger on higher wanted vs healthy and high vs low caloric choice were markedly stronger (*Appendix 5—table 1*, *Appendix 6—table 1*, *Figure 2—figure supplement 3*).

With respect to response time (RT), we found that RT was highest for choices in which taste ratings were similar for both options (*Figure 2e*), while health value did not affect RT (*Figure 2f*). The GLMM of RT indicated an average RT of 2.748 s (SE = 0.096). Tasty choices were associated with faster decisions, decreasing RT by 0.15 s (SE = 0.018, p<0f.001). Longer relative dwell time on the tasty option predicted slower choice in general ($\beta_{dwelltime} = 0.065$, SE = 0.014, p<0.001), but was sped-up for tasty

choices ($\beta_{dwelltime*tastychoice} = -0.13$, SE = 0.017, p<0.001) (see **Appendix 4—table 2** for model specifications, random effects, and an alternative model with additional predictors).

Altogether, we found that participants preferred tasty over healthy options, and that this preference was amplified under hunger. While tasty choices were faster in general, we did not find an effect of hunger state on RT. Finally, our GLMMs indicate that dwell time is an important predictor of choice and RT.

## Hunger affects attention and dietary choice

In line with previous work (**Gluth et al., 2018**; **Krajbich et al., 2010**; **Weilbächer et al., 2021**; **Yang and Krajbich, 2023**), our choice GLMM indicated that looking longer at the tasty option predicted tasty choice. This effect was observed in both conditions to a very similar degree (**Figure 3a**). When analyzing dwell time on the attribute level, however, there was a significant condition difference: Although participants were much more likely to look at food images (taste attribute) than the Nutri-Scores (health attribute) in both conditions, this difference was even more pronounced in the hungry compared to the sated state ($t$(69)=2.595, $P$=.006, $d$=0.312; **Figure 3b**). This effect remained significant after excluding outlier data ($t$(68)=2.392, $P$=.01, $d$=0.29). First and last fixations and transition patterns are shown in **Figure 3—figure supplement 1**.

The analysis so far suggests that dwell time depends on hunger state (**Figure 3b**) and is predictive of choice (**Figure 3a**). To better understand these interactions, we conducted a hierarchical Bayesian mediation analysis testing whether attention (i.e. dwell time) mediates the relationship between hunger state and food choice (**Figure 3C**, **Figure 3—figure supplements 2 and 3**). In line with our GLMM on choice, the direct path between hunger state and food choice was significant ($M_c = 0.27$, $SE_c = 0.12$, $CI_c = [0.03, 0.52]$), meaning hungry individuals were more likely to choose tasty options. Similarly, the path between attention and food choice was significant ($M_b = 5.41$, $SE_b = 0.45$, $CI_b = [4.54, 6.32]$), indicating that longer dwell times on the tasty option were predictive of choosing that option. Furthermore, there was a small yet significant relationship between hunger state and attention ($M_a = 0.01$, $SE_a = 0.01$, $CI_a$=[<0.001, 0.022]), demonstrating that hungry individuals paid relatively more attention to tasty options. Critically, our mediation analysis revealed that the direct path

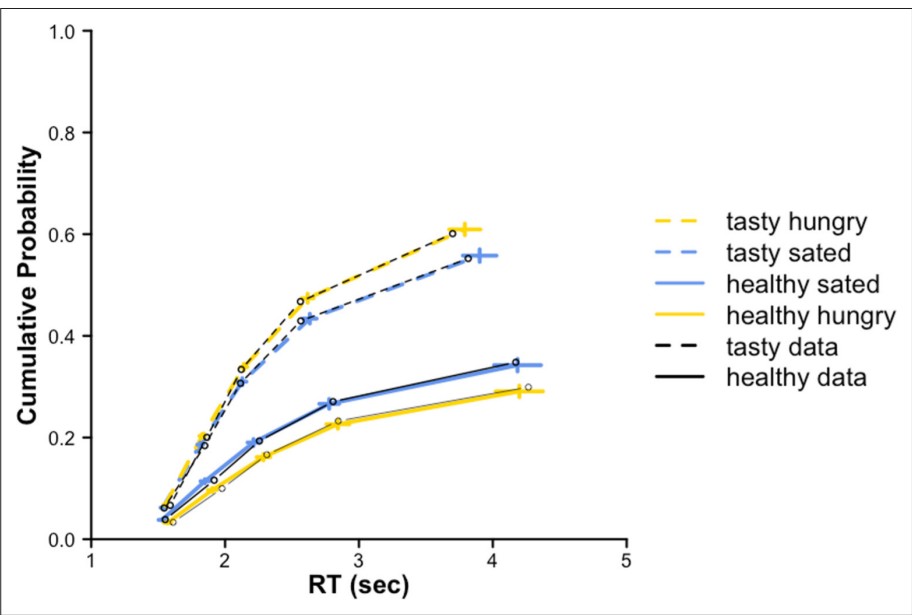

**Figure 4.** Posterior predictive checks maaDDM2 $\phi$. Quantile plots of simulated data with fitted parameters of the maaDDM2 $\phi$ in blue (sated) and yellow (hungry) with highest density intervals (HDI) of each quantile (vertical lines) and behavior. Posterior predictive checks were performed by drawing 1000 parameter values from the individual posterior parameter distribution to simulate the new data.

The online version of this article includes the following figure supplement(s) for figure 4:

**Figure supplement 1.** Posterior predictive checks for multi-attribute attentional DDM (maaDDM).

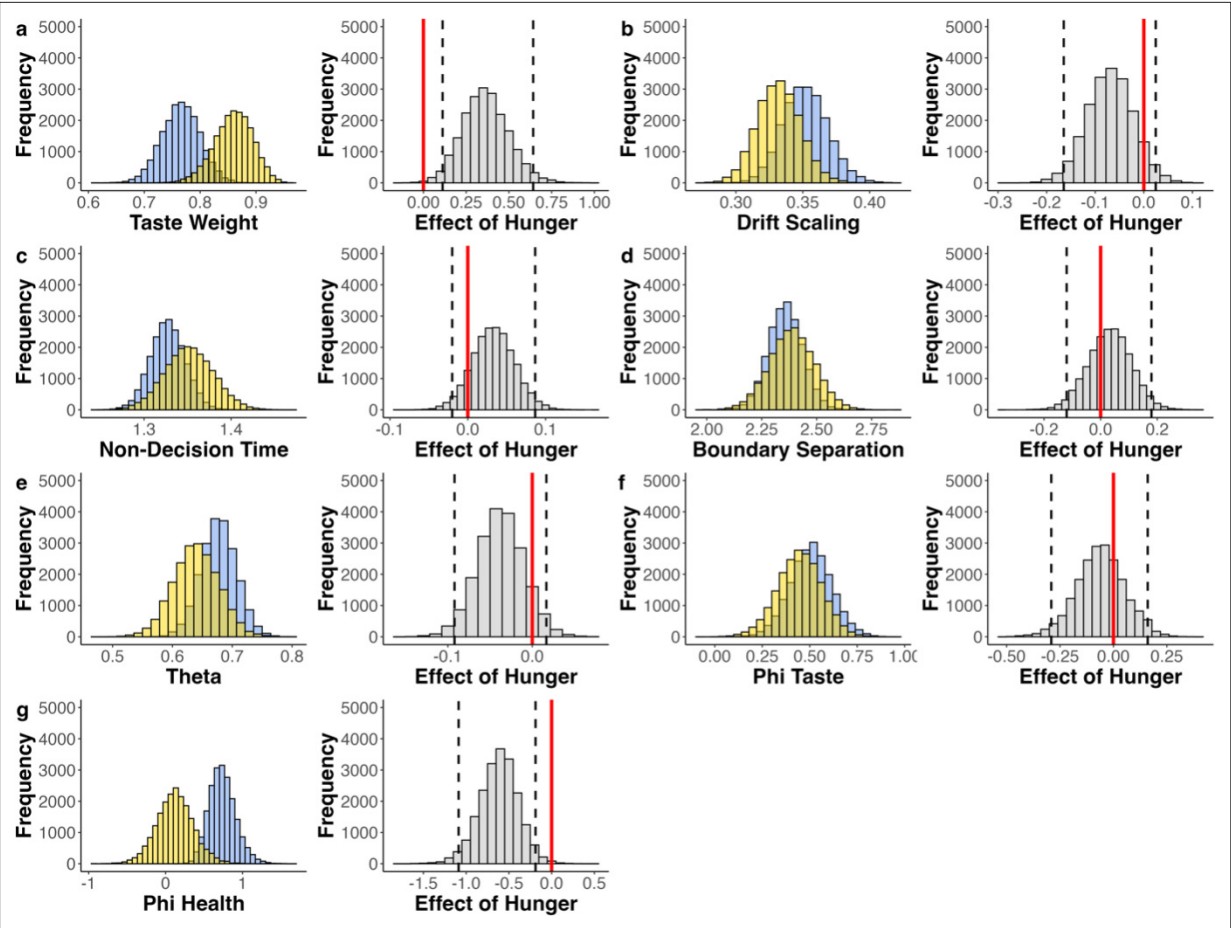

**Figure 5.** Parameter estimates of maaDDM2 $\phi$. Group parameter estimates (blue = sated, yellow = hungry; left panels) and the effect of hunger state (gray; right panels). Dashed black lines indicate the 95% HDI. (**a**) Estimated taste weights. In both conditions the weight is larger than 0.5, indicating a higher weight on taste compared to health. This preference was even stronger under hunger. (**b–f**) Parameter estimates of $d$, nDT, $\alpha$, $\theta$ and $\phi_T$, and the corresponding effects of hunger state. (**g**) Parameter estimates of $\phi_H$ and the corresponding effects of hunger state, showing that the attention-driven discounting of health information was amplified under hunger.

The online version of this article includes the following figure supplement(s) for figure 5:

**Figure supplement 1.** Fitted parameters of multi-attribute attentional DDM (maaDDM).

between hunger state and food choice was no longer significant when attention was considered ($M_{cp} = 0.19$, $SE_{cp} = 0.11$, $CI_{cp}$=[–0.02, 0.41]), while the population-level mediation path (a*b) was significant ($M_{a*b} = 0.08$, $SE_{a*b} = 0.04$, $CI_{a*b}$=[0.01, 0.16]). Alternative mediation models (with wanting ratings or caloric information are reported in **Appendix 7—tables 1–4**).

Altogether, the eye-tracking analyses demonstrated that attention was predictive of choice, and hungry participants' preference for tasty foods was reflected in their dwell time. Finally, attention emerged as a pivotal mediator of the relationship between hunger state and food choice.

## Mechanisms underlying the effect of hunger on attention and dietary choice

In line with our hypotheses, we found that participants were more likely to choose tasty over healthy food items, and this difference was amplified by hunger (**Figure 2b**). Moreover, we demonstrated that attention mediated the effect of hunger on choice (**Figure 3c**). To further elucidate the cognitive processes underlying these effects, we estimated and compared different versions of DDMs against one another using hierarchical Bayesian cognitive modeling. Models varied in terms of whether and how they accounted for attention and whether a starting point bias (towards tasty vs. healthy options) was included (see *Methods*) **Figure 4**. We report parameter estimates for all models (**Figure 5**,

**Table 1.** Quantitative model comparison.

| Model | $\alpha$ | nDT | $d$ | $\omega$ | $\beta$ | $\theta$ | $\phi_1$ | $\phi_2$ | DIC | Rhat |
|---|---|---|---|---|---|---|---|---|---|---|
| DDM | YES | YES | YES | YES | NO | NO | NO | NO | 69646 | 1.002 |
| DDMsp | YES | YES | YES | YES | YES | NO | NO | NO | 69668 | 1.004 |
| aDDM | YES | YES | YES | YES | NO | YES | NO | NO | 65561 | 1.004 |
| aDDMsp | YES | YES | YES | YES | YES | YES | NO | NO | 65587 | 1.003 |
| maaDDM | YES | YES | YES | YES | NO | YES | YES | NO | 65155 | 1.005 |
| maaDDMsp | YES | YES | YES | YES | YES | YES | YES | NO | 65214 | 1.011 |
| **maaDDM2 $\phi$** | **YES** | **YES** | **YES** | **YES** | **NO** | **YES** | **YES** | **YES** | **64002** | **1.017** |
| maaDDM2 $\phi$ sp | YES | YES | YES | YES | YES | YES | YES | YES | 65070 | 1.027 |

The first column states the name of the model; the following nine columns indicate whether the drift diffusion model (DDM) variants included a given parameter or not. $\alpha$ refers to the boundary separation; nDT refers to non-decision time; $d$ refers to the drift scaling parameter; $\omega$ refers to the relative taste compared to health weight; $\beta$ refers to the starting point bias; $\theta$ refers to the discounting of the non-looked upon option; $\phi_1$ refers to the discounting of the non-looked upon attribute, in case the model includes $\phi_1$ and $\phi_2$ they refer to the discounting of taste and heath information, respectively; The deviance information criterion (DIC) was used as goodness-of-fit measure. **Rhat** is the scale reduction factor, to accurately predict posterior distributions, it should be 1.00, according to ***Vuorre and Bolger, 2018*** values within 0.05 are acceptable. The best model (i.e. maaDDM2 $\phi$) is highlighted in bold.

***Figure 5—figure supplement 1*** and ***Appendix 8—figures 1–6***), as well as posterior predictive checks (***Figure 4***, ***Figure 4—figure supplement 1*** and ***Appendix 9—figure 1***) and recoveries (***Appendix 10—figures 1–4***) for the best fitting models.

Initial model comparison revealed that there was no evidence for a starting point effect, as models without starting point consistently outperformed models with starting point. In addition, the multi-attribute attentional DDMs (maaDDM and maaDDM2 $\phi$), which allow modeling discounting of unattended options as well as unattended attributes, outperformed simpler variants (i.e. DDM, aDDM) (***Table 1***).

Further inspection of differences in the maaDDM's parameter estimates between the hungry and sated conditions suggested that hunger increased the weight of taste information relative to health information and exacerbated attribute-wise attentional discounting i.e., lower estimates of parameter $\phi$ (***Figure 5—figure supplement 1***). To shed more light on this attentional effect of hunger, we tested an extension of the maaDDM that assumed two separate attribute-wise discounting parameters for taste and health information (i.e. maaDDM2$\phi$, see also ***Figure 6***). Remarkably, this model provided a substantially improved model fit compared to the maaDDM and all other models (***Table 1***). Moreover, the posterior predictive checks of this model indicated that it provides an exquisite account of the choice and RT data (***Figure 4***). Again, we assessed the robustness of our results by also testing additional models in which health and taste attributes were replaced by Nutri-Scores (***Appendix 11—table 1***, ***Appendix 11—figures 1–8***) and wanting (***Appendix 12—table 1***, ***Appendix 12—figures 1–4***), respectively. Importantly, these complementary modeling analyses yielded comparable quantitative and qualitative results.

Taking a closer look at the group parameter distributions of the winning model (i.e. maaDDM2 $\phi$), we examined the highest density intervals (HDI), which reflects the part of the posterior distributions that contain 95% of all values. We found that participants relative taste weight was larger than 0.5 in both conditions, indicating a higher taste compared to health preference ($HDI_{sated}$=[0.698,0.831]; $HDI_{hungry}$=[0.788,0.922]), as the HDIs did not include 0.5. Critically, this preference was credibly higher under hunger (HDI=[0.122, 0.642]; ***Figure 5a***), as the HDI of the effect of hunger did not include 0. We did not find differences between conditions with respect to the drift scaling parameter $d$ (HDI=[–0.165, 0.025]; ***Figure 5b***), the non-decision time nDT (HDI=[–0.02, 0.087]; ***Figure 5c***), or the boundary separation $\alpha$ (HDI = [–0.12, 0.18]; ***Figure 5d***). Similarly, there were no credible hunger effects with respect to the attentional discounting of the options $\theta$ (HDI = [–0.092, 0.017]; ***Figure 5e***). Looking at the two attribute-wise attentional discounting parameters revealed that there was no condition effect on discounting of the taste attribute (i.e. $\phi_T$, HDI=[–0.291, 0.16]; ***Figure 5f***), but instead, hunger exclusively increased the discounting of health information (i.e. $\phi_H$, HDI=[-1.088,–0.188]; ***Figure 5g***).

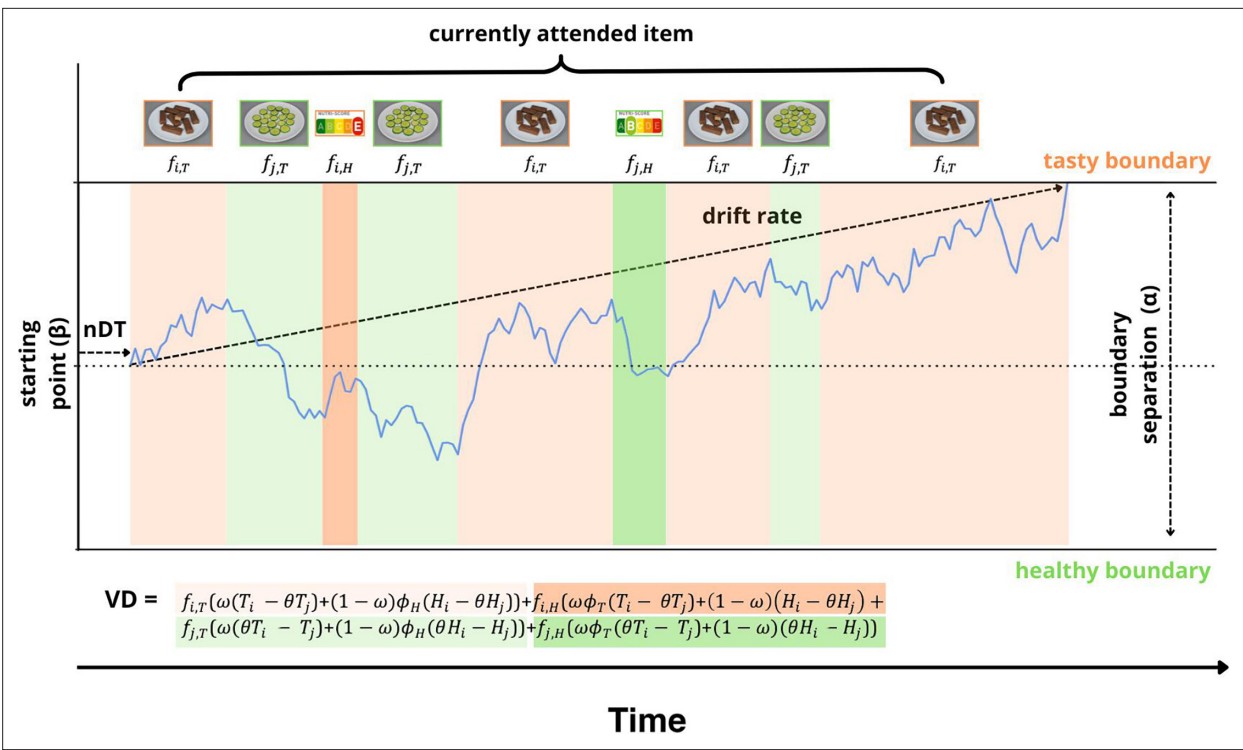

**Figure 6.** Illustration of the maaDDM2 $\phi$. The decision-making process underlying choice and response time (RT) data as conceived by the *maaDDM2$\phi$*. The decision is assumed to emerge from a noisy evidence-accumulation process commencing from the starting point ($\beta$) and terminating at one of the two boundaries (here: 0=healthy boundary and $\alpha$ = tasty boundary) representing the tasty and healthy choice, respectively. The non-decision time (nDT) reflects processes unrelated to the decision itself, here illustrated as stimulus encoding time. The drift rate represents the rate of evidence accumulation. It is determined by the scaled value difference (VD) of the displayed options, which in turn is given by the taste (T) and health (H) ratings of the options, the relative weight of tastiness $\omega$ vs. healthiness (1- $\omega$) as well as the currently attended item on the screen as illustrated by the differently colored segments and the corresponding images. The coloring scheme of the VD equation shows which part of the equation defines the drift rate at any given attended item. Attending to the tasty option (here: chocolate bar with Nutri-Score E), and in particular to its taste information (i.e. the image), increases the drift towards the tasty boundary (orange), while attending to the healthy option (here: cucumber with Nutri-Score B), and in particular to its health information (i.e. the Nutri-Score) increases the drift towards the healthy boundary (green).

Taken together, our extension of the multi-attribute attentional DDM with separate attention parameter for taste and health attributes (i.e. maaDDM2 $\phi$) provided the best quantitative and an excellent qualitative account of the data, and it suggests that hunger affects the relative weighting of taste compared to health information and further increases the discounting of unattended health information during the evidence-accumulation process.

## Discussion

The goal of this study was to elucidate the cognitive mechanisms driving dietary choice under hunger. We found that individuals prefer tasty over healthy food options, and that this preference is amplified by hunger state. This pattern was also reflected in our modeling analyses, revealing that taste was weighted more than health in both, but especially in the hungry condition. Our mediation analysis suggests that the cognitive mechanism underlying the influence of hunger state on food choice is driven by a shift in attention. Specifically, hungry individuals pay more attention to tasty food options in general and the taste attribute in particular, which in turn increases the probability of tasty choices. Again, our cognitive modeling analyses integrated these findings, demonstrating increased attentional discounting of the health attribute under hunger. Together, our findings suggest a nuanced interplay between attention and the significance assigned to the options' underlying attributes in dietary decision-making.

First, in line with previous research (*Cheung et al., 2017*; *Otterbring, 2019*; *Read and Leeuwen, 1998*) we demonstrate that hunger affects dietary choice. Participants were more likely to choose

options that were rated more tasty than healthy, for which reported higher wanting, and that contained higher caloric content. Moreover, our findings indicate that higher taste ratings were strongly predictive of choice across conditions, whereas higher health ratings only predicted choice in the sated condition, albeit less influential than taste ratings (*Figure 2c and d*). Crucially, our modeling analyses endorsed this account: across models, we demonstrate that the relative taste weight was larger in both conditions and particularly in the hungry condition. This finding adds to previous work demonstrating the distinct influence of taste and health attributes in guiding choice (*Barakchian et al., 2021*; *Enax et al., 2016*; *Hutcherson and Tusche, 2022*; *Maier et al., 2020*; *Rramani et al., 2020*), by illustrating that these effects can differ across states. This also aligns with results from a food bidding task (*Fisher and Rangel, 2014*), wherein the authors find that hunger elevated the bids and speculate that this effect was driven by an increased taste but not health valuation of food items.

Second, we show that the valuation is not only influenced by the attributes underlying decision weights, but moreover by attention. Behaviorally, we show that attention mediates the impact of hunger state on food choice, such that hunger state predicts overall less dwell time on the healthy option, thereby increasing the probability of tasty choice. Upon closer examination of attention allocation to the respective attributes, we show that participants, especially when hungry, spent much more time on food images compared to Nutri-Scores. We see two explanations for the excessively high proportion of dwell time on food images compared to Nutri-Scores: First, food images were more important for deciding, as indicated by the relative weight parameter being larger than 0.5. Consequently and in line with previous work (*Orquin et al., 2021*), people pay more attention to more important attributes. Second, food images contain more complex information. Whereas extracting taste information from a food image can be seen as a complex inference process, a (color-coded) nutritional score provides salient and easily discernible evidence about a product's healthiness and consequently requires less dwell time. This account is supported by studies showing that nutritional scores can promote healthy choice despite proportionally little dwell time (e.g. *Bialkova et al., 2014*; *Gabor et al., 2020*; *Rramani et al., 2020*).

To take these putatively different attentional demands of different attributes into account, we extended the maaDDM of *Yang and Krajich, 2023* and developed the maaDDM2 $\phi$, which assumes separate discounting parameters for separate (unattended) attributes. Consistent with the above-mentioned attribute-specific effect of hunger on choices, we found that the attentional discounting of the health attribute but not the taste attribute was amplified under hunger, such that unattended nutritional information had a blunted influence on the evidence accumulation process. Notably, we also found a hunger effect on the attribute discounting parameter $\phi$ in the regular maaDDM (*Figure 5—figure supplement 1*), but our extension allowed us to pinpoint this effect specifically to health information. Given related work on the malleability of attentional discounting on mnemonic demands (*Eum et al., 2023*; *Weilbächer et al., 2021*), we speculate that hunger could impede a person's ability or willingness to maintain health considerations in working memory when attention is currently drawn to the tasty food stimulus.

Several supplementary analyses demonstrate the robustness of our findings. In essence, and in line with a large body of literature (e.g. *Garlasco et al., 2019*; *Otterbring, 2019*; *Read and Leeuwen, 1998*), hungry participants were more likely to choose items which they rated higher in terms of tastiness, wanting and caloric content. Importantly, the pivotal role of attention was also established in the exploratory wanting and calorie analyses. Moreover, we performed a PCA to identify the major components that drive food choices, finding that two factors representing tastiness and healthiness aspects explain 81% of the variance in the data. We see this as a justification to describe and study dietary choices by means of these two attributes, essentially following a series of previous studies in our field (e.g. *Hutcherson and Tusche, 2022*; *Maier et al., 2020*; *Rramani et al., 2020*; *Sullivan and Huettel, 2021*).

The present results and supplementary analyses clearly support the twofold effect of hunger state on the cognitive mechanisms underlying choice. However, we acknowledge potential demand effects arising from the within-subject Protein-shake manipulation. A recent study (*Khalid et al., 2024*) showed that labeling water to decrease or increase hunger affected participants subsequent hunger ratings and food valuations. For instance, participants expecting the water to decrease hunger showed less wanting for food items. DDM modeling suggested that this placebo manipulation affected both drift rate and starting point. The absence of a starting point effect in our data speaks against any prior

bias in participants due to any demand effects. Yet, we cannot rule out that such effects affected the decision-making process, for example by increasing the taste weight (and thus the drift rate) in the hungry condition.

From a neurobiological perspective, both homeostatic and hedonic mechanisms drive eating behavior. While homeostatic mechanisms regulate eating behavior based on energy needs, hedonic mechanisms operate independent of caloric deficit (*Alonso-Alonso et al., 2015*; *Lowe and Butryn, 2007*; *Saper et al., 2002*). Participants' preference for tasty high-caloric food options in the hungry condition aligns with a drive for energy restoration and could thus be taken as an adaptive response to signals from the body. On the other hand, our data shows that participants preferred less healthy options also in the sated condition. Here, hedonic drivers could predominate indicating potentially maladaptive decision-making that could lead to adverse health outcomes if sustained. Notably, our modeling analyses indicated that participants in the sated condition showed reduced attentional discounting of health information, which poses potential for attention-based intervention strategies to counter hedonic hunger. This has been investigated, for example, in behavioral (*Barakchian et al., 2021*; *Bucher et al., 2016*; *Cheung et al., 2017*; *Sullivan and Huettel, 2021*), eye-tracking (*Schomaker et al., 2022*; *Vriens et al., 2020*) and neuroimaging studies (*Hare et al., 2011*; *Hutcherson and Tusche, 2022*) showing that focusing attention on health aspects increased healthy choice. For example, *Hutcherson and Tusche, 2022* compellingly demonstrated that the mechanism through which health cues enhance healthy choice is shaped by increased value computations in the dorsolateral prefrontal cortex (dlPFC) when cue and choice are conflicting (i.e. health cue, tasty choice). In the context of hunger, these findings together with our analyses suggest that drawing people's attention towards health information will promote healthy choice by mitigating the increased attentional discounting of such information in the presence of tempting food stimuli.

In conclusion, our study provides substantial insights into the mechanism underlying dietary choice across metabolic states. Our extension of the multi-attribute attentional DDM revealed that the valuation of food options under hunger is compromised by a relatively lower weighting and a stronger attentional discounting of health information. This modeling extension represents a general contribution to advance research on multi-attribute decision-making, as it allows modeling attribute-specific attentional discounting, which is likely to occur if attributes are described in markedly different formats.

## Materials and methods
### Preregistration
The study was preregistered on Open Science Framework (https://osf.io/tmdw3/). An a-priori power analysis was conducted to determine the required sample size of the experiment using G*Power (*Faul et al., 2009*). The power analysis was targeted on testing an effect of hunger on non-food choices (which were part of the same study but are not reported here). A study by *Skrynka and Vincent, 2019* demonstrated that hunger state affected the discounting of food and other commodities, for which the authors report very large and medium-large effect sizes, respectively. Given inflated effect sizes due to publication bias (*Simonsohn, 2015*), we set our smallest effect size of interest (*Lakens et al., 2018*) to Cohen's $d$=0.3, with an alpha level of 0.05, and a power of 0.8, resulting in a required sample size for a one-tailed paired t-test of 70 participants. In line with *Skrynka and Vincent, 2019*, we expected a larger effect of hunger on food choices (i.e. $d$=0.5) and thus consider the current experiment being sufficiently powered.

### Participants
A total of 70 participants (53 females, 16 males and one diverse, $M_{age} = 25.6$, $SD_{age} = 8.064$, $M_{BMI} = 23.224$, $SD_{BMI} = 4.363$) completed both sessions of the experiment. Participants were recruited from the University of Hamburg using the recruiting system SONA (n=40) and from the city of Hamburg using the job portal Stellenwerk (stellenwerk-hamburg.de) (n=30). Compensation for participation were course credits or money (€12.50 per hour). Individuals were eligible to participate in the study if they were proficient in German and were at least 18 yr old. Exclusion criteria included dietary-related aspects (e.g. diets, vegan, food allergies, and intolerances), physical or mental illnesses, drug use, pregnancy, and breastfeeding. The Local Ethics Committee of the Faculty of Psychology and Human Movement Sciences at the University of Hamburg approved the study.

## Procedure

Before participants signed up for the study, a questionnaire was acquired to assess participants' eligibility and collect demographic information. The latter was used to compute the amount of protein shake participants received in the sated condition (see *Hunger state manipulation*). Hunger state was counterbalanced such that n=36 completed the experiment in the hungry condition first and n=34 in the sated condition first. In the first session, participants were informed about the procedure (*Supplementary file 1*) and provided their informed consent. In both sessions, participants first rated their subjective feeling of hunger (see *VAS, Appendix 3—figure 1*) and mood (see *Appendix 1*). In the sated condition, participants received a protein shake matched to their daily caloric needs (see *Hunger state manipulation*) and rated hunger and mood again. In both sessions, the experiment started with a rating task, followed by hunger and mood ratings, then the choice tasks, and concluded with subsequent hunger and mood ratings before reaching the reward screen (*Figure 1c*). At the end of the first session, participants filled out a questionnaire assessing eating behavior (see *Appendix 2*). Finally, participants were compensated and received their reward. Overall, one session lasted for approximately 2 hr. The second session took place 5–10 d (*M*=7.915, SD = 2.755) after the first.

### Hunger state manipulation

In both conditions, participants came to the lab after an overnight fast. In the sated condition, participants received an individually determined amount of whey protein shake from MyProtein (https://www.myprotein.com/) (flavor: vanilla, or strawberry) amounting to 25% of participants' daily caloric needs in line with Schofield equations (*Schofield, 1985*). The equation considers gender, age, weight, and activity level, which was set to 1.4 ('sedentary') for all participants, in line with *Wever et al., 2021*.

## Experimental tasks and materials

The experiment was implemented in OpenSesame version 3.3 (*Mathôt et al., 2012*), and PyGaze (*Dalmaijer et al., 2014*) was used for the implementation of different eye-tracking functions. Participants completed the experiment on a 24-inch screen with a resolution of 1024×768 pixels. The experiment consisted of three counterbalanced rating blocks, and corresponding choice blocks (i.e. food preference and choice, social preference and choice, intertemporal preference and choice). Here, we report the results of the food rating and food choice task only. The social and intertemporal ratings and choices will be reported separately. Stimuli of the food tasks were taken from the Full4Health Image Collection and included 66 standardized images of food presented on a plate (*Charbonnier et al., 2016*, available at https://osf.io/cx7tp/). Food images were selected based on their familiarity in Germany, and matched with respect to the Nutri-Score, which represents a rating of the nutritional quality of a food item within a product category (from A=balanced nutrition to E=unbalanced nutrition, Federal Ministry of Food and Agriculture). While familiar in Germany and other European countries, participants were also informed about meaning of the Nutri-Score before the experiment started. We included 13 food images of Nutri-Scores A and B each with approximately half sweet (e.g. kiwi) and half savory (e.g. cucumber); 12 food images of Nutri-Score C each half sweet (e.g. dried apricots) and half savory (e.g. olives); and 14 images of Nutri-Scores D and E each with approximately half sweet (e.g. Oreo biscuits) and half savory (e.g. Potato Crisps). Food images (387.2×259.2 pixels) and corresponding Nutri-Scores (166.5×94.1 pixels) were displayed in both rating and choice task. The position of the images was counterbalanced, such that for half the participants, the Nutri-Score was displayed on the upper part of the screen and the food image on the lower part, while for the other half of the participants, the positions were reversed. A gray background (#777777) was used for the entire experiment. The experimental tasks, questionnaires, and stimuli used are available on https://osf.io/pef9t/files/.

### Food rating task

Participants were asked to rate all 66 food images on a continuous scale using the mouse to move the slider and mouse button to log their response (*Figure 1a*). The initial position of the slider was in the center of the scale. Food images appeared after a white fixation dot (1000 ms) in random order. Overall, participants rated items on four scales indicating perceived tastiness ('How tasty would you rate this item? Not tasty at all – very tasty'), healthiness ('How healthy would you rate this item? Not healthy at all – very healthy'), wanting ('How much would you like to eat this item at the end of

the experiment? Not at all – very much'), and perceived caloric content ('How high would you rate the caloric content of this item? Very low– very high'). Text and slider were white. No time limit was imposed in this task.

### Food choice task
In the binary food choice task, participants were asked to select the food image they preferred, knowing that they would be incentivized in line with their choices (see *Incentivization*). Overall, participants made 190 choices per session, including a self-paced break halfway through the task. During the task participants' eye movements were recorded (see *Eye-tracking data*). One trial consisted of a white fixation dot (i.e. participants had to fixate the dot for 1000 ms before the trial began, which ensured calibration at each trial), the option screen (self-paced), and a feedback screen (500 ms). The option screen included two food images and their corresponding Nutri-Scores, in counterbalanced positioning. As for the feedback, a black frame was implemented around the chosen option (*Figure 1b*). This part of the experiment took approximately 25 min.

### Visual analogue scale (VAS)
A VAS (*Sepple and Read, 1989*) was used to assess subjective feeling of hunger and fullness (i.e. 'how hungry/full are you?') on a continuous scale ranging from '0=not hungry/full at all to 100=very hungry/full' (*Parker et al., 2004*).

### Other control measures
Demographic information including gender, age, weight, height, handedness, level of education, and monthly disposable income were recorded before the experimental sessions. In both experimental sessions, additional questions concerning participants last meal and usual breakfast routines were collected. If applicable, women also answered questions with respect to their menstrual cycle. Throughout each session, we assessed participants' mood (see *Appendix 1*). At the end of the first session, we also assessed eating behavior (see *Appendix 2*).

### Incentivization
To ensure ecological and external validity (*Barakchian et al., 2021*) during the choice task, participants received a food item for which they indicated a preference of at least 50 in the food rating task and had chosen in a randomly selected trial in the choice task, at the end of each session. We stored the 66 food items in shelves and a fridge in our lab. After each testing session, inventory was assessed, and stores were refilled.

## Eye-tracking data
During the choice tasks, participants' fixation patterns were recorded using a SR Research EyeLink 1000 Plus eye-tracker for high-quality recording of eye movements and pupillometry with up to 2 kHz sampling rate. A chin rest was used to avoid head movements of the participants and subsequent recalibrations. The distance between screen and chin rest was approximately 93 cm. The eye-tracker was calibrated at the beginning of each choice task and after completing half of the trials.

Preprocessing of eye-tracking data was performed in Matlab (2021b, https://www.mathworks.com/) using the edfmex converter (SR Research Ltd.). Preprocessing included parsing the events into trials and locations. Areas of interest (AOI) were the four positions on the screen, where food images and Nutri-Scores were displayed. We increased these areas by 5% of their original size. Preprocessing resulted in two data frames per participant: one in which the length corresponded to the number of trials and fixation durations and the different AOIs were summed within (for multiple fixations at one location) trials; the length of second data frame corresponded to the total number of fixations of all trials of each participant in each condition.

## Data analysis
### Preprocessing
In line with our preregistration, RTs were preprocessed before further analyses by excluding trials that were >4 SD above the individual mean RT per condition or <250 ms. As we had a different number

of hunger ratings between conditions (participants rated their hunger three times in the hungry and four times in the sated condition; see *Figure 1c*), we evaluated the effectiveness of our hunger state manipulation with a RM-ANOVA on the difference scores in hunger rating (i.e. last timepoint–first timepoint) with condition as a within-subject factor and a paired t-tests to assess differences in hunger ratings at lab arrival. Participants' hunger ratings did not entail extreme outliers, and a Shapiro-Wilk test suggested that hunger ratings were normally distributed. Due to missing data in the VAS and PANAS at timepoint 1 (i.e. upon arrival at the lab) in six participants, the analysis of the hunger state manipulation had a sample size of 64. Reported values include *F*- and *t*-statistics, (Bonferroni-corrected) p-values, and effect sizes based on Cohen's *d*.

## Principal component analysis

Overall, we had six different measures of the presented food stimuli, including subjective ratings of tastiness, wanting, healthiness, and caloric content, as well as objective characteristics such as Nutri-Score and objective total caloric content. These measures were highly correlated (*Appendix 3— figure 2*). To assess whether our preregistered goal to study dietary decisions in terms of contrasting taste vs. health aspects was justified, we performed a PCA on these measures using the R package 'FactoMineR' (*Lê et al., 2008*).

## Generalized linear mixed models

In line with our preregistration, analyses of the food choice task were focused on trials in which one option was rated higher in taste and lower in health compared to the other option (i.e. conflict choices). There were on average 75.68 (SD = 21.96) of these trials per participant. The main analyses comprised two types of generalized linear mixed models (GLMM) using the lme4 package (*Bates et al., 2015*) in R (version: 4.3.1). First, we implemented a mixed-effects logistic regression analysis with tasty vs. healthy choice (*Appendix 4*) as binary outcome with a binomial distribution and a logit link function (see also *Appendix 5* for analysis of wanting vs health and *Appendix 6* for analysis of high caloric versus low caloric choice); second, we implemented a mixed effects regression analysis with RT as dependent variable and a Gamma distribution with an identity link function. In both analyses, models with random intercepts for each participant and random slopes for condition ($AIC_{GLMM2choice}$ = 13017.95, $AIC_{GLMM2RT}$ = 23438.93) outperformed models without random effects ($AIC_{GLMchoice}$ = 13845.4, $AIC_{GLMRT}$ = 24556.97) and those with random intercepts only ($AIC_{GLMM1choice}$ = 13216.59; $AIC_{GLMMRT}$ = 28544.95). In line with our preregistration, we included condition (hungry vs sated) and attention (proportion of dwell time on tasty option) as predictors (*Appendix 4—table 1*). Exploratory models including demographic information as well as scores on participants mood and eating behavior are reported in *Appendix 4—table 1* For the RT model, we used the same predictors as in our choice model with 'choice' (tasty vs healthy) as an additional predictor (*Appendix 4—table 2*). Controlling for the order of testing (i.e. whether participants were first tested in the hungry or the sated session) neither affected choices and RTs, nor the predictive power of the main predictors. Reported values include correlation coefficients, standard errors (SE), z- and p-values.

## Eye-tracking

The eye-tracking analyses were implemented on conflicting trials (i.e. one option was tastier compared to the other option). The analyses included a paired *t*-tests for the difference in relative dwell time on attribute between conditions and a Bayesian within-subject multilevel mediation analysis (*Vuorre and Bolger, 2018*) with choice (tasty vs. healthy) as dependent variable, hunger state as independent variable, and proportion of dwell time on tasty option as mediating variable, using the bmlm package in R (*Vuorre, 2023*). Reported values include *t*-statistics, p-values, and effect sizes based on Cohen's *d* for the *t*-test, as well as correlation coefficients, SEs, and credibility intervals (CI) for the mediation analysis. Convergence for the mediation analysis was assessed via the Gelman-Rubin statistic ('Rhat') (*Gelman and Rubin, 1992*) with a threshold of 1.05 (*Vuorre and Bolger, 2018*).

## Cognitive models

To elucidate the cognitive mechanisms underlying the interaction of attention and decision-making in dietary choice, we preregistered to use the multi-attribute time-dependent drift diffusion model (mtDDM) (*Maier et al., 2020*; *Sullivan and Huettel, 2021*) and extend it with attention-related

parameters for both options (*Krajbich et al., 2010*) and attributes. The core assumption of the mtDDM is that different attributes enter the choice process at different times (e.g. taste information before health information). However, our modeling analyses quickly revealed that there was little to no support for different onset times of the two attributes. In addition, we ran into convergence issues in the parameter recovery with the relative starting time parameter not recovering. Therefore, our modeling analyses focused on models incorporating attentional dynamics, and we refrained from further developing the mtDDM model.

The computational models were fit to choices and RT of all (pre-processed) trials. In case of DDMs that included attentional dynamics, eye-tracking data was used to inform the model (see below). Overall, we tested eight different versions of DDMs, all of them including boundary separation ($\alpha$), non-decision time (nDT), and a drift scaling parameter $d$ as free parameters. Note that attentional DDMs are often estimated with $\alpha$ being fixed and the standard deviation of the drift being a free parameter; here, we followed the convention in the larger DDM community and estimated $\alpha$ while fixing the standard deviation to 1. The definition of the drift rate varied across models, and the (relative) starting point ($\beta$) was either fixed to 0.5 or estimated. For the most basic DDM, the drift rate was determined by multiplying the scaling parameter $d$ with the VD which was given by the taste (*T*) and health (*H*) differences of the two options *i* and *j*, weighted by the free parameters $\omega$ (relative taste weight) and $1 - \omega$ (relative health weight), respectively ($0 \leq \omega \leq 1$). Taste and health values were scaled in line with, such that they would be between one and ten using a generalized distance function (*Berkowitsch et al., 2015*).

$$VD = \omega \left( T_i - T_j \right) + (1 - \omega) \left( H_i - H_j \right) \tag{1}$$

The second model was an attentional DDM (aDDM) (*Krajbich et al., 2010*), which included (next to $\omega$) the relative dwell time on each option and parameter $\theta$, which models a dependency of *VD* on the (dwell) time spent on each option. Specifically, the *VD* in favor of option *i* relative to option *j* depends on the dwell time (f) on the options as follows:

$$VD = f_i(\omega(T_i - \theta T_j) + (1 - \omega)(H_i - \theta H_j)) + f_j(\omega(\theta T_i - T_j) + (1 - \omega)(\theta H_i - H_j)) \tag{2}$$

The third model, the multi-attribute attentional DDM (maaDDM) (*Yang and Krajbich, 2023*) included two attentional parameters to discount the non-looked upon option ($\theta$) and attribute ($\phi$), respectively. Thus, *VD* is defined as follows:

$$VD = f_{i,T}(\omega(T_i - \theta T_j) + (1 - \omega)\phi(H_i - \theta H_j)) + f_{i,H}(\omega\phi(T_i - \theta T_j) + (1 - \omega)(H_i - \theta H_j)) + f_{j,T}(\omega(\theta T_i - T_j) + (1 - \omega)\phi(\theta H_i - H_j)) + f_{j,H}(\omega\phi(\theta T_i - T_j) + (1 - \omega)(\theta H_i - H_j)) \tag{3}$$

Finally, we developed and tested an extension of the maaDDM with two separate $\phi$ parameters for taste ($\phi_T$ and health ($\phi_H$) (maaDDM2 $\phi$). The rationale behind the extension is that in our study (but also other related studies), the attributes representing taste and health differed with respect to image complexity, size, and informational content and consequently might differ with respect to their rate of discounting (*Figure 6*). For the maaDDM2 $\phi$, the *VD* is thus given by:

$$VD = f_{i,T}(\omega(T_i - \theta T_j) + (1 - \omega)\phi_H(H_i - \theta H_j)) + f_{i,H}(\omega\phi_T(T_i - \theta T_j) + (1 - \omega)(H_i - \theta H_j)) + f_{j,T}(\omega(\theta T_i - T_j) + (1 - \omega)\phi_H(\theta H_i - H_j)) + f_{j,H}(\omega\phi_T(\theta T_i - T_j) + (1 - \omega)(\theta H_i - H_j)) \tag{4}$$

For each of these four models, we tested two versions which either allowed the relative starting point parameter ($\beta$) to be free or fixed it to 0.5. Models with fixed $\beta$ consistently provided a more parsimonious account of the data. In addition, we also tested models in which the drift rate was informed by the scaled VD of taste and Nutri-Score (*Appendix 11*) wanting and health, as well as wanting and Nutri-Score (*Appendix 12*). Importantly and in line with our PCA, these models yielded comparable results to those reported in the main text.

## Parameter estimation

Parameter estimation was targeted at testing differences across the two hunger state conditions. Specifically, we estimated a set of 'baseline' parameters for the sated condition as well as the 'change' in each parameter under hunger (i.e. *parameter*<sub>hungry</sub> = *parameter*<sub>sated</sub> + change). Following our previous

work (*Kraemer and Gluth, 2023*), all group-level parameters were drawn from normal distributions N(μ,SD) and half-normal distributions HN(μ,SD) for group mean and group SD, respectively. More specifically, for the 'baseline' parameters, the group mean and SD for $\alpha$ were drawn from N(2,1) and HN(0,3), respectively, the group mean and SD for nDT were drawn from N(–1,1) and HN(0,1), respectively, and the group mean and SD for all remaining parameters were drawn from N(0,0.5) and HN(0,0.5), respectively. For the 'change' parameters, the group mean and SD for $\alpha$ and nDT were drawn from N(0,1) and HN(0,1), respectively, and the group mean and SD for all remaining parameters were drawn from N(0,0.25) and HN(0,0.25), respectively. On the participant-level, all individual parameters were drawn from normal distributions N($\mu_{group}$,$SD_{group}$). Some of these parameter values were then soft-plus transformed (in case of $\alpha$, nDT and $\sigma$) to enforce strictly positive values or phi-transformed (in case of $\beta$ and $\omega$) to enforce values between 0 and 1. In the *Results*, we report transformed parameter values which are easier to interpret, but untransformed values for the effect of hunger to illustrate deviations from 0. Hierarchical Bayesian parameter estimation (*Farrell and Lewandowsky, 2018*) was performed with JAGS, called within R using the R2jags package (*Su and Yajima, 2021*), and accelerated by parallel computing. We used piecewise constant averaging (*Lombardi and Hare, 2021*) to speed up model fit, in particular, of the (ma)aDDMs. For sampling, we used eight chains, with 60,000 iterations, 30,000 burnin samples, and a thinning of 12, resulting in 2500 samples per chain. Convergence was assessed via the Gelman-Rubin statistic ('Rhat') (*Gelman and Rubin, 1992*) with a threshold of 1.05. Model fit was quantified with the Deviance Information Criterion (DIC) (*Spiegelhalter et al., 2002*). For our best-performing models (maaDDM, maaDDMsp, maaDDM2 $\phi$, maaDDM2 $\phi$ sp), we performed posterior predictive checks, by drawing 1000 parameter values from the individual posterior parameter distributions, simulating new data, and checking whether the empirical means fell into the 95% HDI of the simulated choice and RT data (see *Figure 4*, *Figure 4—figure supplement 1* and *Appendix 9—figures 1 and 1*). We implemented parameter recoveries of our best models (*Appendix 10—figures 1–4*).

## Acknowledgements

We would like to thank Soyoung Park for valuable discussions regarding our analyses and the study's implications, and to Stephanie Smith for inspirations on the model development. We thank Sophie Bavard and Chih-Chung Ting for providing useful feedback on our analyses. Finally, we would like to thank Polina Andonova, Anne-Christin Kaufmann, Susen Barth, and Stephanie Lehnert for their support with data collection. Sebastian Gluth was supported by the European Research Council (ERC) under the European Union's Horizon 2020 research and innovation program (Grant Agreement No. 948545).

## Additional information

### Funding

| Funder | Grant reference number | Author |
|---|---|---|
| European Research Council | 10.3030/948545 | Sebastian Gluth |

The funders had no role in study design, data collection and interpretation, or the decision to submit the work for publication.

### Author contributions

Jennifer March, Conceptualization, Data curation, Software, Formal analysis, Validation, Investigation, Visualization, Methodology, Writing - original draft, Project administration, Writing – review and editing; Sebastian Gluth, Conceptualization, Resources, Software, Formal analysis, Supervision, Funding acquisition, Validation, Methodology, Writing – review and editing

### Author ORCIDs

Jennifer March ⓘ https://orcid.org/0009-0002-3133-8170
Sebastian Gluth ⓘ https://orcid.org/0000-0003-2241-5103

### Ethics

The Local Ethics Committee of the Faculty of Psychology and Human Movement Sciences at the University of Hamburg approved the study. Participants gave their informed consent to take part in the experiment and were compensated with course credit or 12,50€ per hour and a food item randomly selected based on their choices during the experiment (see Incentivization).

Reviewer #1 (Public review): https://doi.org/10.7554/eLife.103736.3.sa1
Reviewer #2 (Public review): https://doi.org/10.7554/eLife.103736.3.sa2
Reviewer #3 (Public review): https://doi.org/10.7554/eLife.103736.3.sa3
Author response https://doi.org/10.7554/eLife.103736.3.sa4

## Additional files

### Supplementary files

MDAR checklist

Supplementary file 1. Translated version of the general information sheet handed to participants at the beginning of their first session.

### Data availability

Scripts and data for all behavioral and modeling analyses are available on https://github.com/JenniferMarch/HungryLens/ (copy archived at *March, 2025*).

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

## Appendix 1

### Positive and negative affect scale (PANAS)

Studies have linked mood to dietary decision making (e.g. *Gardner et al., 2014*; *Gibson, 2006*), attention (*Grol et al., 2014*; *Wadlinger and Isaacowitz, 2006*), and hunger state (*Ackermans et al., 2022*; *Hill et al., 1991*; *Horman et al., 2018*). We controlled for mood using the German version (*Breyer and Bluemke, 2016*) of the PANAS (*Watson et al., 1988*), which consists of 10 adjectives assessing positive (e.g. 'active') and negative affect (e.g. 'nervous'), respectively. Items are rated on a five-point Likert scale (1=not at all; 5=extremely). Overall, the instrument demonstrates good reliability (Raykovs p>0.9) and validity (*Breyer and Bluemke, 2016*). For data analyses, mean scores of both subscales were used.

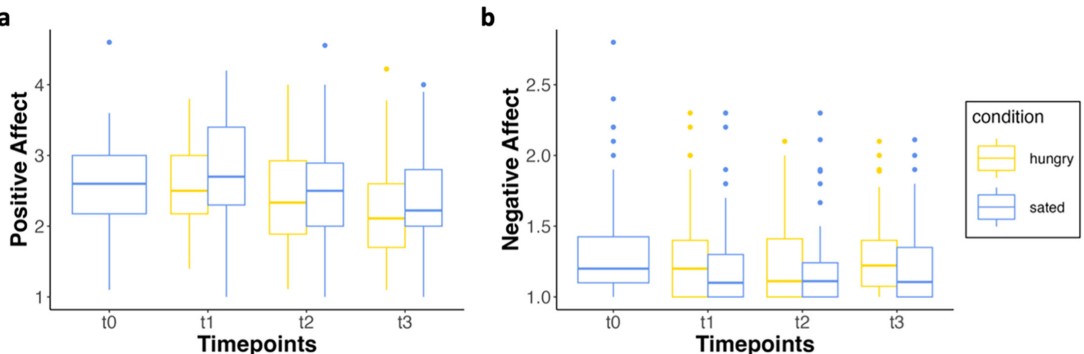

**Appendix 1—figure 1.** Mood across timepoints. (**a**) Average positive affect (PA) scores across t; Welch-corrected RM-ANOVA of PA revealed a main effect of t ($F(1.76, 119.43)=28.179$, p<0.001, partial $\eta^2=0.046$). and condition ($F(1, 68)=5.013$, p=0.028, partial $\eta^2=0.013$): Bonferroni corrected post hoc comparisons demonstrated significant differences in PA between t1 and t3 in the hungry (p=0.007) and the sated condition (p=0.005), all other comparisons did not reach significance. (**b**) Average negative affect (NA) scores across timepoints (t). RM-ANOVA of NA revealed a main effect of condition; Bonferroni-corrected post hoc comparisons were not significant.

# Appendix 2

## Eating behavior

The Fragebogen zum Essverhalten (FEV II) (*Grunert, 1989*) is an abbreviated German version of the 'Dutch Eating Behavior Questionnaire' (*van Strien et al., 1986*) and was used to control eating behavior. The three subscales of the questionnaire describe the basic conceptualizations of human eating behavior, including restraint (e.g. 'I think about my weight when deciding what to eat'), emotionality (e.g. 'When I feel lonely, I want to eat') and externalizing (e.g. 'I eat more than usual when I see others eating.'). Each scale encompasses ten descriptive items that are rated on a five-point Likert scale (1 = 'never' 5 = 'very often'). Overall, all subscales of the questionnaire demonstrate good reliability (restraint: $\alpha$=0.92, emotionality: $\alpha$=0.94, externalizing: $\alpha$=0.89) and construct validity (*Nagl et al., 2016*). For data analyses, mean scores of each subscale were used.

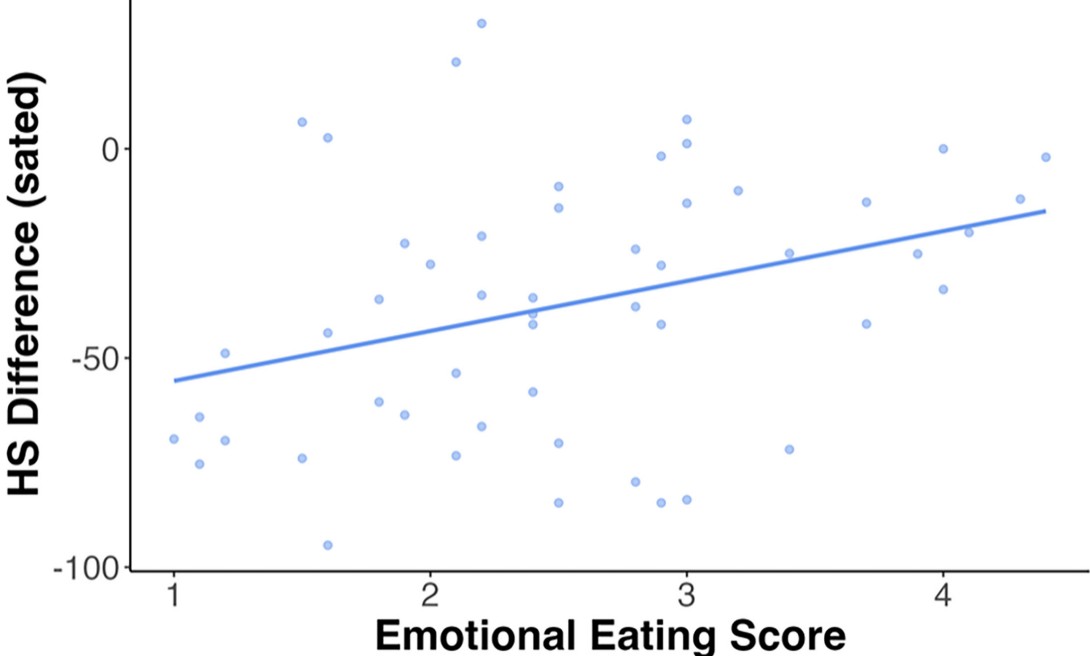

**Appendix 2—figure 1.** Correlations eating behavior and subjective hunger rating. Individuals scoring higher on the subscale of emotional eating, report to be less sated by protein shake. A Pearson's product-moment correlation revealed a moderate positive correlation between emotional eating and the difference in hunger ratings in the sated condition (*r*=0.34, *95% CI*=[0.08, 0.56], p=0.013, n=53). The other scales of the FEV did not yield any significant correlations with difference in hunger state (HS) across conditions and are, therefore, not shown here (see also *Table A2c*)

# Appendix 3

## Correlations in ratings

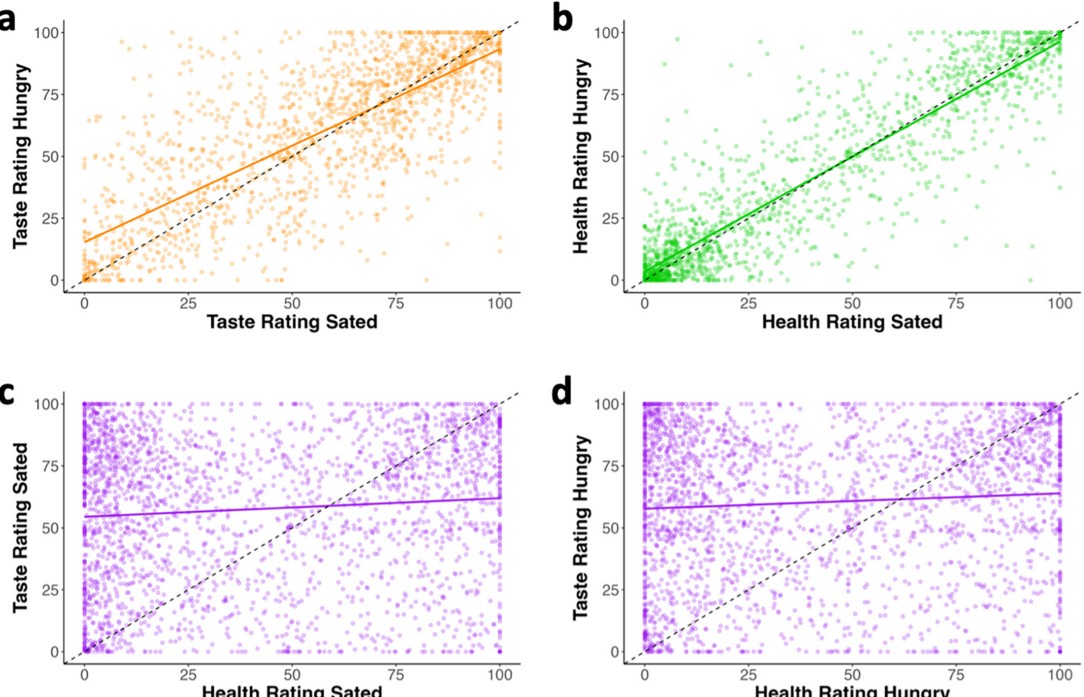

**Appendix 3—figure 1.** Correlation ratings across timepoints. (**a**) Correlation taste across timepoints (t) (*r*=0.778), (**b**) Correlation health across t (*r*=0.916), (**c**) Correlation taste and health in the sated condition (*r*=0.316), (**d**) Correlation taste and health in the hungry condition (*r*=0.301).

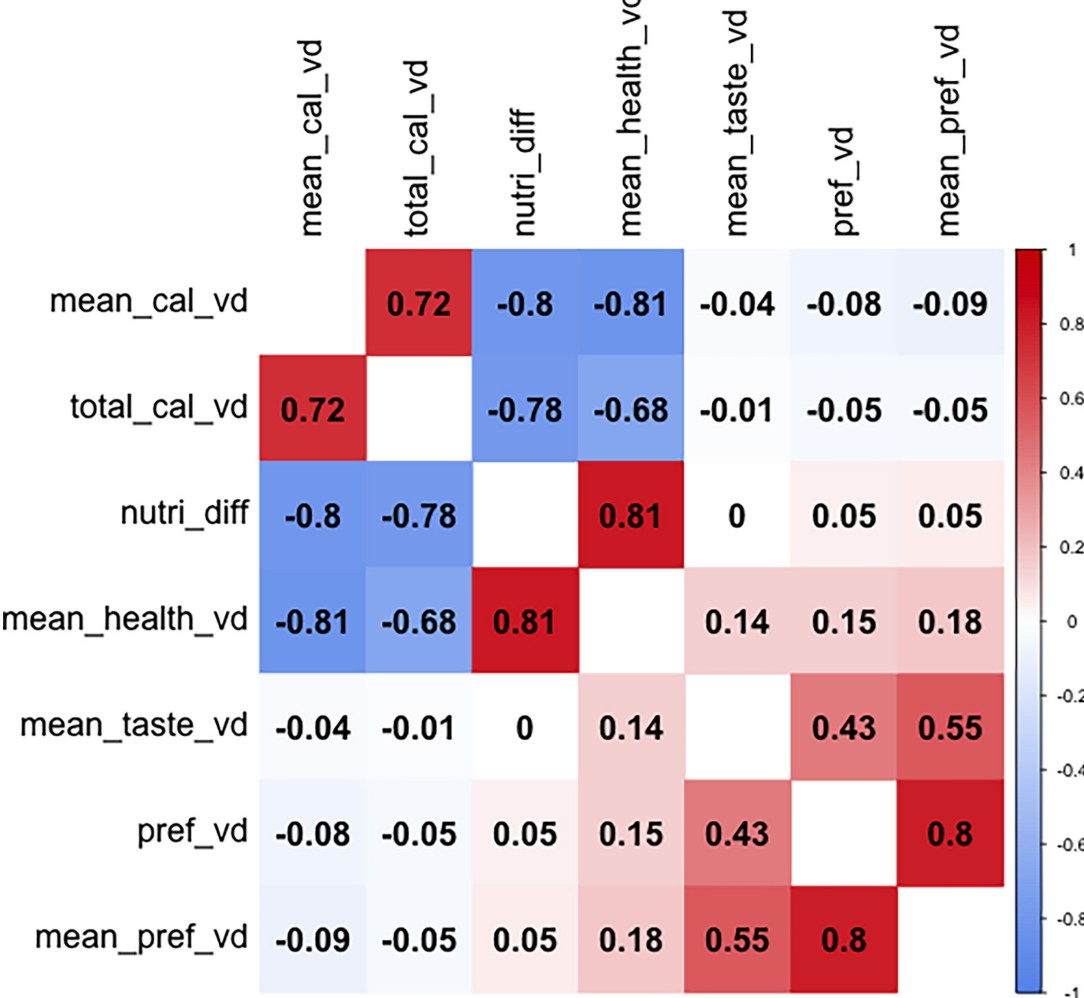

**Appendix 3—figure 2.** Correlations among predictors. vd refers to value difference left – right option.

# Appendix 4

## Specification of general linear models for taste vs health

We build our GLMMs as follows: In the first model, we included only the predictors we had preregistered hypotheses on, namely condition and attention (relative attention to the chosen option) (*Appendix 4—table 1a*). In the second model, we additionally included demographic measures (centralized age and centralized BMI) (*Appendix 4—table 1b*). In the final model, we also included change scores (last timepoint – first timepoint) of positive and negative affect as well as subjective hunger rating, and the three scaled subscales of eating behavior ('restricted,' 'emotional,' and 'externalizing') (*Appendix 4—table 1c*). We rigorously tested best models given the respective predictors as described above, based on AIC and follow-up ANOVAs. In our models, we allowed a maximum of two-way interactions among predictors. For the RT GLMM, we used the same predictors as in the choice GLMM1 and GLMM2 with the addition of choice as predictor (*Appendix 4—table 2*).

**Appendix 4—table 1.** Effect of hunger state on tasty vs healthy choice.

### a) GLMM 1: Results of tasty choice given condition and attention*

| | Fixed effects | | | |
|---|---|---|---|---|
| | Estimate | Std. Error | z value | Pr(>|z|) |
| (Intercept) | 0.832 | 0.102 | 8.164 | <0.001*** |
| conditionsated | –0.211 | 0.103 | –2.05 | 0.04* |
| rel_DT_tasty_option | 0.998 | 0.027 | 36.363 | <0.001*** |

| | Random Effects | | |
|---|---|---|---|
| | Variance | S. D. | Correlation |
| Subject (Intercept) | 0.635 | 0.797 | |
| conditionsated | subject | 0.57 | 0.755 | –0.59 |

### b) GLMM 2: Results of tasty choice given condition, attention, and additional predictors 1[†]

| | Fixed effects | | | |
|---|---|---|---|---|
| | Estimate | Std. Error | z value | Pr(>|z|) |
| (Intercept) | 0.817 | 0.1 | 8.297 | <0.001*** |
| conditionsated | –0.19 | 0.096 | –1.94 | 0.052 |
| rel_DT_tasty_option | 0.999 | 0.027 | 36.331 | <0.001*** |
| rel_dwelldiff_food | 0.093 | 0.028 | 3.267 | 0.001** |
| BMI_cent | 0.015 | 0.019 | 0.774 | 0.439 |
| age_cent | –0.023 | 0.013 | –1.798 | 0.072 |
| conditionsated * age_cent | 0.022 | 0.012 | 1.8 | 0.072 |
| rel_DT_tasty_option * BMI_cent | –0.017 | 0.009 | –1.913 | 0.056 |
| rel_DT_tasty_option * age_cent | –0.011 | 0.004 | –2.645 | 0.008** |

| | Random effects | | |
|---|---|---|---|
| | Variance | S. D. | Correlation |
| Subject (Intercept) | 0.601 | 0.776 | |
| Conditionsated | subject | 0.498 | 0.706 | –0.59 |

### c) GLMM 3: Results of tasty choice given condition, attention and additional predictors 2 [‡]

| | Estimate | Std. Error | z value | Pr(>|z|) |
|---|---|---|---|---|
| (Intercept) | 0.787 | 0.13652 | 5.764 | <0.001*** |

*Continued on next page*

*Continued*

**c) GLMM 3: Results of tasty choice given condition, attention and additional predictors 2 ‡**

|  | Estimate | Std. Error | z value | Pr(>|z|) |
|---|---|---|---|---|
| conditionsated | −0.168 | 0.197 | −0.853 | 0.394 |
| rel_DT_tasty_option | 0.996 | 0.031 | 31.93 | <0.001*** |
| rel_DT_food | 0.101 | 0.031 | 3.248 | 0.001** |
| BMI_cent | 0.009 | 0.02 | 0.47 | 0.638 |
| age_cent | −0.007 | 0.012 | −0.591 | 0.555 |
| PA_change | −0.079 | 0.067 | −1.176 | 0.24 |
| NA_change | −0.055 | 0.084 | −0.657 | 0.511 |
| HS_change | 0.045 | 0.114 | 0.392 | 0.695 |
| external | 0.052 | 0.102 | 0.507 | 0.612 |
| emotional | −0.056 | 0.106 | −0.523 | 0.601 |
| restricted | 0.008 | 0.094 | 0.081 | 0.936 |
| rel_DT_tasty_option * restricted | 0.174 | 0.031 | 5.634 | <0.001*** |

|  | Random effects | | |
|---|---|---|---|
|  | Variance | S. D. | Correlation |
| Subject (Intercep) | 0.194 | 0.443 | |
| Conditionsated | subject | 0.518 | 0.72 | −0.76 |

**d) Model comparison §**

|  | npar | AIC | BIC | logLik | deviance | Chisq | Df | Pr(>|z|) |
|---|---|---|---|---|---|---|---|---|
| GLMM 1 | 6 | 11200 | 11244 | −5594.1 | 11188 | | | |
| GLMM 2 | 12 | 11186 | 11273 | −5580.8 | 11162 | 26.549 | 2 | <0.001*** |
| GLMM 3 | 15 | 8594.5 | 8706.2 | −4281.2 | 8562.4 | | | |

*p-values were calculated using Satterthwaites approximations. Model equation: choice ~ condition + scale(rel_DT_tasty_option) + (1+condition|subject); 'rel_DT_tasty_option' refers to proportion of dwell time on tasty option.

†p-values were calculated using Satterthwaites approximations. Model equation: choice ~ condition + scale(rel_DT_tasty_option) + scale(rel_DT_food)+ BMI_cent + age_cent + condition * age_cent + scale(rel_DT_tasty_option) * BMI_cent+ scale(rel_DT_tasty_option) * age_cent + (1+condition|subject); 'cent' refers to centralized values, 'rel_DT_tasty_option' refers to proportion of dwell time on tasty option, rel_DT_food refers to relative dwell time on food image.

‡p-values were calculated using Satterthwaites approximations. Model equation: choice ~ condition + scale(rel_DT_tasty_option) + scale(rel_DT_food) + BMI_cent + age_cent+ scale(PA_change) + scale(NA_change) + scale(HS_change) + scale(external) + scale(emotional) + scale(restricted) + scale(rel_DT_tasty_option)*scale(restricted) + (1+condition|subject); 'cent' refers to centralized values, 'rel_DT_tasty_option' refers to proportion of dwell time on tasty option, rel_DT_food refers to relative dwell time on food image; change scores (first – last timepoint) of positive affect (PA), negative affect (NA) and hunger state (HS); and the three subscales of FEV

§No direct comparison of model fit between model 3 and models 1 and 2 possible, due to differences in sample size.

**Appendix 4—table 2.** Effect of hunger state on response time.

**a) GLMM RT 1: Response time given condition, choice and attention***

|  | Fixed effects | | |
|---|---|---|---|
|  | Estimate | Std. Error | t value | Pr(>|z|) |
| (Intercept) | 2.748 | 0.096 | 28.644 | <0.001*** |
| conditionsated | 0.032 | 0.099 | 0.327 | 0.744 |
| choice | −0.15 | 0.018 | −8.32 | <0.001*** |

*Appendix 4—table 2 Continued on next page*

*Appendix 4—table 2 Continued*

**a) GLMM RT 1: Response time given condition, choice and attention***

| | Fixed effects | | | |
|---|---|---|---|---|
| | Estimate | Std. Error | t value | Pr(>\|z\|) |
| rel_DT_tasty_option | 0.065 | 0.014 | 4.678 | <0.001*** |
| choice * rel_DT_tasty_option | –0.13 | 0.017 | –7.574 | <0.001*** |

| | Random effects | | |
|---|---|---|---|
| | Variance | S. D. | Correlation |
| Subject (Intercept) | 0.124 | 0.352 | |
| Conditionsated \| subject | 0.16 | 0.4 | –0.51 |
| Residual | 0.122 | 0.349 | |

**b) GLMM RT 2: Response time given condition, choice, attention, and additional predictors†**

| | Fixed effects | | | |
|---|---|---|---|---|
| | Estimate | Std. Error | T value | Pr(>\|z\|) |
| (Intercept) | 2.756 | 0.094 | 29.495 | <0,001*** |
| conditionsated | 0.004 | 0.097 | 0.04 | 0.968 |
| choice | –0.145 | 0.018 | –8.17 | <0,001*** |
| rel_DT_tasty_option | –0.167 | 0.011 | –15.061 | <0,001*** |
| rel_DT_food | 0.066 | 0.014 | 4.865 | <0,001*** |
| BMI_cent | 0.012 | 0.018 | 0.683 | 0.495 |
| age_cent | 0.001 | 0.011 | 0.049 | 0.961 |
| choice * rel_DT_tasty_option | –0.132 | 0.017 | –7.827 | <0,001*** |
| conditionsated * age_cent | 0.012 | 0.011 | 1.076 | 0.282 |
| rel_DT_tasty_option * BMI_cent | –0.004 | 0.004 | –1.044 | 0.296 |
| rel_DT_tasty_option * age_cent | –0.007 | 0.002 | –4.437 | <0,001*** |

| | Random effects | | |
|---|---|---|---|
| | Variance | S. D. | Correlation |
| Subject (Intercept) | 0.117 | 0.342 | |
| Conditionsated \| subject | 0.152 | 0.39 | –0.52 |
| Residual | 0.119 | 0.345 | |

**c) Model comparison**

| | npar | AIC | BIC | logLik | deviance | Chisq | Df | Pr(>\|z\|) |
|---|---|---|---|---|---|---|---|---|
| GLMM RT 1 | 9 | 23304 | 23369 | –11643 | 23286 | | | |
| GLMM RT 2 | 15 | 23028 | 23137 | –11499 | 22998 | 287.55 | 6 | <0.001*** |

*p-values were calculated using Satterthwaites approximations. Model equation: RT ~ condition + choice + scale(rel_DT_tasty_option) + choice * scale(rel_DT_tasty_option) + (1+condition|subject); 'rel_DT_tasty_option' refers to proportion of dwell time on tasty option.

†p-values were calculated using Satterthwaites approximations. Model equation: RT ~ condition + choice + scale(rel_DT_tasty_option) + scale(rel_DT_food) + BMI_cent + age_cent + choice * scale(rel_DT_tasty_option) + condition * age_cent + scale(rel_DT_tasty_option) * BMI_cent + scale(rel_DT_tasty_option) * age_cent + (1+condition|subject); cent' refers to centralized values, 'rel_DT_tasty_option' refers to proportion of dwell time on tasty option, rel_DT_food refers to proportion of dwell time on food image.

# Appendix 5

## Exploratory GLMM (Wanting vs Health)

As for the taste health analyses, we excluded trials in which one option was rated higher on wanting and health, and those that were too fast (<250 ms) and too slow (>4 SD above mean RT). There was on average 85.11 (SD = 14.99) trials per participant. The buildup of the models corresponds to the taste health analyses (SOM3) and models with random intercepts for each participant and random slopes for condition ($AIC_{GLMM2choice}$ = 12946.56), outperformed models without random effects ($AIC_{GLMchoice}$ = 14611) and those with random intercepts only ($AIC_{GLMM1choice}$ = 13229.95). The effects of hunger state on choice (wanting vs health), correspond to those found in the taste vs health analysis (*SOM 3*) and seem to be more robust when accounting for demographic data (*Appendix 5—table 1b*).

**Appendix 5—table 1.** Effect of hunger state on wanted vs healthy choice.

**a) GLMM 1: Results of higher wanted choice given condition and attention***

| | Fixed effects | | | |
| --- | --- | --- | --- | --- |
| | Estimate | Std. Error | z value | Pr(>\|z\|) |
| (Intercept) | 1.245 | 0.136 | 9.181 | <0.001*** |
| conditionsated | −0.325 | 0.113 | −2.864 | 0.004** |
| rel_DT_want_option | 1.012 | 0.039 | 25.958 | <0.001*** |
| conditionsated * rel_DT_want_option | −0.103 | 0.054 | 1.923 | 0.054 |

| | Random effects | | |
| --- | --- | --- | --- |
| | Variance | S. D. | Correlation |
| Subject (Intercept) | 1.178 | 1.085 | |
| Conditionsated \| subject | 0.684 | 0.827 | −0.19 |

**b) GLMM 2: Results of wanting vs health choice given condition, attention and additional predictors 1[†]**

| | Fixed effects | | | |
| --- | --- | --- | --- | --- |
| | Estimate | Std. Error | z value | Pr(>\|z\|) |
| (Intercept) | 1.237 | 0.131 | 9.423 | <0.001*** |
| conditionsated | −0.31 | 0.112 | −2.752 | 0.006** |
| rel_DT_want_option | 1.019 | 0.039 | 26.009 | <0.001*** |
| rel_DT_food | 0.042 | 0.029 | 1.445 | 0.148 |
| BMI_cent | 0.059 | 0.03 | 1.965 | 0.049* |
| age_cent | −0.001 | 0.016 | −0.071 | 0.944 |
| conditionsated * rel_DT_want_option | −0.107 | 0.054 | −1.99 | 0.046* |
| rel_DT_want_option * BMI_cent | −0.013 | 0.007 | −1.768 | 0.077 |
| rel_DT_want_option * age_cent | 0.013 | 0.004 | 3.406 | <0.001*** |
| rel_DT_food * BMI_cent | −0.017 | 0.009 | −1.978 | 0.048* |
| rel_DT_food * age_cent | −0.009 | 0.004 | −2.227 | 0.026* |

| | Random effects | | |
| --- | --- | --- | --- |
| | Variance | S. D. | Correlation |
| Subject (Intercept) | 1.099 | 1.058 | |
| Conditionsated \| subject | 0.671 | 0.819 | −0.18 |

**c) Model comparison**

|        | npar | AIC   | BIC   | logLik  | deviance | Chisq  | Df | Pr(>\|z\|) |
|--------|------|-------|-------|---------|----------|--------|----|-----------|
| GLMM 1 | 7    | 11368 | 11420 | −5677.1 | 11354    |        |    |           |
| GLMM 2 | 14   | 11353 | 11456 | −5662.3 | 11325    | 29.431 | 7  | <0.001*** |

*p-values were calculated using Satterthwaites approximations. Model equation: choice(want) ~ condition + scale(rel_DT_want_option) + condition * (rel_DT_want_option) + (1+condition|subject); 'rel_DT_want_option' refers to proportion of dwell time on the higher wanted option.

†p-values were calculated using Satterthwaites approximations. Model equation: choice ~ condition + scale(rel_DT_want_option) + scale(rel_DT_food) + BMI_cent + age_cent + condition* scale(rel_DT_want_option) + scale(rel_DT_want_option) * BMI_cent + scale(rel_DT_want_option) * age_cent + scale(rel_DT_food) * BMI_cent + scale(rel_DT_food) * age_cent + (1+condition|subject); 'cent' refers to centralized values, 'rel_DT_want_option' refers to proportion of dwell time on the higher wanted option, rel_DT_food refers to proportion of dwell time on food image.

# Appendix 6

## Exploratory GLMM (High caloric vs low caloric)

We only excluded trials that were too fast (<250 ms) and too slow (>4 SD above mean RT). There was on average 185.65 (SD = 6.089) trials per participant. The buildup of the models corresponds to the taste health analyses (*SOM3*) and models with random intercepts for each participant and random slopes for condition $AIC_{GLMM2choice} = 35050.89$, outperformed models without random effects ($AIC_{GLMchoice} = 35830.26$) and those with random intercepts only ($AIC_{GLMM1choice} = 35420.6$). The effects of hunger state on choice (high caloric vs low caloric), correspond to those found in the taste vs health analysis (*SOM3*) and seem to be more robust when accounting for demographic data (*Appendix 6—table 1b*).

**Appendix 6—table 1.** Effect of hunger state on high vs low caloric choice.

a) GLMM 1: Results of high vs low caloric choice given condition and attention*

|  | Fixed effects | | | |
|---|---|---|---|---|
|  | Estimate | Std. Error | z value | Pr(>\|z\|) |
| (Intercept) | −0.0947 | 0.059 | −1.604 | 0.109 |
| conditionsated | −0.261 | 0.054 | −4.86 | <0.001*** |
| rel_DT_hc_option | 1.009 | 0.016 | 61.298 | <0.001*** |
|  | Random effects | | | |
|  | Variance | S. D. | Correlation | |
| Subject (Intercept) | 0.217 | 0.466 | | |
| Conditionsated \| subject | 0.146 | 0.382 | −0.57 | |

b) GLMM 2: Results of high caloric vs low caloric choice given condition, attention and additional predictors 1†

|  | Fixed effects | | | |
|---|---|---|---|---|
|  | Estimate | Std. Error | z value | Pr(>\|z\|) |
| (Intercept) | −0.091 | 0.059 | −1.531 | 0.126 |
| conditionsated | −0.261 | 0.054 | −4.831 | <0.001*** |
| rel_DT_hc_option | 1.017 | 0.017 | 61.154 | <0.001*** |
| rel_DT_food | −0.031 | 0.08 | −1.753 | 0.08 |
| BMI_cent | | 0.012 | −0.674 | 0.501 |
| age_cent | | 0.006 | −0.63 | 0.529 |
| rel_DT_hc_option * rel_DT_food | | 0.018 | −2.382 | 0.017* |
| rel_DT_hc_option * age_cent | | 0.002 | 2.335 | 0.02* |
| rel_DT_food * age_cent | | 0.003 | −3.341 | <0.001*** |
|  | Random effects | | | |
|  | Variance | S. D. | Correlation | |
| Subject (Intercept) | 0.218 | 0.467 | | |
| Conditionsated \| subject | 0.148 | 0.385 | −0.56 | |

c) Model comparison

|  | npar | AIC | BIC | logLik | deviance | Chisq | Df | Pr(>\|z\|) |
|---|---|---|---|---|---|---|---|---|
| GLMM 1 | 6 | 30209 | 30258 | −15099 | 30197 | | | |
| GLMM 2 | 12 | 30195 | 30292 | −15086 | 30171 | 26.383 | 6 | <0.001*** |

*Continued on next page*

*Continued*

**c) Model comparison**

| | npar | AIC | BIC | logLik | deviance | Chisq | Df | Pr(>|z|) |
|---|---|---|---|---|---|---|---|---|

*p-values were calculated using Satterthwaites approximations. Model equation: choice ~ condition + scale(rel_DT_hc_option) + (1+condition|subject); 'rel_DT_hc_option' refers to proportion of dwell time on the higher caloric option.

†p-values were calculated using Satterthwaites approximations. Model equation: choice ~ condition + scale(rel_DT_hc_option) + scale(rel_DT_food) + BMI_cent + age_cent + scale(rel_DT_hc_option) * scale(rel_DT_food) + scale(rel_DT_hc_option) * age_cent + scale(rel_DT_food) * age_cent + (1+condition|subject); 'cent' refers to centralized values, 'rel_DT_hc_option' refers to proportion of dwell time on the higher caloric option., 'rel_DT_food' refers to proportion of dwell time on food image.

# Appendix 7

## Mediation analyses

**Appendix 7—table 1.** Mediation coefficients (wanting).

|     | Mean | SE | Median | 2.50% | 97.50% | $n_{eff}$ | Rhat |
|-----|------|-----|--------|-------|--------|-------|------|
| a   | 0.01 | 0.01 | 0.01 | >0.001 | 0.02 | 11341 | 1 |
| b   | 4.92 | 0.45 | 4.92 | 4.05 | 5.8 | 4860 | 1 |
| cp  | 0.3 | 0.11 | 0.3 | 0.07 | 0.52 | 6213 | 1 |
| me  | 0.07 | 0.04 | 0.07 | >0.001 | 0.15 | 10748 | 1 |
| c   | 0.37 | 0.13 | 0.37 | 0.11 | 0.63 | 6107 | 1 |
| pme | 0.19 | 1.42 | 0.2 | >0.001 | 0.45 | 20097 | 1 |

The data corresponds to the data from the GLMM wanting analyses, that is the trials containing conflicting choices (one option rated higher in wanting, while the other was rated higher in health; see SOM4); a is the effect of hunger state to attention, b is the effect from attention to choice, cp is the indirect effect of hunger state on choice taking attention into account, c is the direct effect of hunger state on choice, when not considering attention, me refers to the mediation effect, thus the combination of paths a and b, pme refers to the proportion of the effect that is mediated. Output refers to posterior mean, standard deviation (=standard error; SE), median, and credible interval, respectively. n_eff refers to the number of effective posterior samples, to obtain confident estimates it is recommended to be >100 **Vuorre and Bolger, 2018**; Rhat is the scale reduction factor, to accurately predict posterior distributions, it should be 1.00, according to **Vuorre and Bolger, 2018** values within .05 are acceptable.

**Appendix 7—table 2.** Standard deviations of subject-level effects (random effects), their covariances, and correlations (wanting).

|        | Mean | SE | Median | 2.50% | 97.50% | n_eff | Rhat |
|--------|------|-----|--------|-------|--------|-------|------|
| tau_a  | 0.04 | 0.01 | 0.04 | 0.03 | 0.05 | 8324 | 1 |
| tau_b  | 3.46 | 0.37 | 3.44 | 2.83 | 4.25 | 6885 | 1 |
| tau_cp | 0.84 | 0.1 | 0.83 | 0.66 | 1.04 | 9260 | 1 |
| covab  | 0.01 | 0.02 | 0.01 | –0.03 | 0.05 | 12091 | 1 |
| corrab | 0.08 | 0.15 | 0.08 | –0.22 | 0.38 | 12408 | 1 |

**Appendix 7—table 3.** Mediation coefficients (calories).

|     | Mean | SE | Median | 2.50% | 97.50% | n_eff | Rhat |
|-----|------|-----|--------|-------|--------|-------|------|
| a   | 0.02 | <0.001 | 0.02 | 0.01 | 0.02 | 8584 | 1 |
| b   | 5.23 | 0.41 | 5.23 | 4.44 | 6.02 | 1885 | 1 |
| cp  | 0.26 | 0.06 | 0.26 | 0.15 | 0.37 | 5982 | 1 |
| me  | 0.1 | 0.03 | 0.1 | 0.05 | 0.15 | 5499 | 1 |
| c   | 0.36 | 0.07 | 0.36 | 0.23 | 0.49 | 5474 | 1 |
| pme | 0.28 | 0.06 | 0.28 | 0.18 | 0.42 | 7635 | 1 |

The data corresponds to the data from the GLMM caloric analyses, that is all trials except for excessively long or short ones (see SOM5); a is the effect of hunger state to attention, b is the effect from attention to choice, cp is the indirect effect of hunger state on choice taking attention into account, c is the direct effect of hunger state on choice, when not considering attention, me refers to the mediation effect, thus the combination of paths a and b, pme refers to the proportion of the effect that is mediated. Output refers to posterior, mean, standard deviation (=standard error; SE), median, and credible interval, respectively. n_eff refers to the number of effective posterior samples, to obtain confident estimates it is recommended to be >100 **Vuorre and Bolger, 2018**; R-hat is the scale reduction factor, to accurately predict posterior distributions, it should be 1.00, according to **Vuorre and Bolger, 2018** values within 0.05 are acceptable.

**Appendix 7—table 4.** Standard deviations of subject-level effects (random effects), their covariances, and correlations (calories).

|        | Mean   | SE      | Median | 2.50%   | 97.50% | n_eff | Rhat |
|--------|--------|---------|--------|---------|--------|-------|------|
| tau_a  | 0.02   | <0.001  | 0.02   | 0.01    | 0.03   | 3544  | 1    |
| tau_b  | 3.31   | 0.31    | 3.29   | 2.76    | 3.96   | 3403  | 1    |
| tau_cp | 0.4    | 0.05    | 0.4    | 0.32    | 0.51   | 8355  | 1    |
| covab  | 0.02   | 0.01    | 0.02   | <0.001  | 0.04   | 6606  | 1    |
| corrab | 0.28   | 0.15    | 0.28   | –0.04   | 0.56   | 8729  | 1    |

## Appendix 8

## Parameter estimates of alternative models

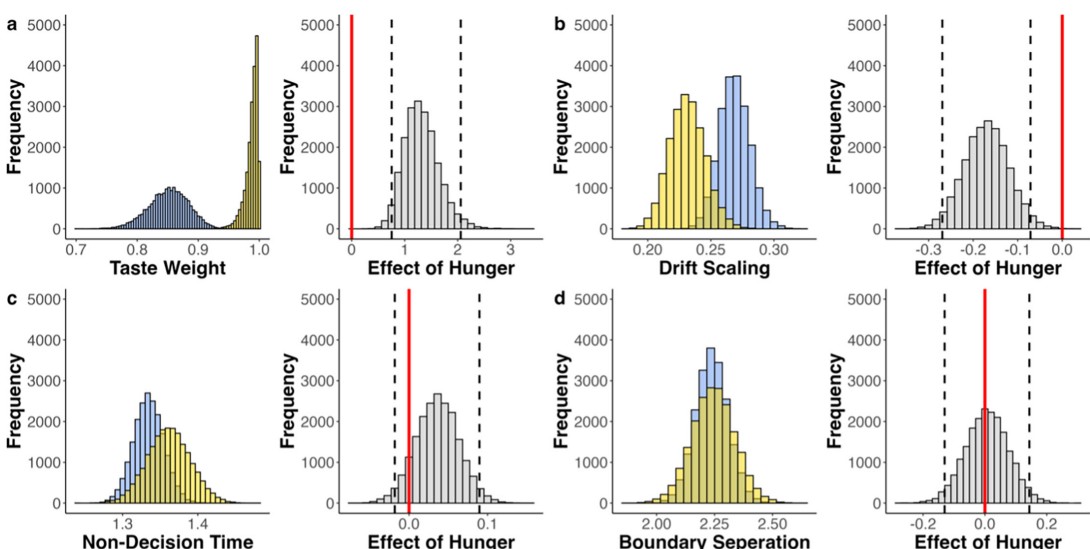

**Appendix 8—figure 1.** Fitted parameters of drift diffusion model (DDM). Fitted parameters across participants (blue = sated, yellow = hungry; left panels) and the effect of hunger state (gray; right panels). Dashed black lines indicate the 95% highest density interval (HDI) edges. (**a**) Estimated relative taste weight across participants. In both conditions the relative taste weight is larger than 0.5, indicating that participants generally weigh taste more than health. There is a positive shift in the distribution of this effect, and the HDI does not include 0, indicating that hungry individuals have a higher relative taste weight (**b**) Estimated drift scaling across participants. There is a negative shift in the distribution of this effect, and the HDI does not include 0, which indicates that hungry individuals accumulate evidence less efficiently. Note, however, that the (better performing) multi-attribute attentional DDM (maaDDM) and maaDDM2*f* indicate that this effect is due to hunger-dependent attentional discounting. (**c, d**) Estimated parameter values for non-decision time (nDT) and boundary separation across participants and the corresponding effects of hunger state.

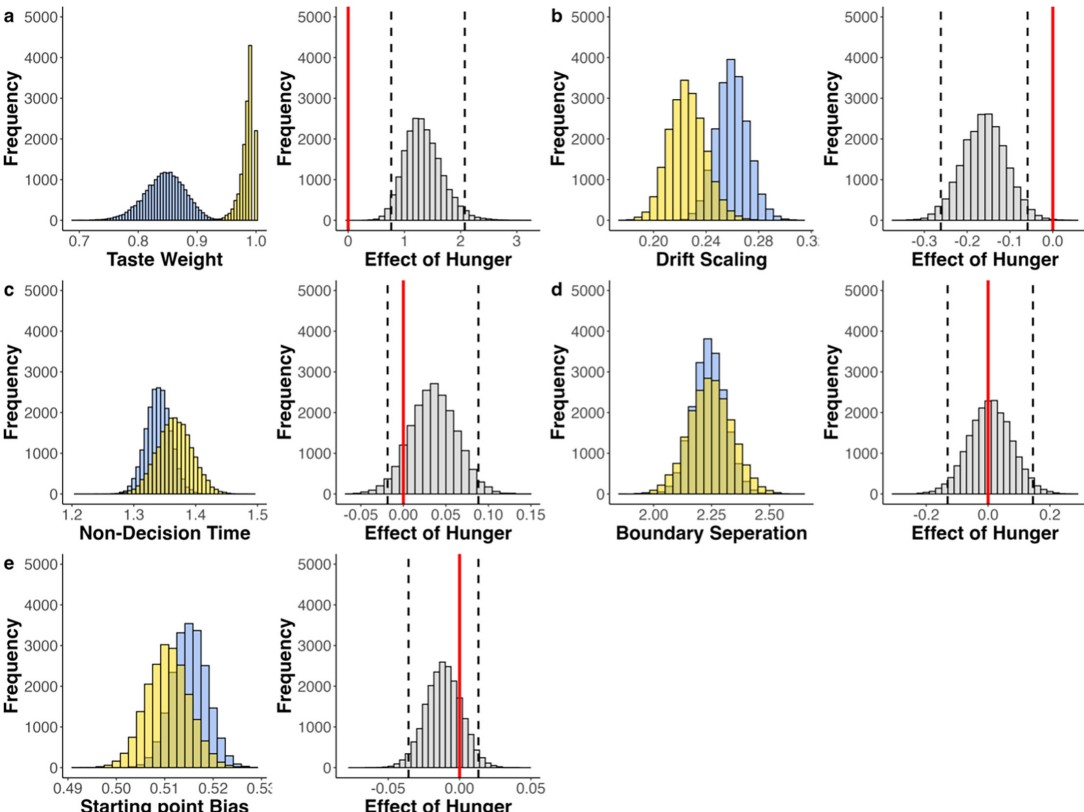

**Appendix 8—figure 2.** Fitted parameters of DDMsp. Fitted parameters across participants (blue = sated, yellow = hungry; left panels) and the effect of hunger state (gray; right panels). Dashed black lines indicate the 95% highest density interval (HDI) edges. (**a**) Estimated relative taste weight across participants. In both conditions, the relative taste weight is larger than 0.5, indicating that participants generally weigh taste more than health. There is a positive shift in the distribution of this effect, and the HDI does not include 0, indicating that hungry individuals have a higher relative taste weight (**b**) Estimated drift scaling across participants. There is a negative shift in the distribution of this effect, and the HDI does not include 0, which indicates that hungry individuals accumulate evidence less efficiently. Note, however, that the (better performing) multi-attribute attentional DDM (maaDDM) and maaDDM2f indicate that this effect is due to hunger-dependent attentional discounting. (**c, d**) Estimated parameter values for non-decision time (nDT) and boundary separation across participants and the corresponding effects of hunger state. (**e**) Estimated parameter values for a relative starting point bias across participants and the corresponding effects of hunger state, indicating that sated individuals are biased towards the taste boundary (sated HDI does not include 0.5), but difference between conditions is not significant.

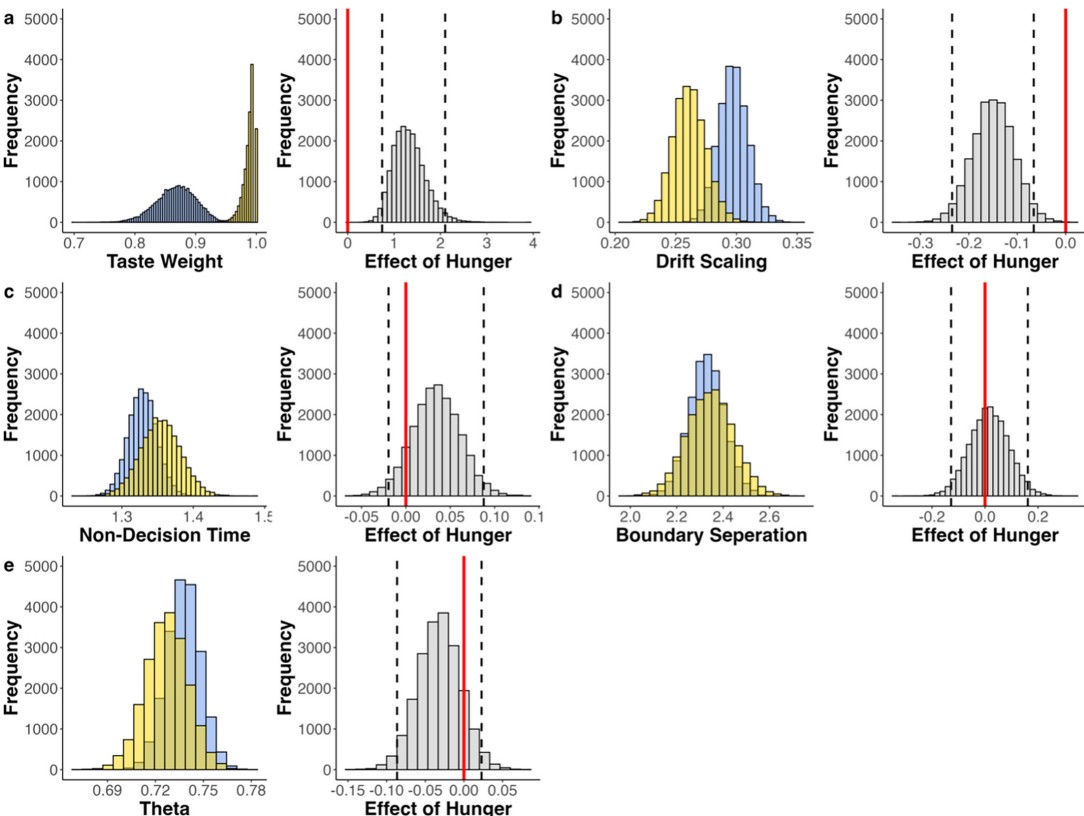

**Appendix 8—figure 3.** Fitted parameters of aDDM. Fitted parameters across participants (blue = sated, yellow = hungry; left panels) and the effect of hunger state (gray; right panels). Dashed black lines indicate the 95% highest density interval (HDI) edges. (**a**) Estimated relative taste weight across participants. In both conditions, the relative taste weight is larger than 0.5, indicating that participants generally weigh taste more than health. There is a positive shift in the distribution of this effect, and the HDI does not include 0, indicating that hungry individuals have a higher relative taste weight (**b**) Estimated drift scaling across participants. There is a negative shift in the distribution of this effect, and the HDI does not include 0, which indicates that hungry individuals accumulate evidence less efficiently. Note, however, that the (better performing) multi-attribute attentional DDM (maaDDM) and maaDDM2*f* indicate that this effect is due to hunger-dependent attentional discounting. (**c-e**) Estimated parameter values for non-decision time (nDT), boundary separation, and theta across participants and the corresponding effects of hunger state.

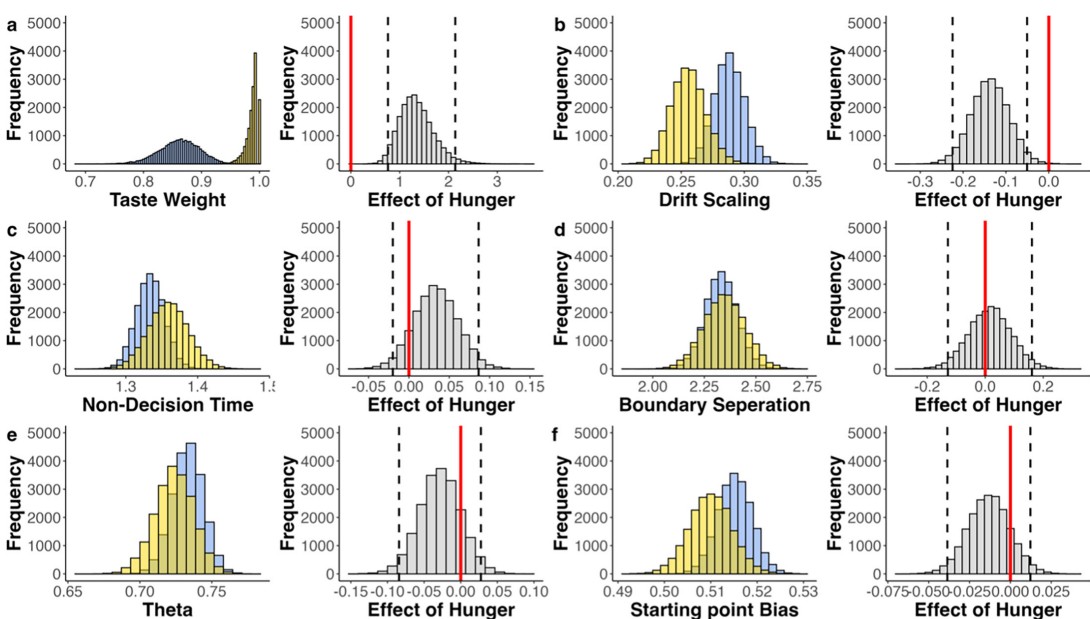

**Appendix 8—figure 4.** Fitted parameters of aDDMsp. Fitted parameters across participants (blue = sated, yellow = hungry; left panels) and the effect of hunger state (gray; right panels). Dashed black lines indicate the 95% highest density interval (HDI) edges. If '0' (red line) is included in HDI, no credible difference between conditions. (**a**) Estimated relative taste weight across participants. In both conditions, the relative taste weight is larger than 0.5, indicating that participants generally weigh taste more than health. There is a positive shift in the distribution of this effect, and the HDI does not include 0, indicating that hungry individuals have a higher relative taste weight (**b**) Estimated drift scaling across participants. There is a negative shift in the distribution of this effect, and the HDI does not include 0, which would indicate that hungry individuals accumulate evidence less efficiently. Note, however, that the (better performing) multi-attribute attentional DDM (maaDDM) and maaDDM2f indicate that this effect is due to hunger-dependent attentional discounting. (**c-e**) Estimated parameter values for non-decision time (nDT), boundary separation, and theta across participants and the corresponding effects of hunger state. (**f**) Estimated parameter values for a relative starting point bias across participants and the corresponding effects of hunger state, indicating that sated individuals are biased towards the taste boundary (sated HDI does not include 0.5), but difference between conditions is not significant.

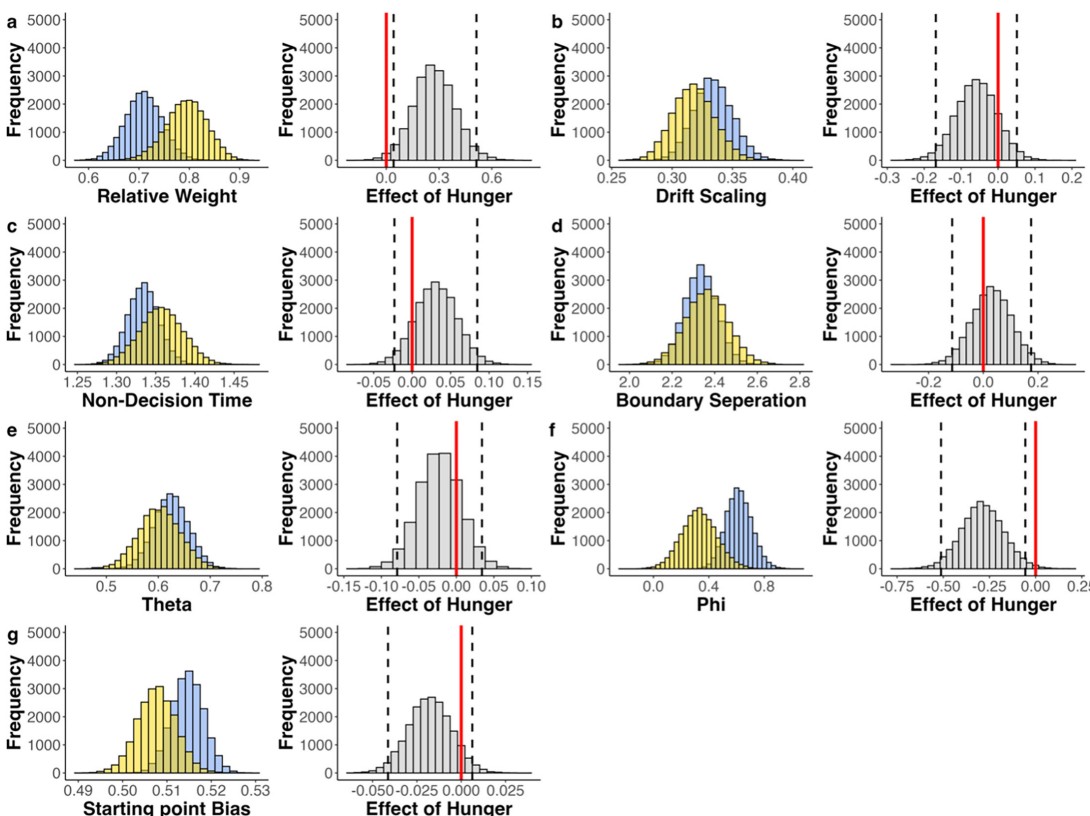

**Appendix 8—figure 5.** Fitted parameters of maaDDMsp. Fitted parameters across participants (blue = sated, yellow = hungry; left panels) and the effect of hunger state (gray; right panels). Dashed black lines indicate the 95% highest density interval (HDI) edges. If '0' (red line) is included in HDI, no credible difference between conditions. (**a**) Estimated relative taste weight across participants. In both conditions, the relative taste weight is larger than 0.5, indicating that participants generally weigh taste more than health. There is a positive shift in the distribution of this effect, and the HDI does not include 0, indicating that hungry individuals have a higher relative taste weight (**b-e**). Estimated parameter values for drift scaling, non-decision time (nDT), boundary separation, and theta across participants and the corresponding effects of hunger state. (**f**) Estimated parameter values for phi across participants. The corresponding effect of hunger indicates that hungry participants discount the non-looked upon attribute more strongly (**g**). Estimated parameter values for a relative starting point bias across participants and the corresponding effects of hunger state indicate that sated individuals are biased towards the taste boundary (sated HDI does not include 0.5), but difference between conditions is not significant.

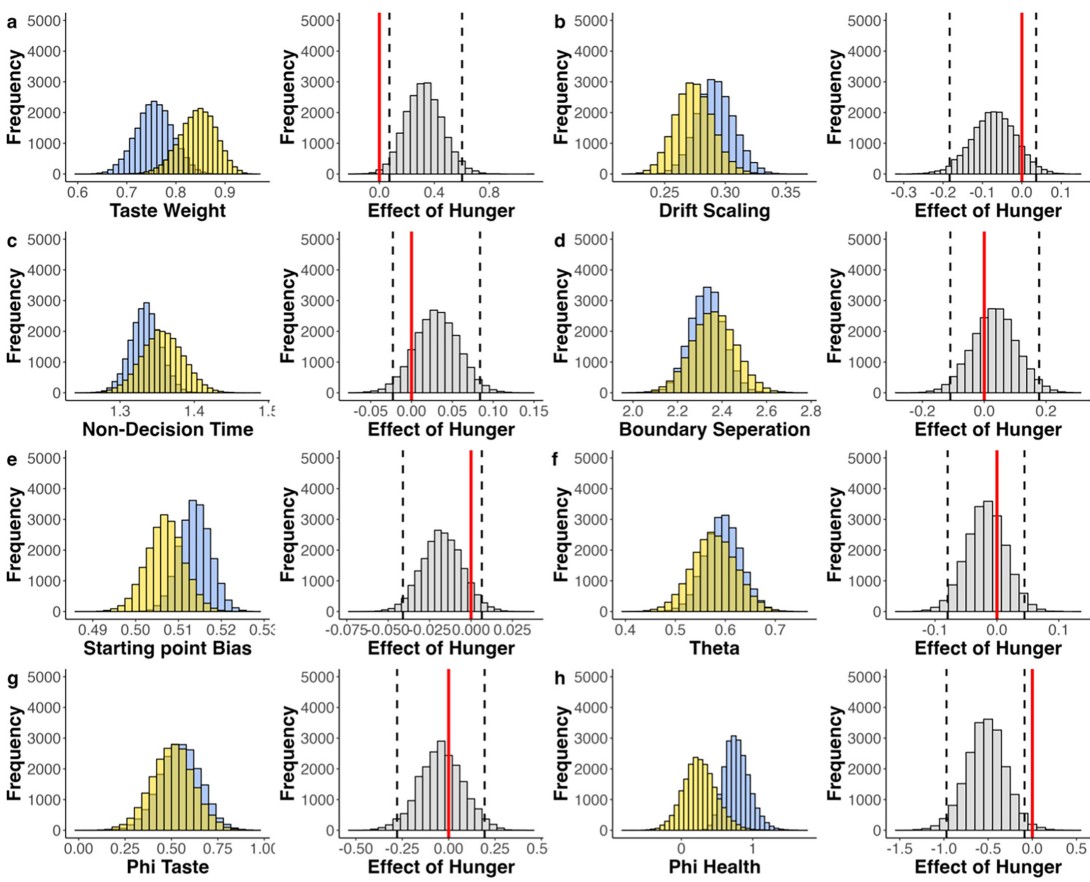

**Appendix 8—figure 6.** Fitted parameters of maaDDM2 $\phi$ sp. Fitted parameters across participants (blue = sated, yellow = hungry; left panels) and the effect of hunger state (gray; right panels). Dashed black lines indicate the 95% highest density interval (HDI) edges. If '0' (red line) is included in HDI, no credible difference between conditions. (**a**) Estimated relative taste weight across participants. In both conditions, the relative taste weight is larger than 0.5, indicating that participants generally weigh taste more than health. There is a marginal positive shift in the distribution of this effect, indicating that hungry individuals have a higher relative taste weight (**b-d**). Estimated parameter values for drift scaling, non-decision time (nDT), boundary and separation across participants and the corresponding effects of hunger state. (**e**) Estimated parameter values for a relative starting point bias across participants and the corresponding effects of hunger state, indicating that sated individuals are biased. towards the taste boundary (sated HDI does not include 0.5), but difference between conditions is not significant. (**f**) Estimated parameter values for theta across participants and the corresponding effects of hunger state. (**g**) Estimated parameter values for $\phi_T$ and the corresponding effects of hunger state. (**h**) Parameter estimates of $\phi_H$ and the corresponding effects of hunger state, showing that the attention-driven discounting of health information was amplified under hunger.

## Appendix 9

### Posterior predictive checks of alternative models

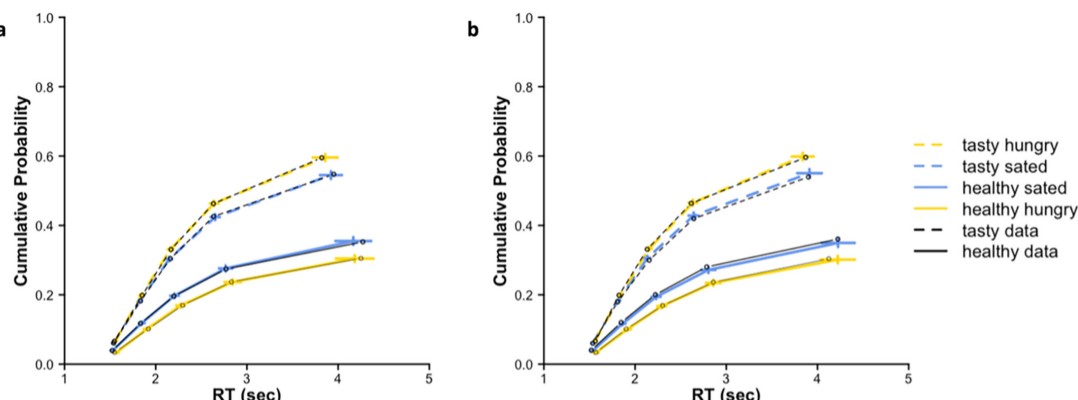

**Appendix 9—figure 1.** Posterior predictive checks maaDDMsp and maaDDM2 $\phi$ sp. Quantile plots of simulated data with fitted parameters of (**a**) the *maaDDMsp* and (**b**) the *maaDDM2 $\phi$ sp* in blue (sated) and yellow (hungry) with highest density intervals (HDIs) of each quantile (vertical lines) and behavior.

## Appendix 10

## Parameter recovery

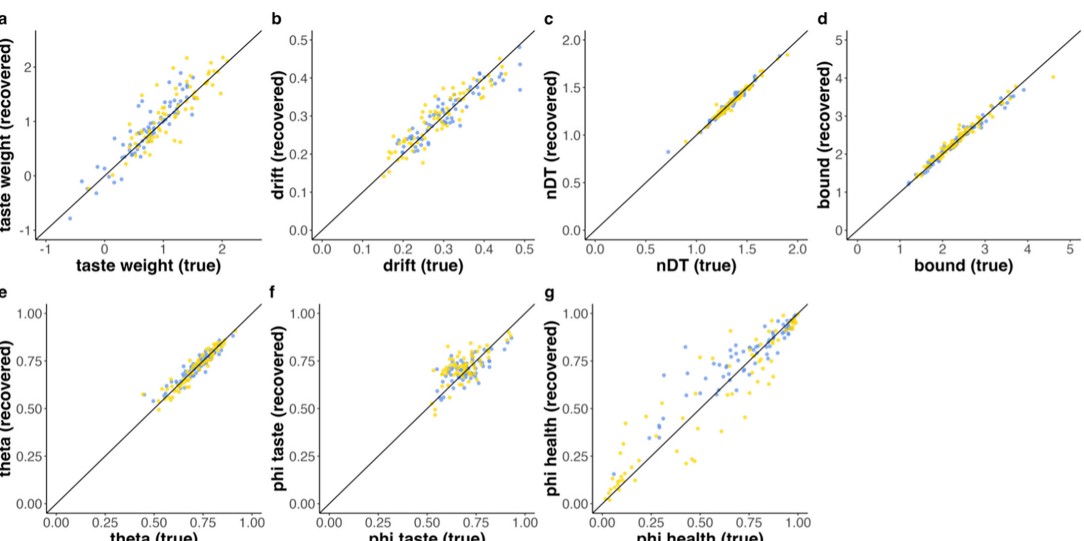

**Appendix 10—figure 1.** Parameter recovery maaDDM2 $\phi$. We generated data based on the means of each parameter and simulated 70 datasets with 180 trials each using empirical subjective value-ratings and gaze patterns. (**a**) The correlation between true and recovered weight parameter was $r$=0.924 in the sated (blue), and $r$=0.913 in the hungry (yellow) condition. (**b**) The correlation between true and recovered drift scaling parameter was $r$=0.921 in the sated (blue), and $r$=0.91 in the hungry (yellow) condition. (**c**) The correlation between true and recovered non-decision time parameter was $r$=0.984 in the sated (blue), and $r$=0.986 in the hungry (yellow) condition. (**d**) The correlation between true and recovered boundary separation parameter was $r$=0.989 in the sated (blue), and $r$=0.982 in the hungry (yellow) condition. (**e**) The correlation between true and recovered theta parameter was $r$=0.937 in the sated (blue), and $r$=0.953 in the hungry (yellow) condition. (**f**) The correlation between true and recovered taste phi parameter was $r$=0.704 in the sated (blue), and $r$=0.712 in the hungry (yellow) condition. (**g**) The correlation between true and recovered health phi parameter was $r$=0.928 in the sated (blue), and $r$=0.94 in the hungry (yellow) condition.

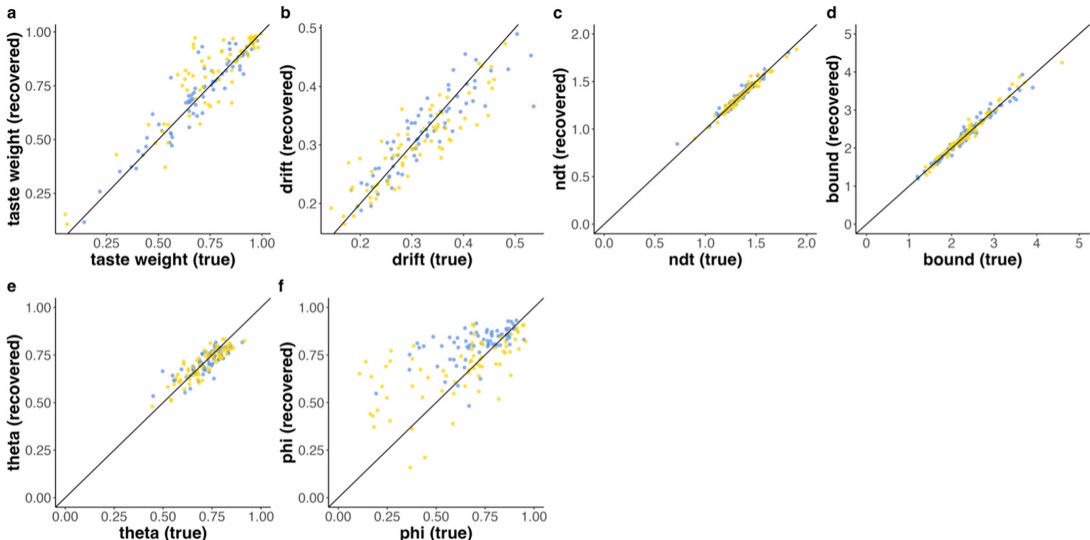

**Appendix 10—figure 2.** Parameter recovery multi-attribute attentional DDM (maaDDM). We generated data based on the means of each parameter and simulated 70 datasets with 180 trials each using empirical subjective value-ratings and gaze patterns. (**a**) The correlation between true and recovered taste weight parameter was
*Appendix 10—figure 2 continued on next page*

*Appendix 10—figure 2 continued*

*r*=0.914 in the sated (blue), and *r*=0.867 in the hungry (yellow) condition. (**b**) The correlation between true and recovered drift scaling parameter was *r*=0.902 in the sated (blue), and *r*=0.875 in the hungry (yellow) condition. (**c**) The correlation between true and recovered non-decision time parameter was *r*=0.976 in the sated (blue), and *r*=0. 985 in the hungry (yellow) condition. (**d**) The correlation between true and recovered boundary separation parameter was *r*=0.983 in the sated (blue), and *r*=0.985 in the hungry (yellow) condition. (**e**) The correlation between true and recovered theta parameter was *r*=0.799 in the sated (blue), and *r*=0.872 in the hungry (yellow) condition. (**f**) The correlation between true and recovered phi parameter was *r*=0.615 in the sated (blue), and *r*=0.618 in the hungry (yellow) condition.

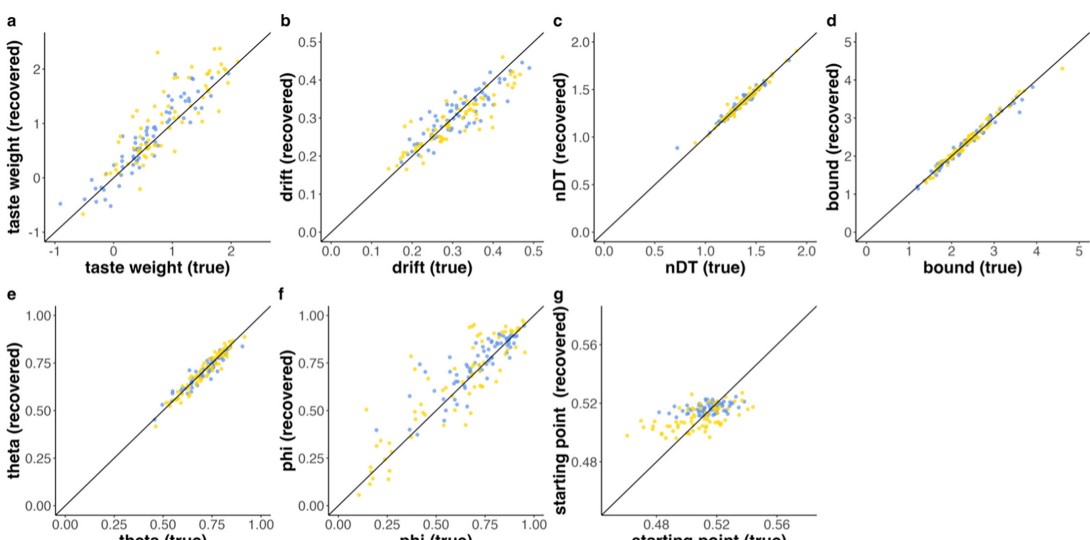

**Appendix 10—figure 3.** Parameter recovery maaDDMsp. We generated data based on the means of each parameter and simulated 70 datasets with 180 trials each using empirical subjective value-ratings and gaze patterns. (**a**) The correlation between true and recovered relative taste weight parameter was *r*=0.936 in the sated (blue), and *r*=0.888 in the hungry (yellow) condition. (**b**) The correlation between true and recovered drift scaling parameter was *r*=0.876 in the sated (blue), and *r*=0.885 in the hungry (yellow) condition. (**c**) The correlation between true and recovered non-decision time parameter was *r*=0.977 in the sated (blue), and *r*=0.984 in the hungry (yellow) condition. (**d**) The correlation between true and recovered boundary separation parameter was *r*=0.988 in the sated (blue), and *r*=0.99 in the hungry (yellow) condition. (**e**) The correlation between true and recovered theta parameter was *r*=0.943 in the sated (blue), and *r*=0.962 in the hungry (yellow) condition. (**f**) The correlation between true and recovered phi parameter was *r*=0.863 in the sated (blue), and *r*=0.864 in the hungry (yellow) condition. (**g**) The correlation between true and recovered starting point bias parameter was *r*=0.417 in the sated (blue), and *r*=0.577 in the hungry (yellow) condition.

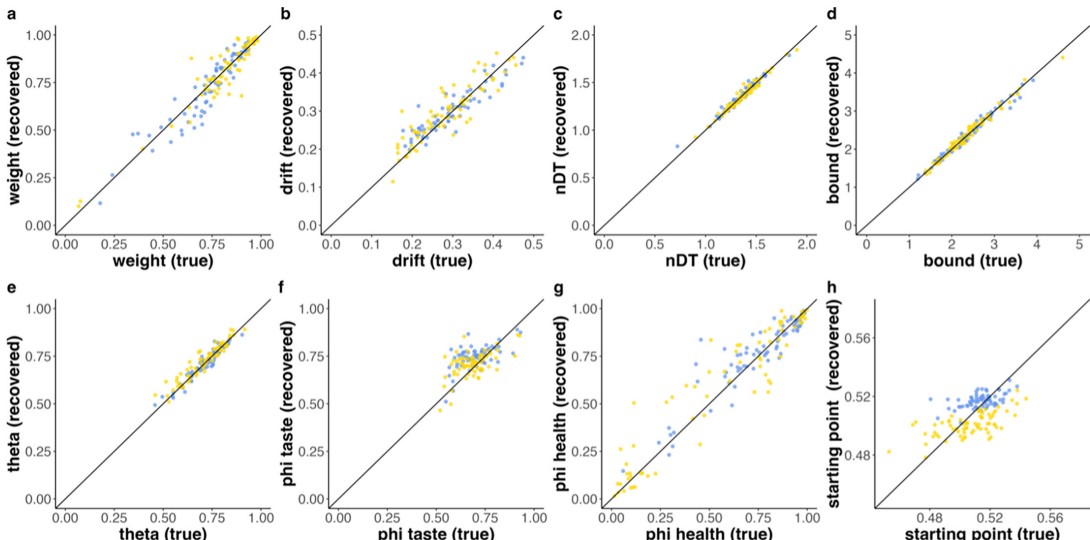

**Appendix 10—figure 4.** Parameter recovery maaDDM2 $\phi$ sp. We generated data based on the means of each parameter and simulated 70 datasets with 180 trials each using empirical subjective value-ratings and gaze patterns. (**a**) The correlation between true and recovered weight parameter was $r=0.938$ in the sated (blue), and $r=0.915$ in the hungry (yellow) condition. (**b**) The correlation between true and recovered drift scaling parameter was $r=0.912$ in the sated (blue), and $r=0.892$ in the hungry (yellow) condition. (**c**) The correlation between true and recovered non-decision time parameter was $r=0.982$ in the sated (blue), and $r=0.984$ in the hungry (yellow) condition. (**d**) The correlation between true and recovered boundary separation parameter was $r=0.99$ in the sated (blue), and $r=0.989$ in the hungry (yellow) condition. (**e**) The correlation between true and recovered theta parameter was $r=0.956$ in the sated (blue), and $r=0.947$ in the hungry (yellow) condition. (**f**) The correlation between true and recovered taste phi parameter was $r=0.594$ in the sated (blue), and $r=0.593$ in the hungry (yellow) condition. (**g**) The correlation between true and recovered health phi parameter was $r=0.921$ in the sated (blue), and $r=0.939$ in the hungry (yellow) condition. (**h**) The correlation between true and recovered starting point bias parameter was $r=0.521$ in the sated (blue), and $r=0.62$ in the hungry (yellow) condition.

## Appendix 11

# Models with Nutri-Score reflecting health value

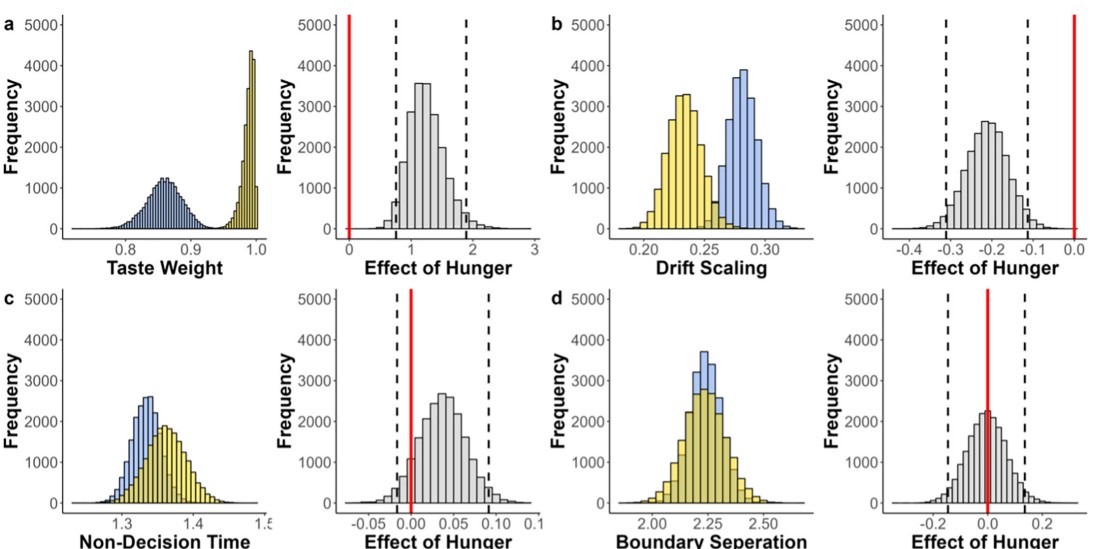

**Appendix 11—figure 1.** Fitted parameters of DDM of models with Nutri-Score reflecting health value. Fitted parameters across participants (blue = sated, yellow = hungry; left panels) and the effect of hunger state (gray; right panels). Dashed black lines indicate the 95% highest density interval (HDI) edges. (**a**) Estimated relative taste weight across participants. In both conditions, the relative taste weight is larger than 0.5, indicating that participants generally weigh taste more than health. There is a positive shift in the distribution of this effect, and the HDI does not include 0, indicating that hungry individuals have a higher relative taste weight (**b**) Estimated drift scaling across participants. There is a negative shift in the distribution of this effect, and the HDI does not include 0, which indicates that hungry individuals accumulate evidence less efficiently. Note, however, that the (better performing) multi-attribute attentional DDM (maaDDM) and maaDDM2*f* indicate that this effect is due to hunger-dependent attentional discounting. (**c, d**) Estimated parameter values for non-decision time (nDT) and boundary separation across participants and the corresponding effects of hunger state.

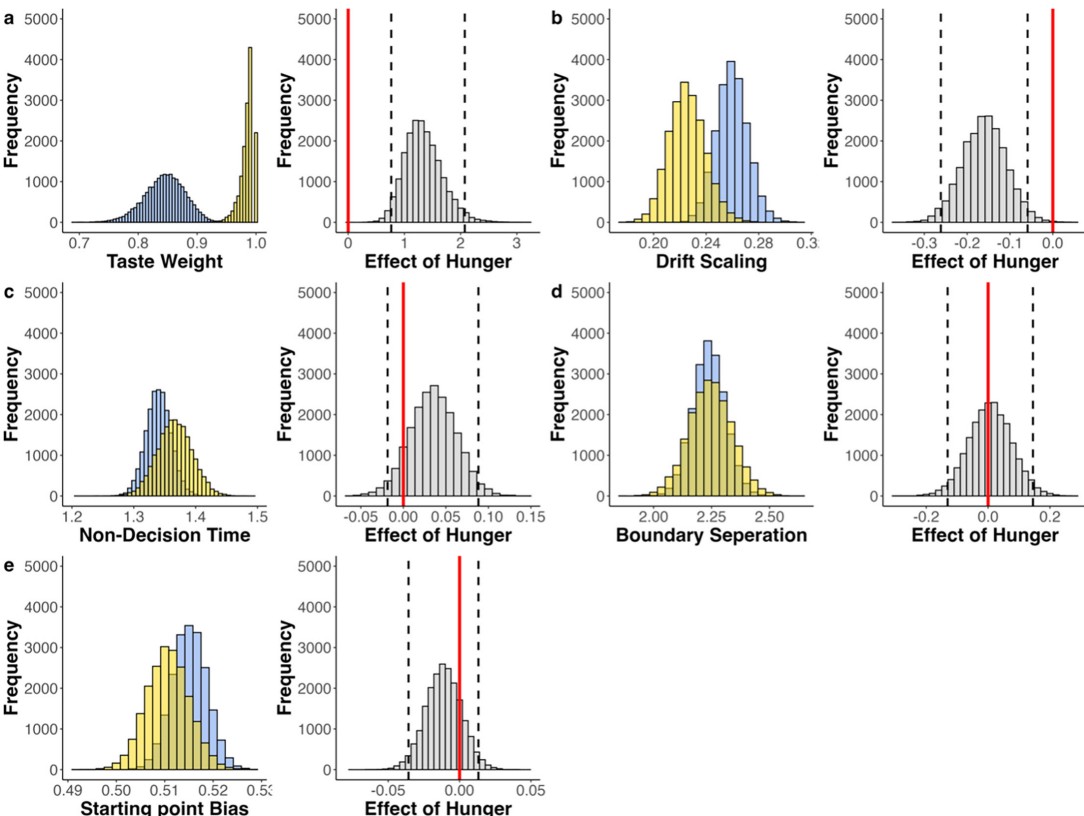

**Appendix 11—figure 2.** Fitted parameters of DDMsp of models with Nutri-Score reflecting health value. Fitted parameters across participants (blue = sated, yellow = hungry; left panels) and the effect of hunger state (gray; right panels). Dashed black lines indicate the 95% highest density interval (HDI) edges. (**a**) Estimated relative taste weight across participants. In both conditions, the relative taste weight is larger than 0.5, indicating that participants generally weigh taste more than health. There is a positive shift in the distribution of this effect, and the HDI does not include 0, indicating that hungry individuals have a higher relative taste weight (**b**) Estimated drift scaling across participants. There is a negative shift in the distribution of this effect, and the HDI does not include 0, which indicates that hungry individuals accumulate evidence less efficiently. Note, however, that the (better performing) multi-attribute attentional DDM (maaDDM) and maaDDM2f indicate that this effect is due to hunger-dependent attentional discounting. (**c, d**) Estimated parameter values for non-decision time (nDT) and boundary separation across participants and the corresponding effects of hunger state. (**e**) Estimated parameter values for a relative starting point bias across participants and the corresponding effects of hunger state, indicating that sated individuals are biased towards the taste boundary (sated HDI does not include 0.5), but difference between conditions is not significant.

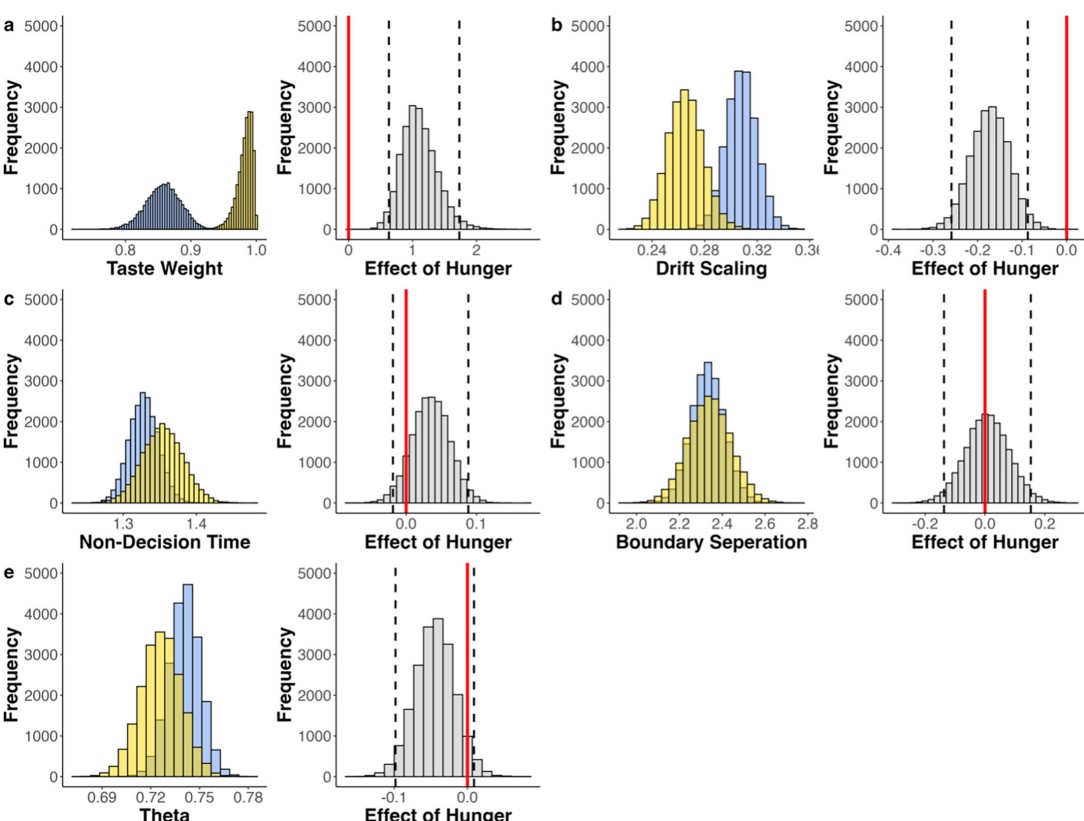

**Appendix 11—figure 3.** Fitted parameters of aDDM of models with Nutri-Score reflecting health value. Fitted Parameters across participants (blue = sated, yellow = hungry; left panels) and the effect of hunger state (gray; right panels). Dashed black lines indicate the 95% highest density interval (HDI edges). (**a**) Estimated relative taste weight across participants. In both conditions, the relative taste weight is larger than 0.5, indicating that participants generally weigh taste more than health. There is a positive shift in the distribution of this effect, and the HDI does not include 0, indicating that hungry individuals have a higher relative taste weight (**b**) Estimated drift scaling across participants. There is a negative shift in the distribution of this effect, and the HDI does not include 0, which indicates that hungry individuals accumulate evidence less efficiently. Note, however, that the (better performing) multi-attribute attentional DDM (maaDDM) and maaDDM2f indicate that this effect is due to hunger-dependent attentional discounting. (**c-e**) Estimated parameter values for non-decision time (nDT), boundary separation, and theta across participants and the corresponding effects of hunger state.

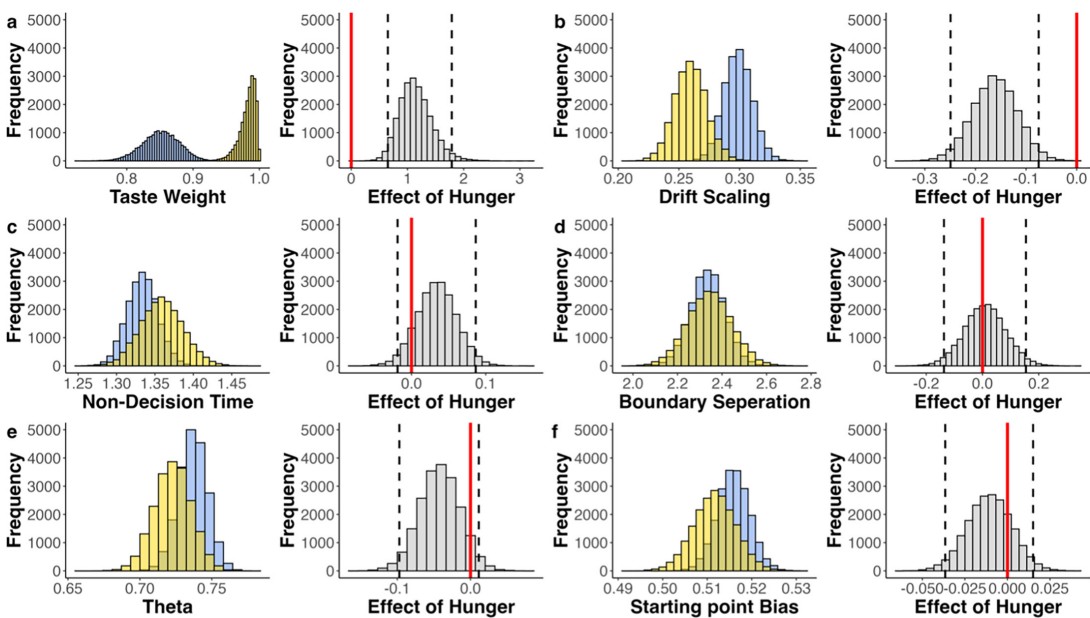

**Appendix 11—figure 4.** Fitted parameters of aDDMsp of models with nutri-score reflecting health value. Fitted parameters across participants (blue = sated, yellow = hungry; left panels) and the effect of hunger state (gray; right panels). Dashed black lines indicate the 95% highest density interval (HDI) edges. If '0' (red line) is included in HDI, no credible difference between conditions. (**a**) Estimated relative taste weight across participants. In both conditions, the relative taste weight is larger than 0.5, indicating that participants generally weigh taste more than health. There is a positive shift in the distribution of this effect, and the HDI does not include 0, indicating that hungry individuals have a higher relative taste weight (**b**) Estimated drift scaling across participants. There is a negative shift in the distribution of this effect, and the HDI does not include 0, which would indicate that hungry individuals accumulate evidence less efficiently. Note, however, that the (better performing) multi-attribute attentional DDM (maaDDM) and maaDDM2*f* indicate that this effect is due to hunger-dependent attentional discounting. (**c-e**) Estimated parameter values for non-decision time (nDT), boundary separation, and theta across participants and the corresponding effects of hunger state. (**f**) Estimated parameter values for a relative starting point bias across participants and the corresponding effects of hunger state, indicating that sated individuals are biased towards the taste boundary (sated HDI does not include 0.5), but difference between conditions is not significant.

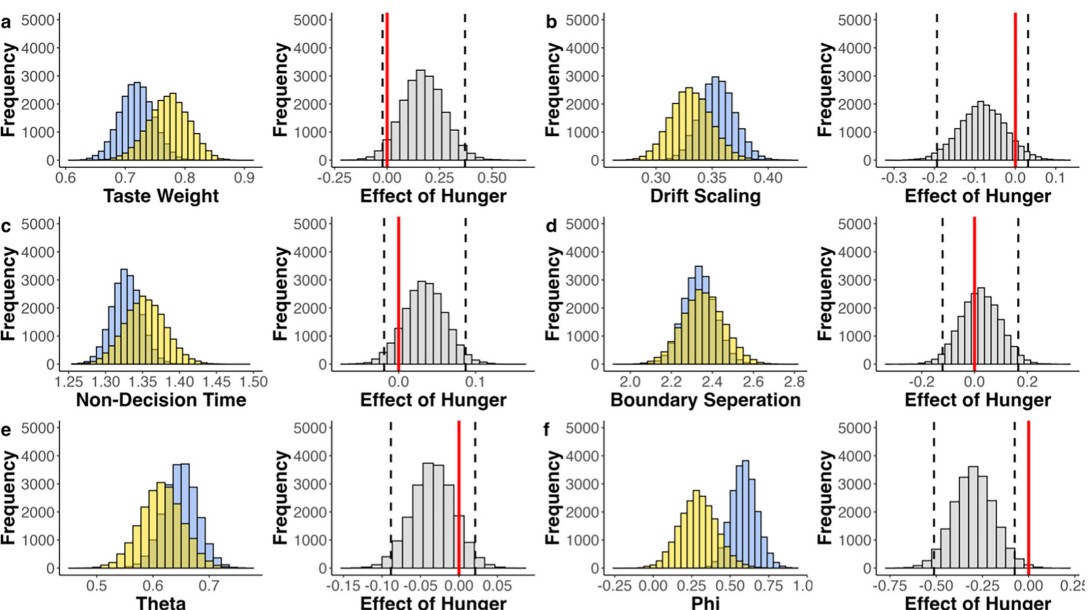

**Appendix 11—figure 5.** Fitted parameters of multi-attribute attentional DDM (maaDDM) of models with nutri-score reflecting health value. Fitted parameters across participants (blue = sated, yellow = hungry; left panels) and the effect of hunger state (gray; right panels). Dashed black lines indicate the 95% highest density interval (HDI) edges. If '0' (red line) is included in HDI, no credible difference between conditions (**a**) Estimated relative taste weight across participants. In both conditions, the relative taste weight is larger than 0.5, indicating that participants generally weigh taste more than health. There is a positive shift in the distribution of this effect, and the HDI does not include 0, indicating that hungry individuals have a higher relative taste weight (**b-e**). Estimated parameter values for drift scaling, non-decision time (nDT), boundary separation, and theta across participants and the corresponding effects of hunger state. (**f**) Estimated parameter values for phi across participants. The corresponding effect of hunger indicates that hungry participants discount the non-looked-upon attribute more strongly.

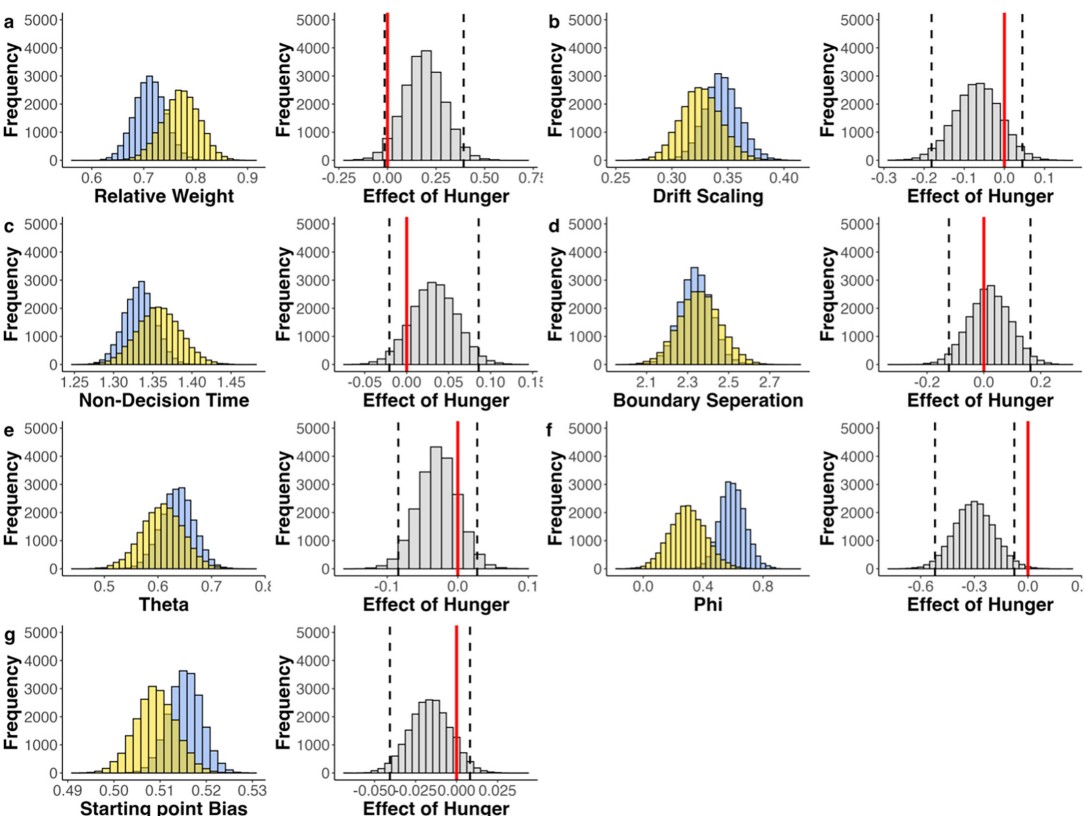

**Appendix 11—figure 6.** Fitted parameters of maaDDMsp of models with nutri-score reflecting health value. Fitted parameters across participants (blue = sated, yellow = hungry; left panels) and the effect of hunger state (gray; right panels). Dashed black lines indicate the 95% highest density interval (HDI) edges. If '0' (red line) is included in HDI, no credible difference between conditions. (**a**) Estimated relative taste weight across participants. In both conditions the relative taste weight is larger than 0.5, indicating that participants generally weigh taste more than health. There is a positive shift in the distribution of this effect, and the HDI does not include 0, indicating that hungry individuals have a higher relative taste weight (**b-e**). Estimated parameter values for drift scaling, non-decision time (nDT), boundary separation, and theta across participants and the corresponding effects of hunger state. (**f**) Estimated parameter values for phi across participants. The corresponding effect of hunger indicates that hungry participants discount the non-looked upon attribute more strongly (**g**). Estimated parameter values for a relative starting point bias across participants and the corresponding effects of hunger state indicate that sated individuals are biased towards the taste boundary (sated HDI does not include 5), but difference between conditions is not significant.

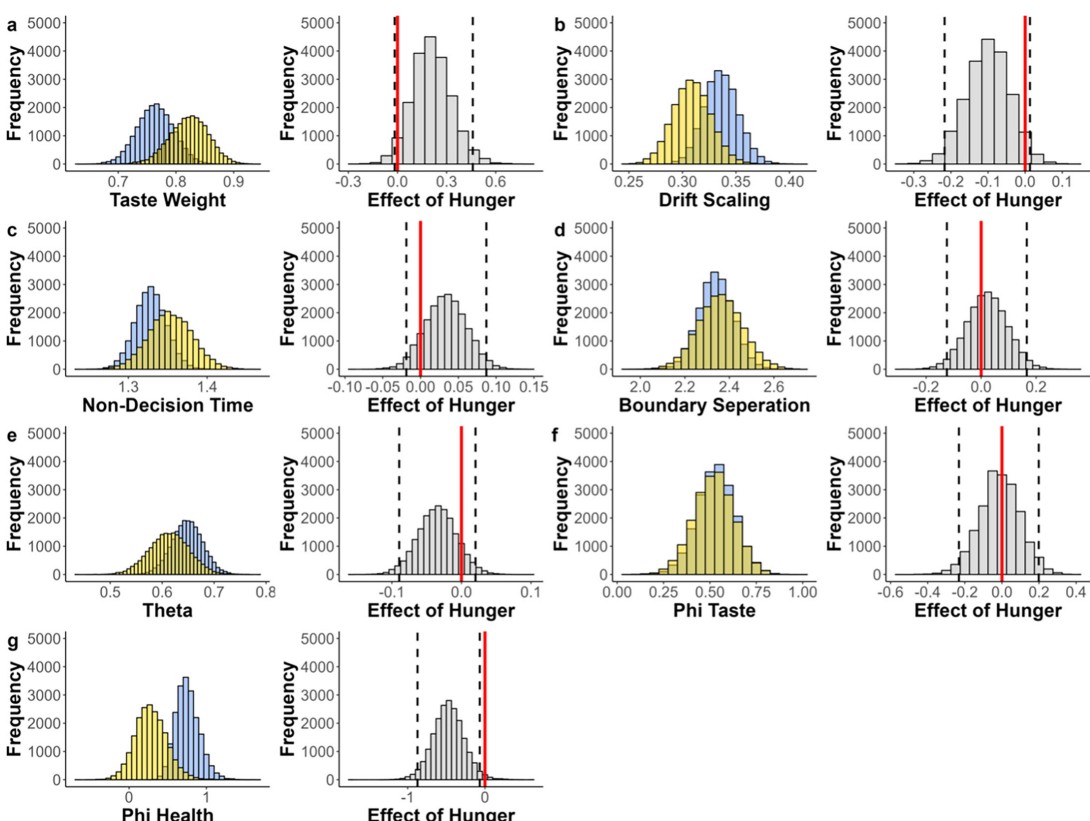

**Appendix 11—figure 7.** Fitted parameters of maaDDM2 $\phi$ of models with Nutri-Score reflecting health value. Fitted parameters across participants (blue = sated, yellow = hungry; left panels) and the effect of hunger state (gray; right panels). Dashed black lines indicate the 95% highest density interval (HDI) edges. If '0' (red line) is included in HDI, no credible difference between conditions. (**a**) Estimated relative taste weight across participants. In both conditions, the relative taste weight is larger than 0.5, indicating that participants generally weigh taste more than Nutri-Score. There is a marginal positive shift in the distribution of this effect, indicating that hungry individuals have a higher relative taste weight (**b–f**). Estimated parameter values for $d$, nDT, $\alpha$, $\theta$, and $\phi_T$ across participants and the corresponding effects of hunger state. (**g**) Parameter estimates of $\phi_H$ and the corresponding effects of hunger state, showing that the attention-driven discounting of health information was amplified under hunger.

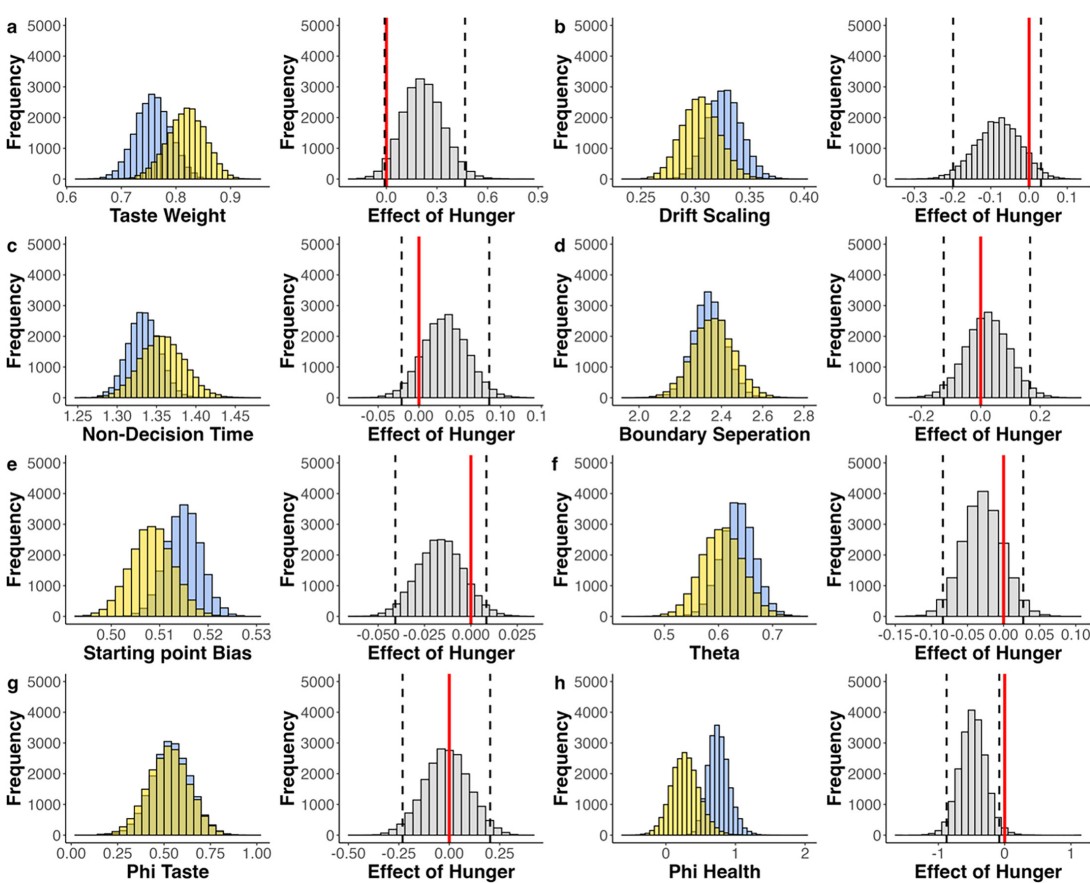

**Appendix 11—figure 8.** Fitted parameters of maaDDM2 $\phi$ sp of Models with Nutri-Score reflecting health value. Fitted parameters across participants (blue = sated, yellow = hungry; left panels) and the effect of hunger state (gray; right panels). Dashed black lines indicate the 95% highest density interval (HDI) edges. If '0' (red line) is included in HDI, no credible difference between conditions. (**a**) Estimated relative taste weight across participants. In both conditions, the relative taste weight is larger than 0.5, indicating that participants generally weigh taste more than Nutri-Score. There is a marginal positive shift in the distribution of this effect, indicating that hungry individuals have a higher relative taste weight (**b-d**). Estimated parameter values for drift scaling, non-decision time (nDT), boundary and separation across participants and the corresponding effects of hunger state. (**e**) Estimated parameter values for a relative starting point bias across participants and the corresponding effects of hunger state, indicating that sated individuals are biased towards the taste boundary (sated HDI does not include 0.5), but difference between conditions is not significant. (**f**) Estimated parameter values for theta across participants and the corresponding effects of hunger state. (**g**) Estimated parameter values for $\phi_T$ and the corresponding effects of hunger state. (**h**) Parameter estimates of $\phi_H$ and the corresponding effects of hunger state, showing that the attention-driven discounting of health information was amplified under hunger.

**Appendix 11—table 1.** Quantitative model comparison of models with nutri-score reflecting health value.

| Model | $\alpha$ | nDT | $d$ | $\omega$ | $\beta$ | $\theta$ | $\phi 1$ | $\phi 2$ | DIC | Rhat |
|---|---|---|---|---|---|---|---|---|---|---|
| DDM | YES | YES | YES | YES | NO | NO | NO | NO | 69784 | 1.004 |
| DDMsp | YES | YES | YES | YES | YES | NO | NO | NO | 69668 | 1.004 |
| aDDM | YES | YES | YES | YES | NO | NO | NO | NO | 65632 | 1.002 |
| aDDMsp | YES | YES | YES | YES | YES | NO | NO | NO | 65637 | 1.003 |
| maaDDM | YES | YES | YES | YES | NO | YES | NO | NO | 65255 | 1.002 |
| maaDDMsp | YES | YES | YES | YES | YES | YES | NO | NO | 65288 | 1.009 |

*Appendix 11—table 1 Continued on next page*

*Appendix 11—table 1 Continued*

| Model | $\alpha$ | nDT | $d$ | $\omega$ | $\beta$ | $\theta$ | $\phi 1$ | $\phi 2$ | DIC | Rhat |
|---|---|---|---|---|---|---|---|---|---|---|
| maaDDM2 $\phi$ | YES | YES | YES | YES | NO | YES | YES | YES | 64122 | 1.016 |
| maaDDM2 $\phi$ sp | YES | YES | YES | YES | YES | YES | YES | YES | 65167 | 1.027 |

The first column states the name of the model; the following nine columns indicate whether the DDM variants included a given parameter or not. The deviance information criterion (DIC) was used as goodness-of-fit measure. The best model (i.e. maaDDM2 $\phi$) is highlighted in bold.

## Appendix 12

## Models with wanting reflecting taste value

Quantitative model comparison of models with wanting reflecting taste

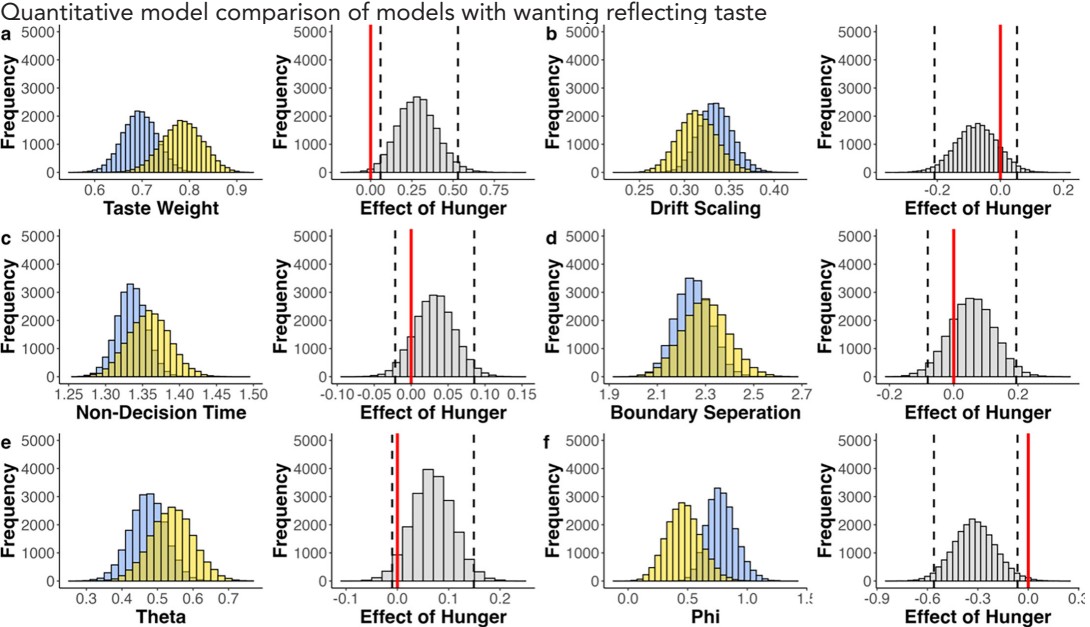

**Appendix 12—figure 1.** Fitted parameters of maaDDM of models with wanting reflecting taste value. Fitted parameters across participants (blue = sated, yellow = hungry; left panels) and the effect of hunger state (gray; right panels). Dashed black lines indicate the 95% highest density interval (HDI) edges. If '0' (red line) is included in HDI, no credible difference between conditions (a) Estimated relative taste weight across participants. In both conditions, the relative taste weight is larger than 0.5, indicating that participants generally weigh taste more than health. There is a positive shift in the distribution of this effect, and the HDI does not include 0, indicating that hungry individuals have a higher relative taste weight (b-e). Estimated parameter values for drift scaling, non-decision time (nDT), boundary separation, and theta across participants and the corresponding effects of hunger state. (f) Estimated parameter values for phi across participants. The corresponding effect of hunger indicates that hungry participants discount the non-looked upon attribute more strongly.

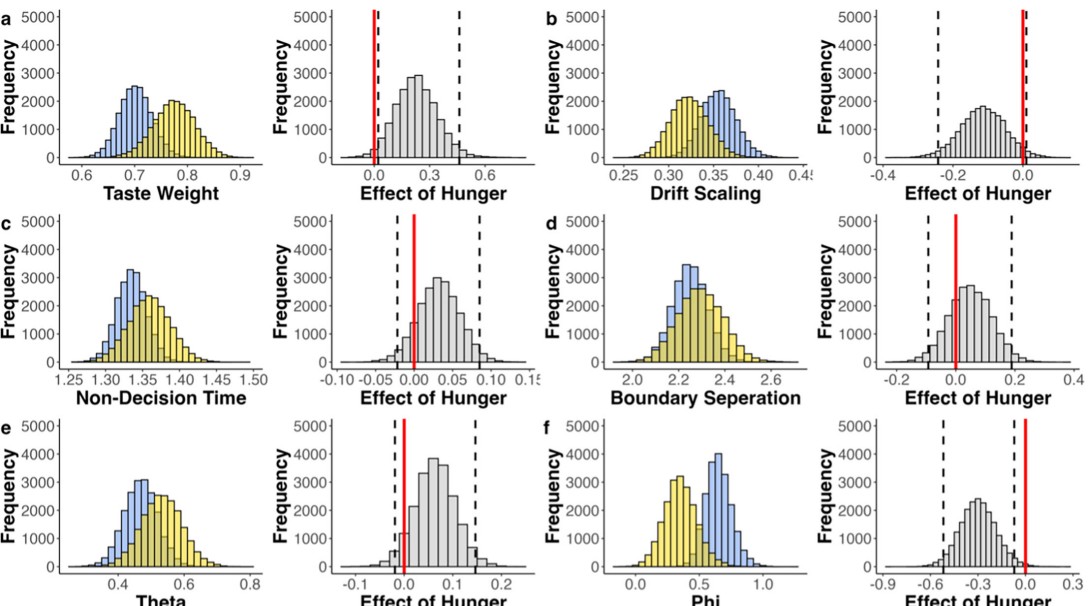

**Appendix 12—figure 2.** Fitted parameters of maaDDM of models with wanting reflecting taste value and Nutri-Score reflecting health value. Fitted parameters across participants (blue = sated, yellow = hungry; left panels) and the effect of hunger state (gray; right panels). Dashed black lines indicate the 95% highest density interval (HDI) edges. If '0' (red line) is included in HDI, no credible difference between conditions (**a**) Estimated relative taste weight across participants. In both conditions, the relative taste weight is larger than 0.5, indicating that participants generally weigh taste more than health. There is a positive shift in the distribution of this effect, and the HDI does not include 0, indicating that hungry individuals have a higher relative taste weight (**b-e**). Estimated parameter values for drift scaling, non-decision time (nDT), boundary separation, and theta across participants and the corresponding effects of hunger state. (**f**) Estimated parameter values for phi across participants. The corresponding effect of hunger indicates that hungry participants discount the non-looked upon attribute more strongly.

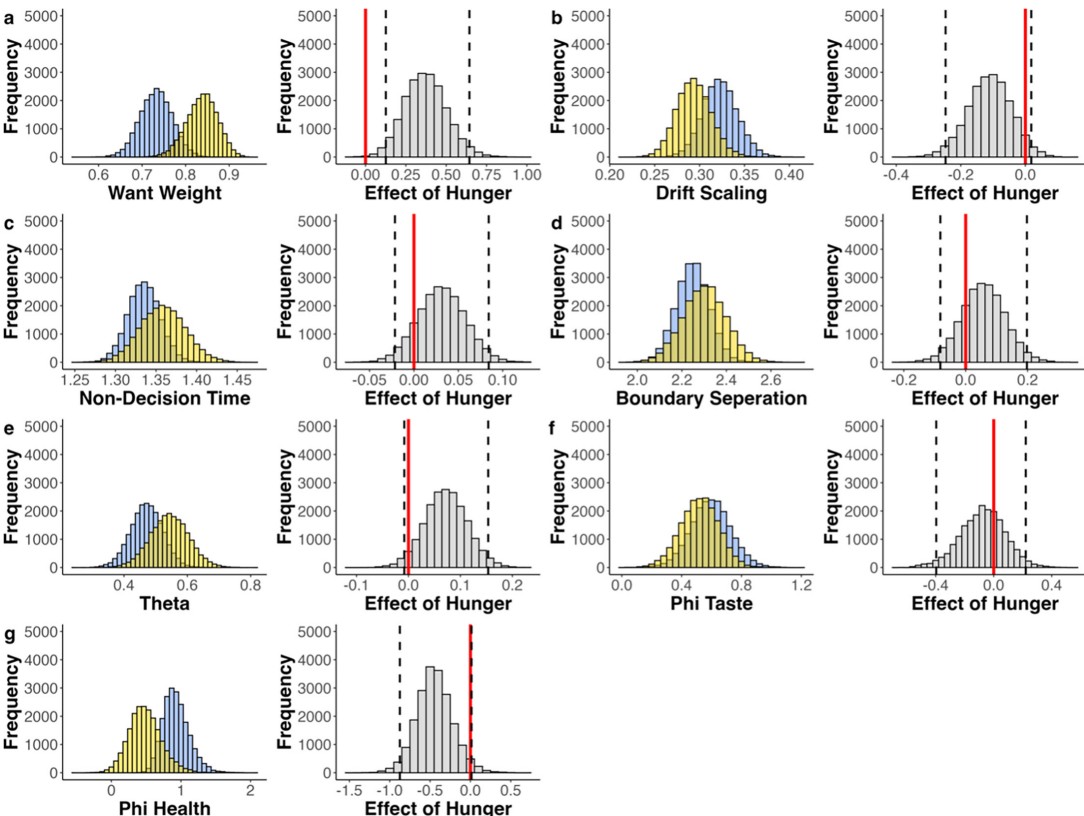

**Appendix 12—figure 3.** Fitted parameters of maaDDM2 $\phi$ of models with wanting reflecting taste value. Fitted parameters across participants (blue = sated, yellow = hungry; left panels) and the effect of hunger state (gray; right panels). Dashed black lines indicate the 95% highest density interval (HDI) edges. If '0' (red line) is included in HDI, no credible difference between conditions. (**a**) Estimated relative taste weight across participants. In both conditions, the relative taste weight is larger than 0.5, indicating that participants generally weigh taste more than health. There is a positive shift in the distribution of this effect, and the HDI does not include 0, indicating that hungry individuals have a higher relative taste weight (**b-f**). Estimated parameter values for $d$, nDT, $\alpha$, $\theta$, and $\phi_T$ n across participants and the corresponding effects of hunger state. (**g**) Parameter estimates of $\phi_H$ and the corresponding effects of hunger state, showing that the attention-driven discounting of health information was amplified under hunger.

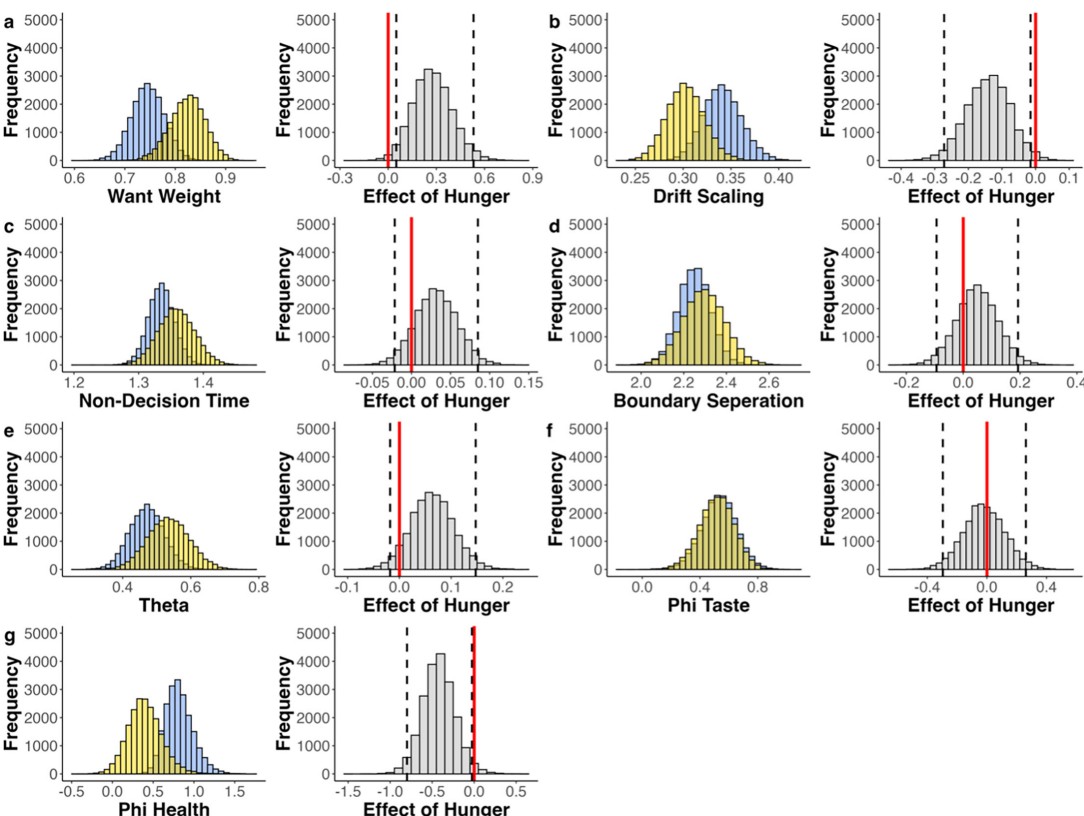

**Appendix 12—figure 4.** Fitted Parameters of maaDDM2 $\phi$ of Models with Wanting Reflecting Taste Value and Nutri-Score Reflecting Health Value. Fitted parameters across participants (blue = sated, yellow = hungry; left panels) and the effect of hunger state (gray; right panels). Dashed black lines indicate the 95% highest density interval (HDI) edges. If '0' (red line) is included in HDI, no credible difference between conditions. (**a**) Estimated relative taste weight across participants. In both conditions, the relative taste weight is larger than 0.5, indicating that participants generally weigh taste more than health. There is a positive shift in the distribution of this effect, and the HDI does not include 0, indicating that hungry individuals have a higher relative taste weight (**b**) Estimated drift scaling across participants. There is a negative shift in the distribution of this effect, and the HDI does not include 0, which indicates that hungry individuals accumulate evidence less efficiently (**c-f**). Estimated parameter values for nDT, $\alpha$, $\theta$, and $\phi_T$ n across participants and the corresponding effects of hunger state. (**g**) Parameter estimates of $\phi_H$ and the corresponding effects of hunger state, showing that the attention-driven discounting of health information was amplified under hunger.

**Appendix 12—table 1.** Quantitative model comparison of models with wanting reflecting taste value.

| Model | $\alpha$ | nDT | $d$ | $\omega$ | $\beta$ | $\theta$ | $\phi_1$ | $\phi_2$ | DIC | Rhat |
|---|---|---|---|---|---|---|---|---|---|---|
| maaDDM1 | YES | YES | YES | YES | NO | YES | YES | NO | 68582 | 2.992 |
| maaDD2 | YES | YES | YES | YES | NO | YES | YES | NO | 68838 | 2.282 |
| maaDDM2 $\phi$1 | YES | YES | YES | YES | NO | YES | YES | YES | 68501 | 1.02 |
| maaDDM2 $\phi$2 | YES | YES | YES | YES | NO | YES | YES | YES | 68785 | 1.039 |

The first column states the name of the model; the following nine columns indicate whether the drift diffusion model (DDM) variants included a given parameter or not. The deviance information criterion (DIC) was used as goodness-of-fit measure. 1 refers to models which used health ratings to reflect health value, 2 reflects models using Nutri-Scores reflecting health ratings.1 Irrespective of how health was defined the quantitively best model in the wanting analyses was the maaDDM2 $\phi$. The high maximum Rhat values for maaDDM1 and maaDDM2 suggest that the modeling results may not be reliable. After closer inspection, we found that the high values are driven by a single participant, suggesting that the model only struggles to capture characteristics in the data of that participant.

