## [Editor Report · eLife Assessment]

This is an **important** study showing that people who are hungry (vs. sated) put more weight on taste (vs. health) in their food choices. The experiment is well-designed and includes choice behavior, eye-tracking, and state-of-the-art computational modeling, resulting in **compelling** evidence supporting the conclusions.

---

## [Referee Report · Reviewer #1 (Public review)]

Summary:

In this article, the authors set out to understand how people's food decisions change when they are hungry vs. sated. To do so, they used an eye-tracking experiment where participants chose between two food options, each presented as a picture of the food plus its "Nutri-Score". In both conditions, participants fasted overnight, but in the sated condition, participants received a protein shake before making their decisions. The authors find that participants in the hungry condition were more likely to choose the tastier option. Using variants of the attentional drift diffusion model, they further find that the best fitting model has different attentional discounts on the taste and health attributes, and that the attentional discount on the health information was larger for the hungry participants.

Strengths:

The article has many strengths. It uses a food-choice paradigm that is established in neuroeconomics. The experiment uses real foods, with accurate nutrition information, and incentivized choices. The experimental manipulation is elegant in its simplicity - administering a high-calorie protein shake. It is also commendable that the study was within-participant. The experiment also includes hunger and mood ratings to confirm the effectiveness of the manipulation. The modeling work is impressive in its rigor - the authors test 8 different variants of the DDM, including recent models like the maaDDM, as well as some completely new variants (maaDDM2phi and 2phisp). The model fits decisively favor the maaDDM2phi.

Weaknesses:

While I do appreciate the within-participant design, it does raise a small concern about potential demand effects. The authors' results would have been more compelling if they had replicated when only analyzing the first session from each participant. However, the authors did demonstrate that there was no effect of order on the results, which helps to alleviate this concern.

---

## [Referee Report · Reviewer #2 (Public review)]

Summary:

This study investigates the effect of fed vs hungry state on food decision making.

70 participants performed a computerized food choice task with eye tracking. Food images came from a validated set with variability in food attributes. Foods ranged from low caloric density unprocessed (fruits) to high caloric density processed foods (chips and cookies).

Prior to the choice task participants rated images for taste, health, wanting, and calories. In the choice task participants simply selected one of two foods. They were told to pick the one they preferred. Screens consisted of two food pictures along with their "Nutri-Score". They were told that one preferred food would be available for consumption at the end.

A drift-diffusion model (DDM) was fit to the reaction time values. Eye tracking was used to measure dwell time on each part of the monitor.

Findings: participants tended to select the item they had rated as "tastier", however, health also contributed to decisions.

Strengths:

The most interesting and innovative aspect of the paper is the use of the DDM models to infer from reaction time and choice the relative weight of the attributes.

Were the ratings re-done at each session? E.g. were all tastiness ratings for the sated session made while sated? This is relevant as one would expect the ratings of tastiness and wanting to be affected by current fed state.

Weaknesses:

My main criticism, which doesn't affect the underlying results, is that the labeling of food choices as being taste- or health-driven is misleading. Participants were not cued to select health vs taste. Studies in which people were cued to select for taste vs health exist (and are cited here). Also, the label "healthy" is misleading, as here it seems to be strongly related to caloric density. A high-calorie food is not intrinsically unhealthy (even if people rate it as such). The suggestion that hunger impairs making healthy decisions is not quite the correct interpretation of the results here (even though everyone knows it to be true). Another interpretation is that hungry people in negative calorie balance simply prefer more calories.

Comments on revisions: No further comments - all my questions addressed.

---

## [Referee Report · Reviewer #3 (Public review)]

Summary:

This well-powered study tested the effects of hunger on value-based dietary decision-making. The main hypothesis was that attentional mechanisms guide choices toward unhealthier and tastier options when participants are hungry, and are in the fasted state compared to satiated states. Participants were tested twice - in a fasted state and in a satiated state after consuming a protein shake. Attentional mechanisms were measured during dietary decision-making by linking food choices and reaction times to eye-tracking data and mathematical drift-diffusion models. The results showed that hunger makes high-conflict food choices more taste-driven and less health-driven. This effect was formally mediated by relative dwell time, which approximates attention drawn to chosen relative to unchosen options. Computational modeling showed that a drift-diffusion model, which assumed that food choices result from a noisy accumulation of evidence from multiple attributes (i.e., taste and health) and discounted non-looked attributes and options, best explained observed choices and reaction times.

Strengths:

This study's findings are valuable for understanding how energy states affect decision-making and provide an answer to how hunger can lead to unhealthy choices. These insights are relevant to psychology, behavioral economics, and behavioral change intervention designs.

The study has a well-powered sample size and hypotheses were pre-registered. The analyses comprised classical linear models and non-linear computational modeling to offer insight into putative cognitive mechanisms.

In summary the study advances the understanding of the links between energy states and value-based decision-making by showing that depleting is powerful for shaping the formation of food preferences. Moreover, the computational analysis part offers a plausible mechanistic explanation at the algorithmic level of observed effects.

Weaknesses:

Some parts of the positioning of the hunger state manipulation and the interpretation of its effects could be improved.

On the positioning side, it does not seem like a 'bad' decision to replenish energy states when hungry by preferring tastier, more often caloric options. In this sense, it is unclear whether the observed behavior in the fasted state is a fallacy or a response to signals from the body. The introduction does mention these two aspects of preferring more caloric food when hungry. However, some ambiguity remains about whether the study results indeed reflect suboptimal choice behavior or a healthy adaptive behavior to restore energy stores.

On the interpretation side, previous work has shown that beliefs about the nourishing and hunger-killing effectiveness of drinks or substances influence subjective and objective markers of hunger, including value-based dietary decision-making, and attentional mechanisms approximated by computational models and the activation of cognitive control regions in the brain. The present study shows differences between the protein shake and a natural history condition (fasted, state). This experimental design, however, cannot rule between alternative interpretations of observed effects. Notably, effects could be due to (a) the drink's active, nourishing ingredients, (b) to consuming a drink versus nothing, or (c) both.

Comments on revisions:

The authors addressed all my comments appropriately and I have no further requests. Thank you for the added discussion of findings and extra analyses.

---

## [Author Response]

The following is the authors’ response to the original reviews.

We thank the editors and the reviewers for their time and constructive comments, which helped us to improve our manuscript “The Hungry Lens: Hunger Shifts Attention and Attribute Weighting in Dietary Choice” substantially. In the following we address the comments in depth:

R1.1: First, in examining some of the model fits in the supplements, e.g. Figures S9, S10, S12, S13, it looks like the "taste weight" parameter is being constrained below 1. Theoretically, I understand why the authors imposed this constraint, but it might be unfairly penalizing these models. In theory, the taste weight could go above 1 if participants had a negative weight on health. This might occur if there is a negative correlation between attractiveness and health and the taste ratings do not completely account for attractiveness. I would recommend eliminating this constraint on the taste weight.

We appreciate the reviewer’s suggestion to test a multi-attribute attentional drift-diffusion model (maaDDM) that does not constrain the taste and health weights to the range of 0 and 1. We tested two versions of such a model. First, we removed the phi-transformation, allowing the weight to take on any value (see Author response image 1). The results closely matched those found in the original model. Partially consistent with the reviewer’s comment, the health weight became slightly negative in some individuals in the hungry condition. However, this model had convergence issues with a maximal Rhat of 4.302. Therefore, we decided to run a second model in which we constrained the weights to be between -1 and 2. Again, we obtained effects that matched the ones found in the original model (see Author response image 2), but again we had convergence issues. These convergence issues could arise from the fact that the models become almost unidentifiable, when both attention parameters (theta and phi) as well as the weight parameters are unconstrained.

**Author response image 1. sa4fig1:** 

**Author response image 2. sa4fig2:** 

R1.2: Second, I'm not sure about the mediation model. Why should hunger change the dwell time on the chosen item? Shouldn't this model instead focus on the dwell time on the tasty option?

We thank the reviewer for spotting this inconsistency. In our GLMMs and the mediation model, we indeed used the proportion of dwell time on the tasty option as predictors and mediator, respectively. The naming and description of this variable was inconsistent in our manuscript and the supplements. We have now rephrased both consistently.

R1.3: Third, while I do appreciate the within-participant design, it does raise a small concern about potential demand effects. I think the authors' results would be more compelling if they replicated when only analyzing the first session from each participant. Along similar lines, it would be useful to know whether there was any effect of order.R3.2: On the interpretation side, previous work has shown that beliefs about the nourishing and hunger-killing effectiveness of drinks or substances influence subjective and objective markers of hunger, including value-based dietary decision-making, and attentional mechanisms approximated by computational models and the activation of cognitive control regions in the brain. The present study shows differences between the protein shake and a natural history condition (fasted, state). This experimental design, however, cannot rule between alternative interpretations of observed effects. Notably, effects could be due to (a) the drink's active, nourishing ingredients, (b) consuming a drink versus nothing, or (c) both. […]R3 Recommendation 1:Therefore, I recommend discussing potential confounds due to expectancy or placebo effects on hunger ratings, dietary decision-making, and attention. […] What were verbatim instructions given to the participants about the protein shake and the fasted, hungry condition? Did participants have full knowledge about the study goals (e.g. testing hunger versus satiation)? Adding the instructions to the supplement is insightful for fully harnessing the experimental design and frame.

Both reviewer 1 and reviewer 3 raise potential demand/ expectancy effects, which we addressed in several ways. First, we have translated and added participants’ instructions to the supplements SOM 6, in which we transparently communicate the two conditions to the participants. Second, we have added a paragraph in the discussion section addressing potential expectancy/demand effects in our design:

“The present results and supplementary analyses clearly support the two-fold effect of hunger state on the cognitive mechanisms underlying choice. However, we acknowledge potential demand effects arising from the within-subject Protein-shake manipulation. A recent study (Khalid et al., 2024) showed that labeling water to decrease or increase hunger affected participants subsequent hunger ratings and food valuations. For instance, participants expecting the water to decrease hunger showed less wanting for food items. DDM modeling suggested that this placebo manipulation affected both drift rate and starting point. The absence of a starting point effect in our data speaks against any prior bias in participants due to any demand effects. Yet, we cannot rule out that such effects affected the decision-making process, for example by increasing the taste weight (and thus the drift rate) in the hungry condition.”

Third, we followed Reviewer 1’s suggestion and tested, whether the order of testing affected the results. We did so by adding “order” to the main choice and response time (RT) GLMM. We neither found an effect of order on choice (*βorder*=-0.001, *SE*=0.163, *p*<.995), nor on RT (*βorder*=0.106, *SE*=0.205, *p*<.603) and the original effects remain stable (see Author response table 1a and Author response table 1 2a below). Further, we used two ANOVAs to compare models with and without the predictor “order”. The ANOVAs indicated that GLMMs without “order” better explained choice and RT (see Author response table 1b and Author response table 2b). Taken together, these results suggest that demand effects played a negligible role in our study.

**Author response table 1. sa4table1:** 

a) GLMM: Results of Tasty vs Healthy Choice Given Condition, Attention and Order
	Fixed Effects
	Estimate	Std. Error	z value	Pr (> |z|)
(Intercept)	0.833	0.127	6.536	<0.001***
conditionsated	–0.211	0.103	–2.05	0.04*
scale(rel_taste_DT)	0.998	0.027	36.363	<0.001***
order	–0.001	0.163	–0.006	0.995
	Random effects
	Variance	S.D.	Correlation
Subject	0.635	0.797	
Conditionsated | subject	0.57	0.755	–0.59
b) Model Comparison
	npar	AIC	BIC	logLik	deviance	Chisq	Df	Pr(>|z|)
GLMM (w/o order)	6	11200	11244	–5594.1	11188			
GLMM (w order)	7	11202	11253	–5594.1	11188	0	1	0.995

Note. p-values were calculated using Satterthwaites approximations. Model equation: *choice* ~ _condition + scale(_rel_taste_DT) *+ order + (1+condition|subject);* rel_taste_DT refers to the relative dwell time on the tasty option; order with hungry/sated as the reference.

**Author response table 2. sa4table2:** 

a) GLMM: Results of Tasty vs Healthy Choice Given Condition, Attention and Order
	Fixed Effects
	Estimate	Std. Error	z value	Pr (> |z|)
(Intercept)	2.698	0.138	19.913	<0.001***
conditionsated	0.03	0.099	0.312	0.755
tasty choice	–1.15	0.018	–8.321	<0.001***
scale(rel_taste_DT)	0.065	0.014	4.679	<0.001***
order	0.106	0.204	0.052	0.603
tasty choice * scale(rel taste_DT)	–0.131	0.017	–7.575	<0.001***
	Random effects
	Variance	S.D.	Correlation
Subject (Intercept)	0.128	0.358	
Conditionsated | subject	0.16	0.4	–0.54
Residual	0.122	0.349	
b) Model Comparison
	npar	AIC	BIC	logLik	deviance	Chisq	Df	Pr(>|z|)
GLMM (w/o order)	9	23304	23369	–11643	23286			
GLMM (w order)	10	23306	23378	–11643	23286	0.27	1	0.603

Note. p-values were calculated using Satterthwaites approximations. Model equation: *RT* ~ *choice + condition + scale(rel_taste_DT) + order + choice * scale(rel_taste_DT) (1+condition|subject);* rel_taste_DT refers to the relative dwell time on the tasty option; order with hungry/sated as the reference.

R1.4: Fourth, the authors report that tasty choices are faster. Is this a systematic effect, or simply due to the fact that tasty options were generally more attractive? To put this in the context of the DDM, was there a constant in the drift rate, and did this constant favor the tasty option?

We thank the reviewer for their observant remark about faster tasty choices and potential links to the drift rate. While our starting point models show that there might be a small starting point bias towards the taste boundary, which would result in faster tasty decisions, we took a closer look at the simulated value differences as obtained in our posterior predictive checks to see if the drift rate was systematically more extreme for tasty choices (Author response image 3). In line with the reviewer’s suggestion that tasty options were generally more attractive, tasty decisions were associated with higher value differences (i.e., further away from 0) and consequently with faster decisions. This indicates that the main reason for faster tasty choices was a higher drift rate in those trials (as a consequence of the combination of attribute weights and attribute values rather than “a constant in the drift rate”), whereas a strong starting point bias played only a minor role.

**Author response image 3. sa4fig3:** 

Note. Value Difference as obtained from Posterior Predictive Checks of the maaDDM2𝜙 in hungry and sated condition for healthy (green) and tasty (orange) choices.

R1.5: Fifth, I wonder about the mtDDM. What are the units on the "starting time" parameters? Seconds? These seem like minuscule effects. Do they align with the eye-tracking data? In other words, which attributes did participants look at first? Was there a correlation between the first fixations and the relative starting times? If not, does that cast doubt on the mtDDM fits? Did the authors do any parameter recovery exercises on the mtDDM?

We thank Reviewer 1 for their observant remarks about the mtDDM. In line with their suggestion, we have performed a parameter recovery which led to a good recovery of all parameters except relative starting time (rst). In addition, we had convergence issues of rst as revealed by parameter Rhats around 20. Together these results indicate potential limitations of the mtDDM when applied to tasks with substantially different visual representations of attributes leading to differences in dwell time for each attribute (see Figure 3b and Figure S6b). We have therefore decided not to report the mtDDM in the main paper, only leaving a remark about convergence and recovery issues.

R2: My main criticism, which doesn't affect the underlying results, is that the labeling of food choices as being taste- or health-driven is misleading. Participants were not cued to select health vs taste. Studies in which people were cued to select for taste vs health exist (and are cited here). Also, the label "healthy" is misleading, as here it seems to be strongly related to caloric density. A high-calorie food is not intrinsically unhealthy (even if people rate it as such). The suggestion that hunger impairs making healthy decisions is not quite the correct interpretation of the results here (even though everyone knows it to be true). Another interpretation is that hungry people in negative calorie balance simply prefer more calories.

First, we agree with the reviewer that it should be tested to what extent participants’ choice behavior can be reduced to contrasting taste vs. health aspects of their dietary decisions (but note that prior to making decisions, they were asked to rate these aspects and thus likely primed to consider them in the choice task). Having this question in mind, we performed several analyses to demonstrate the suitability of framing decisions as contrasting taste vs. health aspects (including the PCA reported in the Supplemental Material).

Second, we agree with the reviewer in that despite a negative correlation (Author response image 4) between caloric density and health, high-caloric items are not intrinsically unhealthy. This may apply only to two stimuli in our study (nuts and dried fruit), which are also by our participants recognized as such.

Finally, Reviewer 2’s alternative explanation, that hungry individuals prefer more calories is tested in SOM5. In line with the reviewer’s interpretation, we show that hungry individuals indeed are more likely to select higher caloric options. This effect is even stronger than the effect of hunger state on tasty vs healthy choice. However, in this paper we were interested in the effect of hunger state on tasty vs healthy decisions, a contrast that is often used in modeling studies (e.g., Barakchian et al., 2021; Maier et al., 2020; Rramani et al., 2020; Sullivan & Huettel, 2021). In sum, we agree with Reviewer 2 in all aspects and have tested and provided evidence for their interpretation, which we do not see to stand in conflict with ours.

**Author response image 4. sa4fig4:** 

Note. strong negative correlation between health ratings and objective caloric content in both hungry (*r*=-.732, *t*(64)=-8.589, *p*<.001) and sated condition (*r*=-.731, *t*(64)=-8.569, *p*<.001).

R3.1: On the positioning side, it does not seem like a 'bad' decision to replenish energy states when hungry by preferring tastier, more often caloric options. In this sense, it is unclear whether the observed behavior in the fasted state is a fallacy or a response to signals from the body. The introduction does mention these two aspects of preferring more caloric food when hungry. However, some ambiguity remains about whether the study results indeed reflect suboptimal choice behavior or a healthy adaptive behavior to restore energy stores.

We thank Reviewer 3 for this remark, which encouraged us to interpret the results also form a slightly different perspective. We agree that choosing tasty over healthy options under hunger may be evolutionarily adaptive. We have now extended a paragraph in our discussion linking the cognitive mechanisms to neurobiological mechanisms:

“From a neurobiological perspective, both homeostatic and hedonic mechanisms drive eating behaviour. While homeostatic mechanisms regulate eating behaviour based on energy needs, hedonic mechanisms operate independent of caloric deficit (Alonso-Alonso et al., 2015; Lowe & Butryn, 2007; Saper et al., 2002). Participants’ preference for tasty high caloric food options in the hungry condition aligns with a drive for energy restoration and could thus be taken as an adaptive response to signals from the body. On the other hand, our data shows that participants preferred less healthy options also in the sated condition. Here, hedonic drivers could predominate indicating potentially maladaptive decision-making that could lead to adverse health outcomes if sustained. Notably, our modeling analyses indicated that participants in the sated condition showed reduced attentional discounting of health information, which poses potential for attention-based intervention strategies to counter hedonic hunger. This has been investigated for example in behavioral (Barakchian et al., 2021; Bucher et al., 2016; Cheung et al., 2017; Sullivan & Huettel, 2021), eye-tracking (Schomaker et al., 2022; Vriens et al., 2020) and neuroimaging studies (Hare et al., 2011; Hutcherson & Tusche, 2022) showing that focusing attention on health aspects increased healthy choice. For example, Hutcherson and Tusche (2022) compellingly demonstrated that the mechanism through which health cues enhance healthy choice is shaped by increased value computations in the dorsolateral prefrontal cortex (dlPFC) when cue and choice are conflicting (i.e., health cue, tasty choice). In the context of hunger, these findings together with our analyses suggest that drawing people’s attention towards health information will promote healthy choice by mitigating the increased attentional discounting of such information in the presence of tempting food stimuli.”

**Recommendations for the authors:**
R1: The Results section needs to start with a brief description of the task. Otherwise, the subsequent text is difficult to understand.

We included a paragraph at the beginning of the results section briefly describing the experimental design.

R1/R2: In Figure 1a it might help the reader to have a translation of the rating scales in the figure legend.

We have implemented an English rating scale in Figure 1a.

R2: Were the ratings redone at each session? E.g. were all tastiness ratings for the sated session made while sated? This is relevant as one would expect the ratings of tastiness and wanting to be affected by the current fed state.

The ratings were done at the respective sessions. As shown in S3a there is a high correlation of taste ratings across conditions. We decided to take the ratings of the respective sessions (rather than mean ratings across sessions) to define choice and taste/health value in the modeling analyses, for several reasons. First, by using mean ratings we might underestimate the impact of particularly high or low ratings that drove choice in the specific session (regression to the mean). Second, for the modeling analysis in particular, we want to model a decision-making process at a particular moment in time. Consequently, the subjective preferences in that moment are more accurate than mean preferences.

R2: It would be helpful to have a diagram of the DDM showing the drifting information to the boundary, and the key parameters of the model (i.e. showing the nDT, drift rate, boundary, and other parameters). (Although it might be tricky to depict all 9 models).

We thank the reviewer for their recommendation and have created Figure 6, which illustrates the decision-making process as depicted by the maaDDM2phi.

R3.1: Past work has shown that prior preferences can bias/determine choices. This effect might have played a role during the choice task, which followed wanting, taste, health, and calorie ratings during which participants might have already formed their preferences. What are the authors' positions on such potential confound? How were the food images paired for the choice task in more detail?

The data reported here, were part of a larger experiment. Next to the food rating and choice task, participants also completed a social preference rating and choice task, as well as rating and choice tasks for intertemporal discounting. These tasks were counterbalanced such that first the three rating tasks were completed in counterbalanced order and second the three choice tasks were completed in the same order (e.g. food rating, social rating, intertemporal rating; food choice, social choice, intertemporal choice). This means that there were always two other tasks between the food rating and food choice task. In addition, to the temporal delay between rating and choice tasks, our modeling analyses revealed that models including a starting point bias performed worse than those without the bias. Although we cannot rule out that participants might occasionally have tried to make their decision before the actual task (e.g., by keeping their most/least preferred option in mind and then automatically choosing/rejecting it in the choice task), we think that both our design as well as our modeling analyses speak against any systematic bias of preference in our choice task. The options were paired such that approximately half of the trials were random, while for the other half one option was rated healthier and the other option was rated tastier (e.g., Sullivan & Huettel, 2021)

R3.2: In line with this thought, theoretically, the DDMs could also be fitted to reaction times and wanting ratings (binarized). This could be an excellent addition to corroborate the findings for choice behavior.

We have implemented several alternative modeling analyses, including taste vs health as defined by Nutri-Score (Table S12 and Figures S22-S30) and higher wanted choice vs healthy choice (Table S13; Figure S30-34). Indeed, these models corroborate those reported in the main text demonstrating the robustness of our findings.

R3.3: The principal component analysis was a good strategy for reducing the attribute space (taste, health, wanting, calories, Nutriscore, objective calories) into two components. Still, somehow, this part of the results added confusion to harnessing in which of the analyses the health attribute corresponded only to the healthiness ratings and taste to the tastiness ratings and if and when the components were used as attributes. This source of confusion could be mitigated by more clearly stating what health and taste corresponded to in each of the analyses.

We thank the reviewer for this recommendation and have now reported the PCA before reporting the behavioural results to clarify that choices are binarized based on participants’ taste and health ratings, rather than the composite scores. We have chosen this approach, as it is closer to our hypotheses and improves interpretability.

R3.4: From the methods, it seems that 66 food images were used, and 39 fell into A, B, C, and D Nutriscores. How were the remaining 27 images selected, and how healthy and tasty were the food stimuli overall?

The selection of food stimuli was done in three steps: First, from Charbonnier and collegues (2016) standardized food image database (available at https://osf.io/cx7tp/) we excluded food items that were not familiar in Germany/unavailable in regular German supermarkets. Second, we excluded products that we would not be able to incentivize easily (i.e., fastfood, pastries and items that required cooking/baking/other types of preparation). Third, we added the Nutri Scores to the remaining products aiming to have an equal number of items for each Nutri-Score, of which approximately half of the items were sweet and the other half savory. This resulted in a final stimuli-set of 66 food images (13 items = A; 13 items=B; 12 items=C; 14 items = D; 14 items = E). The experiment with including the set of food stimuli used in our study is also uploaded here: https://osf.io/pef9t/.With respect to the second question, we would like to point out that preference of food stimuli is very individual, therefore we obtained the ratings (taste, health, wanting and estimated caloric density) of each participant individually. However, we also added the objective total calories, which is positively correlated subjective caloric density and negatively correlated with Nutri-Score (coded as A=5; B=4; C=3; D=2; E=1) and health ratings (see Figure S7).

R3.5: It seems that the degrees of freedom for the paired t-test comparing the effects of the condition hungry versus satiated on hunger ratings were 63, although the participant sample counted 70. Please verify.

This is correct and explained in the methods section under data analysis: “Due to missing values for one timepoint in six participants (these participants did not fill in the VAS and PANAS before the administration of the Protein Shake in the sated condition) the analyses of the hunger state manipulation had a sample size of 64.”

R3.5: Please add the range of BMI and age of participants. Did all participants fall within a healthy BMI range

The BMI ranged from 17.306 to 48.684 (see Author response image 5), with the majority of participants falling within a normal BMI (i.e., between 18.5 and 24.9). In our sample, 3 participants had a BMI lager than 30. By using subject as a random intercept in our GLMMs we accounted for potential deviations in their response.

**Author response image 5. sa4fig5:** 

R3.5: Defining the inference criterion used for the significance of the posterior parameter chains in more detail can be pedagogical for those new to or unfamiliar with inferences drawn from hierarchical Bayesian model estimations and Bayesian statistics.

We have added an explanation of the highest density intervals and what they mean with respect to our data in the respective result section.